# RHOJ controls EMT-associated resistance to chemotherapy

Maud Debaugnies[1,2], Sara Rodríguez-Acebes[3], Jeremy Blondeau[1], Marie-Astrid Parent[1], Manuel Zocco[1], Yura Song[1], Viviane de Maertelaer[4,5], Virginie Moers[1], Mathilde Latil[1], Christine Dubois[1], Katia Coulonval[4,5], Francis Impens[6], Delphi Van Haver[6], Sara Dufour[6], Akiyoshi Uemura[7], Panagiota A. Sotiropoulou[1], Juan Méndez[3] & Cédric Blanpain[1,8 ✉]

The resistance of cancer cells to therapy is responsible for the death of most patients with cancer[1]. Epithelial-to-mesenchymal transition (EMT) has been associated with resistance to therapy in different cancer cells[2,3]. However, the mechanisms by which EMT mediates resistance to therapy remain poorly understood. Here, using a mouse model of skin squamous cell carcinoma undergoing spontaneous EMT during tumorigenesis, we found that EMT tumour cells are highly resistant to a wide range of anti-cancer therapies both in vivo and in vitro. Using gain and loss of function studies in vitro and in vivo, we found that RHOJ—a small GTPase that is preferentially expressed in EMT cancer cells—controls resistance to therapy. Using genome-wide transcriptomic and proteomic profiling, we found that RHOJ regulates EMT-associated resistance to chemotherapy by enhancing the response to replicative stress and activating the DNA-damage response, enabling tumour cells to rapidly repair DNA lesions induced by chemotherapy. RHOJ interacts with proteins that regulate nuclear actin, and inhibition of actin polymerization sensitizes EMT tumour cells to chemotherapy-induced cell death in a RHOJ-dependent manner. Together, our study uncovers the role and the mechanisms through which RHOJ acts as a key regulator of EMT-associated resistance to chemotherapy.

Cancer is one of the main causes of mortality worldwide and resistance to therapy is responsible for treatment failure in the majority of patients[1]. EMT is a developmental process in which epithelial cancer cells lose cell–cell adhesion and acquire mesenchymal features, including increased invasiveness and motility[4]. EMT in cancer controls tumour initiation, progression, metastasis and resistance to anti-cancer therapies[2,5]. Transcription factors that promote EMT, such as Zeb1, Zeb2 and Twist1, mediate resistance to chemotherapy and radiotherapy in different cancer cell lines in vitro and after their transplantation into immunocompromised mice[6,7]. The conditional deletion of Twist1 or Snai1 in a mouse model of pancreatic adenocarcinoma sensitizes tumour cells to gemcitabine[8]. EMT in a mammary tumour model promotes resistance to different chemotherapeutic drugs[9]. Several mechanisms have been proposed to account for the resistance to therapy associated with EMT in cancer cell lines in vitro[3,8,9]. However, whether these mechanisms are responsible for EMT-induced resistance to therapy in primary tumours in vivo remains unclear.

Here we used a genetic mouse model of squamous cell carcinoma (SCC) presenting spontaneous EMT to determine the functional response of different cell populations to chemotherapy. We found that, in primary skin SCCs presenting EMT, EPCAM⁻ tumour cells were highly resistant to chemotherapy. Using genetic gain and loss of function studies, we identified RHOJ, a small Rho GTPase, as a key regulator that promotes resistance to a broad range of chemotherapeutic agents in EMT tumour cells. Proteomic analysis combined with functional assays revealed that RHOJ controls the formation of nuclear F-actin fibres and the activation of dormant origins of replication, which promotes DNA repair and prevents chemotherapy induced cell death.

### EMT-associated chemotherapy resistance in SCC

To investigate the role and the mechanisms of resistance to therapy in EMT tumour cells in vivo, we used a genetically induced mouse model of skin SCCs combining the expression of oncogenic Kras ($Kras^{G12D}$) with the deletion of Trp53 and the expression of a YFP reporter in hair follicle lineages[10] ($Lgr5^{creER}Kras^{G12D}Trp53^{cKO}Rosa\text{-}YFP$). The YFP reporter, combined with EPCAM staining, enables us to identify tumour cells that undergo EMT and lose the expression of epithelial markers (YFP⁺EPCAM⁻) (Extended Data Fig. 1a).

To assess whether EMT is associated with resistance to chemotherapy in skin SCC, we treated $Lgr5^{creER}Kras^{G12D}Trp53^{cKO}Rosa\text{-}YFP$ mice that were induced with tamoxifen and presented primary SCCs with cisplatin and 5-fluorouracil (5FU), the standard chemotherapy used to treat human patients with metastatic SCCs[11]. About 14% of SCCs were very sensitive

[1]Laboratory of Stem Cells and Cancer, Université Libre de Buxelles (ULB), Brussels, Belgium. [2]CHU Saint-Pierre, Université Libre de Bruxelles (ULB), Brussels, Belgium. [3]DNA Replication Group, Molecular Oncology Programme, Spanish National Cancer Research Centre, Madrid, Spain. [4]Institute of Interdisciplinary Research (IRIBHM), Université Libre de Bruxelles (ULB), Brussels, Belgium. [5]ULB-Cancer Research Center (U-crc), Université Libre de Bruxelles (ULB), Brussels, Belgium. [6]VIB Center for Medical Biotechnology, VIB Proteomics Core, Department of Biomolecular Medicine, Ghent University, Ghent, Belgium. [7]Department of Retinal Vascular Biology, Nagoya City University Graduate School of Medical Sciences, Nagoya, Japan. [8]WELBIO, Université Libre de Bruxelles (ULB), Brussels, Belgium. ✉e-mail: Cedric.Blanpain@ulb.be

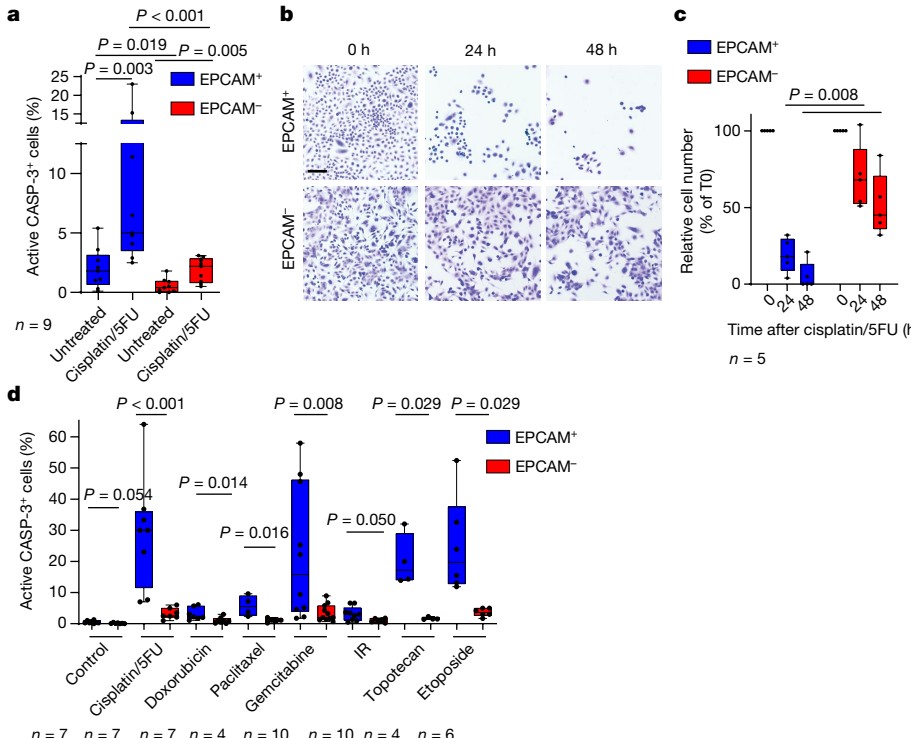

**Fig. 1 | EMT is associated with intrinsic resistance to chemotherapy in SCCs.**
**a**, FACS quantification of the percentage of active caspase-3-positive cells in YFP⁺EPCAM⁺ and YFP⁺EPCAM⁻ cells from control and 24 h cisplatin/5FU-treated SCCs. $n$ = 9 SCCs. **b,c**, Microscopy analysis of the cellular density of EPCAM⁺ and EPCAM⁻ tumour cells (**b**) and quantification of the number of cells (**c**) 24 h and 48 h after cisplatin/5FU administration in vitro. Scale bar, 100 μm. $n$ = 5 independent experiments. **d**, FACS quantification of the percentage of activated caspase-3-positive cells in YFP⁺EPCAM⁺ and YFP⁺EPCAM⁻ cells 24 h

after cisplatin/5FU administration ($n$ = 7); doxorubicin ($n$ = 7), paclitaxel ($n$ = 4), gemcitabine ($n$ = 10), topotecan ($n$ = 4) or etoposide ($n$ = 6) administration; and 24 h after receiving 10 Gy irradiation. $n$ values represent the number of biological independent experiments. For **a**, **c** and **d**, statistical analysis was performed using two-sided Mann–Whitney $U$-tests. $P$ values were adjusted for multiple comparisons using Bonferroni correction. For the box plots, the centre line shows the median, the box limits represent the 25th and 75th percentiles, and the whiskers show the minimum and maximum values.

to chemotherapy and completely disappeared after 4 weeks of treatment (fully responding tumours). About 32% of SCCs did not respond and progressed during chemotherapy (non-responding tumours). Most SCCs (54%) presented a partial response to chemotherapy. Using immunostaining and fluorescence-activated cell sorting (FACS) analysis, we found that responding tumours consisted of YFP⁺ epithelial tumour cells expressing the epithelial marker K14 or a mixture of epithelial and EMT tumour cells. The tumours that did not respond were enriched for EMT tumour cells (K14⁻VIM⁺ or EPCAM⁻) (Extended Data Fig. 1b–j). These data suggest that chemotherapy either preferentially kills EPCAM⁺ tumour cells or induces EMT promoting the switch from EPCAM⁺ to EPCAM⁻ tumour cells during chemotherapy.

To distinguish between these two possibilities, we investigated apoptosis in EPCAM⁺ and EPCAM⁻ tumour cells 24 h after chemotherapy administration. EPCAM⁻ tumour cells were very resistant to chemotherapy-induced cell death in vivo as only 2% of EPCAM⁻ tumour cells were caspase-3 positive compared with 9% of epithelial EPCAM⁺ tumour cells. We recently identified EMT subpopulations of tumour cells presenting different degrees of EMT in mouse skin SCCs[12]. We found that the different EMT subpopulations were similarly resistant to chemotherapy (Fig. 1a and Extended Data Fig. 1k–p). Similar to what we found in vivo, EMT tumour cells were profoundly resistant to chemotherapy in vitro in the absence of the tumour microenvironment, whereas EPCAM⁺ epithelial tumour cells were even more sensitive (Fig. 1b,c). In addition to cisplatin/5FU treatment, EPCAM⁻ tumour cells were resistant to the topoisomerase I inhibitor topotecan, the topoisomerase II inhibitor etoposide and the anti-metabolite gemcitabine. EPCAM⁻ tumour cells were also resistant to the anti-microtubule

paclitaxel, the DNA intercalant and topoisomerase II inhibitor doxorubicin and ionizing radiation, although these treatments were less efficient in inducing cell death in EPCAM⁺ tumour cells (Fig. 1d). Together, these data demonstrate that EMT in skin SCCs is associated with resistance to a wide range of genotoxic drugs in a cellular autonomous manner both in vivo and in vitro.

## RHOJ mediates EMT-associated resistance to therapy

To define the mechanisms by which EMT tumour cells are resistant to chemotherapy, we assessed the expression of putative mediators of resistance to therapy in EPCAM⁺ and EPCAM⁻ tumour cells from skin SCCs[10,12]. We found that EMT tumour cells expressed higher levels of apoptosis inhibitors (*Birc2, Birc3)*, glutathione-metabolism-related genes (*Gpx3*, *Gpx7*, *Gpx8*, *Gstm2*) and ABC multidrug transporters (*Abcb1a*, *Abcb1b*, *Abcg2*). We also found that RHOJ—a small GTPase of the Cdc42 subfamily that was previously described to mediate resistance to therapy in melanoma[13,14]—was expressed at a much higher level in EMT tumour cells (Fig. 2a and Extended Data Fig. 2a). To investigate whether RHOJ contributes to EMT-associated resistance to therapy in skin SCCs, we first assessed whether short hairpin RNA (shRNA)-mediated *Rhoj* knockdown (KD) sensitized EMT tumour cells to chemotherapy. KD of *Rhoj* but not *Rhoq*, its closest homologue, increased apoptosis after cisplatin/5FU treatment in EMT tumour cells, which was accompanied by a strong decrease in living tumour cells 48 h after chemotherapy. shRNA KD of *Rhoj* and *Rhoq* in EPCAM⁺ tumour cells did not change their response to chemotherapy (Fig. 2b,c and Extended Data Fig. 2b–d). *Rhoj* KD in EMT tumour cells was associated with a slight decrease in cell

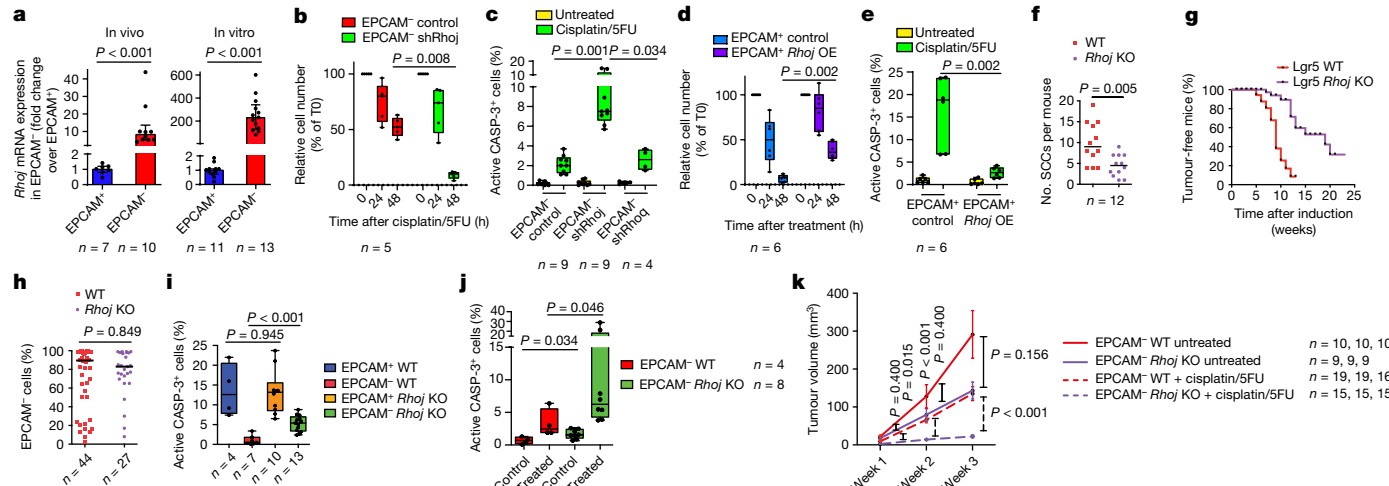

**Fig. 2 | RHOJ mediates resistance to chemotherapy associated with EMT.**
**a**, RT−qPCR analysis of *Rhoj* expression in EPCAM⁻ cells compared with in EPCAM⁺ cells in vivo (left) and in vitro (right). Data are median ± interquartile range, normalized to the housekeeping gene *Tbp*. *n* values indicate the number of independent experiments. **b**, The relative cell number of EPCAM⁻ control and *Rhoj* shRNA KD cells 24 h and 48 h after cisplatin/5FU administration in vitro. *n* values indicate the number of independent experiments. **c**, FACS quantification of the percentage of activated caspase-3-positive cells in EPCAM⁻ control, *Rhoj* shRNA KD and *Rhoq* shRNA KD cells 24 h after cisplatin/5FU administration. *n* values indicate the number of independent experiments. **d**,**e**, The relative cell number (**d**) and FACS quantification of the percentage of activated caspase-3-positive cells (**e**) of control and RHOJ-overexpressing (OE) EPCAM⁺ cells 24 h and 48 h after cisplatin/5FU administration in vitro. *n* = 6 independent experiments. **f**,**g**, The median number of SCCs per mouse (**f**) and kinetics of tumour appearance (**g**) in control and *Rhoj*-KO mice. *n* values indicate the

number of mice. **h**, FACS quantification of the percentage of EPCAM⁻ cells of tumours from WT and *Rhoj*-KO SCCs. Data are median. *n* values indicate the number of mice. **i**,**j**, FACS quantification of the percentage of activated caspase-3-positive cells in EPCAM⁺ and EPCAM⁻ cells 24 h after cisplatin/5FU administration from WT (*n* = 4 and 7 SCCs) and *Rhoj*-KO mice (*n* = 10 and 13 SCCs) in vivo (**i**) and in EPCAM⁻ cells from WT-derived (*n* = 4) and *Rhoj*-KO-derived (*n* = 8) cell lines 24 h after cisplatin/5FU administration in vitro (**j**). *n* values indicate the number of independent experiments. **k**, The tumour volume after cisplatin/5FU administration to mice that were grafted with WT and *Rhoj*-KO tumour cells compared with untreated mice. Data are mean ± s.e.m. *n* values indicate the number of mice. For **a**–**f** and **h**–**k**, statistical analysis was performed using two-sided Mann−Whitney *U*-tests. *P* values were adjusted using Bonferroni correction when multiple comparisons were performed. For the box plots, the centre line shows the median, the box limits represent the 25th and 75th percentiles, and the whiskers show the minimum and maximum values.

growth in vitro and migration consistent with previous findings[15–17] and did not affect the expression of EMT or epithelial markers (Extended Data Fig. 2e–h).

Next, we assessed whether RHOJ overexpression can confer resistance to chemotherapy in sensitive epithelial tumour cells. RHOJ overexpression in EPCAM⁺ tumour cells strongly decreased the proportion of apoptotic tumour cells 24 h after cisplatin/5FU administration and leads to increased cell survival 48 h after chemotherapy (Fig. 2d,e and Extended Data Fig. 3a,b). The overexpression of RHOJ in EPCAM⁺ tumour cells slightly decreased the growth of cells in vitro and did not affect the expression of EMT or epithelial markers (Extended Data Fig. 2h and 3c).

To determine whether RHOJ also mediates EMT resistance to therapy in vivo, we generated *Rhoj* conditional knockout (*Rhoj* KO) tumours using *Lgr5^{creER}Kras^{G12D}p53^{cKO}Rhoj^{cKO}Rosa-YFP* mice (Extended Data Fig. 3d,e). *Rhoj* KO slightly decreased the number of tumours per mouse and increased their latency, suggesting that RHOJ promotes SCC initiation (Fig. 2f,g). The proportion of EPCAM⁺ to EPCAM⁻ tumour cells was unchanged after *Rhoj* deletion (Fig. 2h and Extended Data Fig. 3f), showing that RHOJ does not control EMT per se. *Rhoj*-KO tumour cells presented a decrease in proliferation as shown by Ki-67 immunostaining (Extended Data Fig. 3g,h). Administration of cisplatin/5FU to mice with tumours showed that *Rhoj*-KO tumours presented increased cell death in EPCAM⁻ tumour cells (Fig. 2i and Extended Data Fig. 3i). Administration of chemotherapy in cultured cells in vitro showed that *Rhoj*-KO EPCAM⁻ tumour cells were also more sensitive to cisplatin/5FU (Fig. 2j). To assess whether *Rhoj* deletion affects the long-term response of tumours to chemotherapy, we transplanted EPCAM⁻ *Rhoj*-WT and *Rhoj*-KO tumour cells into immunodeficient mice and treated mice presenting growing tumours with chemotherapy. In the absence

of chemotherapy, *Rhoj*-WT EPCAM⁻ tumours grew a bit faster compared with *Rhoj*-KO EPCAM⁻ tumours, consistent with their respective growth in vitro. After cisplatin/5FU administration, *Rhoj*-WT EPCAM⁻ tumours continued to grow, although more slowly than in the absence of chemotherapy. By contrast, the size of *Rhoj*-KO EPCAM⁻ tumours was stable during the 3 weeks of cisplatin/5FU treatment, showing the short-term and long-term sensitization to chemotherapy after *Rhoj* deletion (Fig. 2k).

To assess the relevance of our findings to human cancers, we generated *RHOJ* KD using short hairpin RNA in MDA-MB-231 cells— a human triple-negative breast cancer cell line presenting EMT that expresses high levels of RHOJ and displaying p53 mutations. *RHOJ* KD in MDA-MB-231 cells decreased cell growth and increased cell death after administration of chemotherapy (Extended Data Fig. 4a–d).

## Transcriptome and proteome control by RHOJ

To define mechanistically how RHOJ promotes resistance to therapy in EMT tumour cells, we first performed bulk RNA-sequencing (RNA-seq) analysis of shRNA control and *Rhoj*-KD EPCAM⁻ tumour cells. When considering genes that were deregulated by twofold, 487 genes were downregulated, and 460 genes were upregulated by *Rhoj* KD in EPCAM⁻ cells. Gene Ontology analysis (GO) revealed that the genes downregulated in *Rhoj*-KD EPCAM⁻ cells are involved in DNA replication, cellular response to interferon and actin remodelling, whereas genes implicated in cell adhesion, lipid metabolism, apoptosis and microtubule and actin cytoskeleton were upregulated in *Rhoj*-KD EPCAM⁻ cells (Extended Data Fig. 5a–d).

In addition to the transcriptional regulation of the cellular effectors of DNA damage response (DDR) that accompanied genotoxic stress

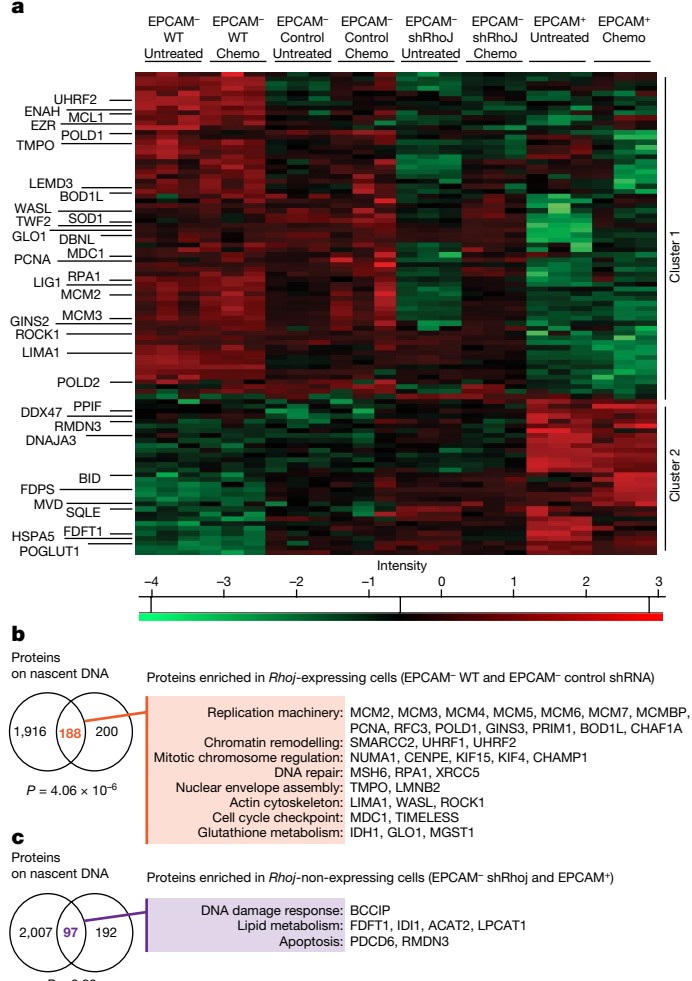

**a**

EPCAM⁻ WT Untreated | EPCAM⁻ WT Chemo | EPCAM⁻ Control Untreated | EPCAM⁻ Control Chemo | EPCAM⁻ shRhoJ Untreated | EPCAM⁻ shRhoJ Chemo | EPCAM⁺ Untreated | EPCAM⁺ Chemo

Cluster 1
Cluster 2

UHRF2
ENAH — MCL1
EZR — POLD1
TMPO

LEMD3
— BOD1L
WASL
TWF2 — SOD1
GLO1 — DBNL
PCNA — MDC1
LIG1 — RPA1
— MCM2
GINS2 — MCM3
ROCK1
LIMA1

POLD2

DDX47 — PPIF
RMDN3
DNAJA3

— BID
FDPS
MVD — SQLE
HSPA5 — FDFT1
POGLUT1

Intensity
−4  −3  −2  −1  0  1  2  3

**b**

Proteins on nascent DNA

1,916 | **188** | 200

$P = 4.06 \times 10^{-6}$

Proteins enriched in *Rhoj*-expressing cells (EPCAM⁻ WT and EPCAM⁻ control shRNA)

Replication machinery: MCM2, MCM3, MCM4, MCM5, MCM6, MCM7, MCMBP, PCNA, RFC3, POLD1, GINS3, PRIM1, BOD1L, CHAF1A
Chromatin remodelling: SMARCC2, UHRF1, UHRF2
Mitotic chromosome regulation: NUMA1, CENPE, KIF15, KIF4, CHAMP1
DNA repair: MSH6, RPA1, XRCC5
Nuclear envelope assembly: TMPO, LMNB2
Actin cytoskeleton: LIMA1, WASL, ROCK1
Cell cycle checkpoint: MDC1, TIMELESS
Glutathione metabolism: IDH1, GLO1, MGST1

**c**

Proteins on nascent DNA

2,007 | **97** | 192

$P = 0.93$

Proteins enriched in *Rhoj*-non-expressing cells (EPCAM⁻ shRhoj and EPCAM⁺)

DNA damage response: BCCIP
Lipid metabolism: FDFT1, IDI1, ACAT2, LPCAT1
Apoptosis: PDCD6, RMDN3

**Fig. 3 | Proteomic analysis after *Rhoj* KO in EMT tumour cells. a**, The relative intensities of selected upregulated proteins identified in RHOJ-expressing cells (EPCAM⁻ WT and EPCAM⁻ control shRNA) and in RHOJ-non-expressing cells (EPCAM⁺ and EPCAM⁻ *Rhoj* shRNA KD) in untreated conditions and after 24 h of cisplatin/5FU treatment. Two major clusters were observed. *n* = 3 technical replicates. *Z*-scored log₂-transformed label-free quantitation (LFQ) intensities are shown. Statistical analysis was performed using two-way ANOVA; *P* < 0.05. Individual *P* values are provided as Source Data. **b,c**, The overlap between a published proteomic study of proteins associated with replicative fork (iPond)[20] and the proteins enriched in RHOJ-expressing cells (**b**) or the proteins enriched in RHOJ-non-expressing cells (**c**) in the untreated condition and 24 h after cisplatin/5FU administration. The *P* value was calculated using the hypergeometric test.

induced by chemotherapy, DDR factors are regulated at the protein level by protein stability, and post-translational modifications that control DNA repair and cell survival[18]. To define the effect of RHOJ on protein expression in EMT tumour cells, we performed proteomics analysis of EPCAM⁺ and EPCAM⁻ FACS-isolated primary tumour cells from *Lgr5^creER Kras^G12D p53^cKO Rosa-YFP* mice and YFP⁺EPCAM⁻ cells that were transduced with viruses expressing control or *Rhoj* shRNA in untreated conditions and 24 h after cisplatin/5FU administration. We identified the proteins of which the expression level was significantly regulated after *Rhoj* KD (*P* < 0.05, two-way analysis of variance (ANOVA)) and found that 681 proteins were significantly differentially regulated between RHOJ-expressing and non-expressing samples. *Rhoj*-KD tumour cells presented decreased expression of nuclear proteins, actin regulators, glutathione metabolism and proteins implicated in DNA replication. By contrast, proteins implicated in endoplasmic reticulum metabolic

processes, cholesterol metabolism and apoptosis were upregulated in *Rhoj*-KD tumour cells (Fig. 3a and Extended Data Fig. 5e,f). Phalloidin staining showed an increase in the number of actin stress fibres after *Rhoj* KD, consistent with previous reports[16,19] (Extended Data Fig. 5g). Using western blotting, we validated the higher expression of proteins regulating DNA replication (POLD, PCNA) and repair (pRPA2) as well as actin cytoskeleton (N-WASP) in EPCAM⁻ tumour cells compared with in EPCAM⁺ tumour cells, and their downregulation in EPCAM⁻ *Rhoj*-KO cells (Extended Data Fig. 5h–k). The deletion of *Rhoj* in EPCAM⁻ tumour cells did not affect the expression of EMT or epithelial markers (Extended Data Fig. 5l,m).

Importantly, the proteins differentially regulated by *Rhoj* KD were strongly enriched for proteins found on newly replicated DNA after replicative stress (iPOND)[20] (Fig. 3b,c). These included several replisome components, such as the helicase MCM2–7; different subunits of the DNA polymerases and their accessory factors (such as PCNA, PRIM1, POLD1); components of the PCNA-loading replication factor C (RFC3); protective factors of stalled replication forks (such as BODL1 and TIMELESS); initiation factors (GINS3); and chromatin assembly factors (CHAF1A). Other proteins identified in the proteomics assays included chromatin-remodelling factors (SMARCC2, UHRF1, UHRF2); intra-S-phase checkpoint signalling mediator (MDC1); spindle-associated proteins (such as CHAMP1, KIF15, KIF4, NUMA1, CENPE); proteins implicated in nuclear envelope assembly, including LEM protein (TMPO) and lamin (LMNB2); established RHOJ partners (WASL and ROCK1)[21]; single-strand DNA-binding protein RPA1; and different DNA repair factors, such as the core-component of non-homologous end-joining XRCC5 and the component of post-replicative DNA mismatch repair MSH6. The important overlap between proteins regulated by RHOJ expression and proteins associated with nascent DNA and/or recruited during replicative stress suggests that RHOJ controls DNA replication and DNA repair.

## RHOJ promotes DNA repair in EMT tumour cells

To assess whether chemotherapy in EPCAM⁺ and EPCAM⁻ as well as EPCAM⁻ *Rhoj*-KO tumour cells induces DNA damage and differentially activates the DDR pathways, we investigated the level of phosphorylation of ATM and ATR substrates, two main kinases that activate the DDR after DNA damage[22]. EPCAM⁺, EPCAM⁻ and EPCAM⁻ *Rhoj*-KO tumour cells similarly increased the level of phosphorylated ATM/ATR substrates after cisplatin/5FU administration (Fig. 4a), showing that RHOJ does not control ATM and ATR activation after chemotherapy. ATR and ATM inhibitors did not sensitize EPCAM⁻ cells to chemotherapy (Extended Data Fig. 6a–c), suggesting that EPCAM⁻ resistance to therapy is independent of ATR- and ATM-mediated checkpoint. To determine whether RHOJ inhibits EMT tumour cells apoptosis by promoting DNA repair and/or replicative tolerance to damaged DNA, as suggested by the proteomic data, we first investigated the level of histone 2A.X phosphorylated at Ser139 (γ-H2AX), a mark of DNA damage after chemotherapy. Western blot, immunofluorescence and FACS quantification showed that EPCAM⁻ cells presented a significant reduction in γ-H2AX compared with EPCAM⁺ and EPCAM⁻ *Rhoj*-KO tumour cells after cisplatin/5FU administration (Fig. 4b,c and Extended Data Fig. 6d,e), supporting the notion that RHOJ prevents accumulation of DNA damage in EMT tumour cells.

To assess at which step of the DDR pathway RHOJ reduces the extent of DNA damage or regulates DNA repair, we analysed the expression and subcellular localization of key mediators of the DDR (53BP1, RPA2 and RAD51) after cisplatin administration using immunofluorescence microscopy. Clustering of 53BP1 at the site of DNA damage is an early event in the DDR[23] and acts together with other cofactors including RIF1 to block the over-resection of DNA at the break[24]. We found that, 12 h after cisplatin administration, 53BP1 nuclear foci were more abundant in EPCAM⁻ tumour cells compared with in EPCAM⁺ and EPCAM⁻ *Rhoj*-KD

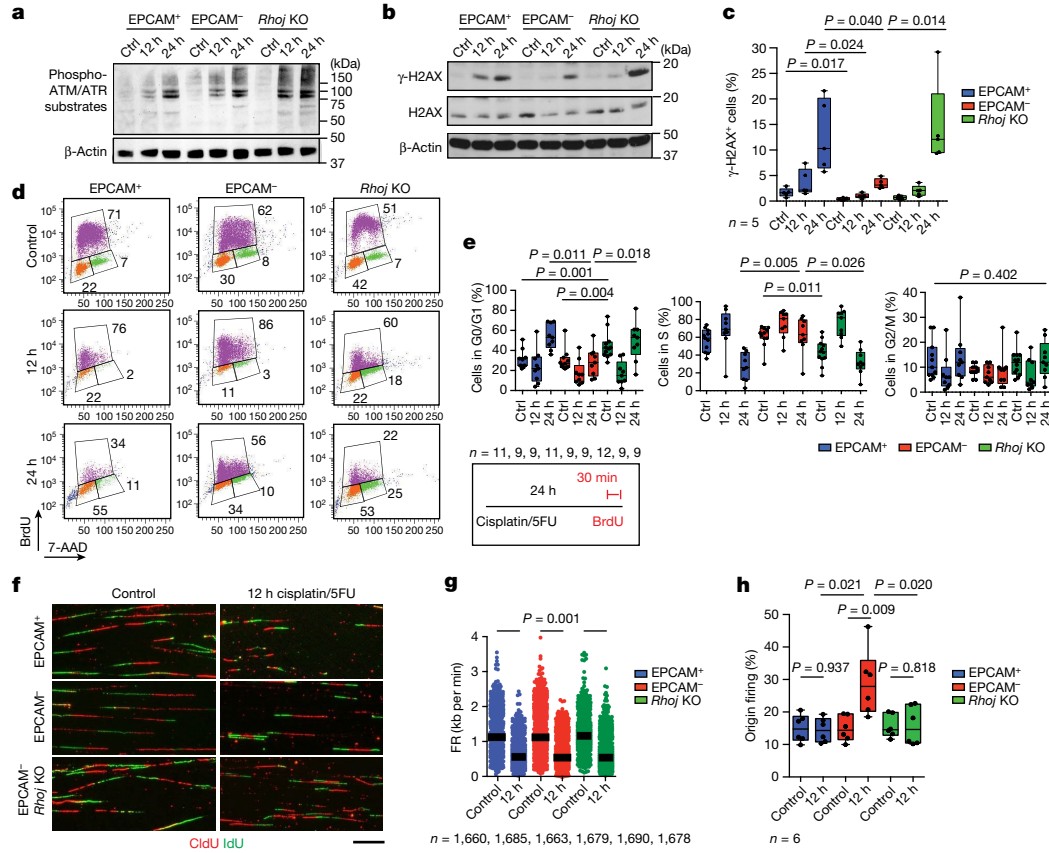

**Fig. 4 | RHOJ promotes DNA repair and the activation of new origins of DNA replication after chemotherapy in EMT tumour cells. a**, Western blot analysis of phosphorylation of ATM/ATR substrates in EPCAM⁺, EPCAM⁻ and *Rhoj*-KO EPCAM⁻ cells untreated and treated with cisplatin/5FU for 12 h and 24 h. $n = 3$. **b**,**c**, Western blot analysis (**b**; $n = 3$) and FACS analysis quantification (**c**; $n = 5$) of the percentage of cells expressing γ-H2AX in EPCAM⁺, EPCAM⁻ and EPCAM⁻ *Rhoj*-KO cells, 12 h and 24 h after cisplatin/5FU administration. β-Actin was the loading control. *n* values represent the number of independent primary cultured cell lines. **d**,**e**, Representative cell cycle FACS profile (**d**) and quantification (**e**) of the percentage of cells in G0/G1, S and G2/M phase 12 h and 24 h after cisplatin/5FU treatment of EPCAM⁺, EPCAM⁻ and EPCAM⁻ *Rhoj*-KO BrdU-7AAD-labelled cells. *n* values indicate the number of independent experiments. **f**, Representative stretched DNA fibres labelled with CldU and IdU from EPCAM⁺, EPCAM⁻ and EPCAM⁻ *Rhoj*-KO cells untreated and treated with cisplatin/5FU for 12 h.

$n = 6$. Scale bar, 10 μm. **g**, Fork rate (FR) values in EPCAM⁺, EPCAM⁻ and EPCAM⁻ *Rhoj*-KO cells untreated and treated with cisplatin/5FU for 12 h. Data are median. Statistical analysis was performed using ANOVA, with condition, experience and their interaction; the *P* value was adjusted using the two-way post hoc Sidak test. *n* values represent the number of forks pooled from six independent experiments. **h**, The percentage of origin of firing in EPCAM⁺, EPCAM⁻ and EPCAM⁻ *Rhoj*-KO cells untreated and treated with cisplatin/5FU for 12 h. From left to right, total number of DNA fibres scored: $n = 3,020, 3,123, 3,065, 3,091, 3,078$ and 3,069, in six independent experiments. For **c**, **e** and **h**, statistical analysis was performed using Kruskal–Wallis tests followed by two-sided Mann–Whitney *U*-tests; *P* values were adjusted for multiple comparisons using Bonferroni correction. For the box plots, the centre line shows the median, the box limits represent the 25th and 75th percentiles, and the whiskers show the minimum and maximum values.

cells. RPA2 focus formation is associated with single-stranded DNA generated by DNA end-resection at damage sites and stalled replication forks after replicative stress[25,26]. RAD51 focus formation after DNA damage is associated with homologous-recombination-mediated DNA repair of double-stranded breaks and is recruited at stalled forks generated during replicative stress to mediate fork reversal and protect the DNA from nucleolytic degradation[27]. The number of RPA2 and RAD51 nuclear foci was lower in EPCAM⁻ cells compared with in EPCAM⁺ and EPCAM⁻ *Rhoj*-KD cells (Extended Data Fig. 6f). These results are consistent with an increase in DNA repair and a decrease in replicative stress in the presence of RHOJ in EPCAM⁻ tumour cells, in good accordance with the γ-H2AX data.

## RHOJ actives new origins of DNA replication

To determine whether the cell cycle is differentially regulated in EPCAM⁺, EPCAM⁻ and EPCAM⁻ *Rhoj*-KD tumour cells in response to chemotherapy, we assessed DNA content and BrdU incorporation using BrdU/7AAD FACS analysis. The cell cycle profile of EPCAM⁺ and

EPCAM⁻ cells was comparable in the untreated condition, whereas *Rhoj*-KO EPCAM⁻ tumour cells presented an increase in G0/G1 at the expense of S phase. At 12 h after cisplatin/5FU treatment, the proportion of BrdU-positive cells increased in all of the conditions. However, 24 h after cisplatin/5FU administration, the percentage of cells in S phase was strongly reduced in EPCAM⁺ and *Rhoj*-KO tumour cells, whereas EPCAM⁻ tumour cells continued to synthesize DNA, as revealed by BrdU incorporation (Fig. 4d,e). Genotoxic agents cause DNA damage in replicative cells by increasing replicative stress, which in turn activates intra-S-phase checkpoint pathways to facilitate replication completion and avoid double-stranded break formation arising from fork collapse, thereby limiting cell death[28]. The level of EdU incorporation was greatly reduced in all tumour cells 24 h after cisplatin/5FU administration, consistent with the activation of the intra-S-phase checkpoint (Extended Data Fig. 7a,b). ATR kinase is a major regulator of the replicative stress checkpoint, whereby it targets different proteins— including CHK1 and WEE1, which control cyclin-dependent kinase (CDK) activity—to activate cell cycle arrest in S phase and prevent mitotic catastrophe[29]. Higher activation of CDKs, including CDK1 and CDK4,

was observed in EPCAM⁻ tumour cells compared with in EPCAM⁺ and *Rhoj*-KO EPCAM⁻ cells after chemotherapy (Extended Data Fig. 7c–f) suggesting that RHOJ enables EMT tumour cells to progress in the cell cycle and continue DNA synthesis after chemotherapy.

To test whether the higher resistance of EPCAM⁻ tumour cells to chemotherapy is related to the higher ability to cope with replicative stress, we evaluated the sensitivity of EPCAM⁺, EPCAM⁻ and *Rhoj*-KO tumour cells to the replicative-stress-inducing drug aphidicolin. Aphidicolin is a selective inhibitor of DNA polymerases that induces fork stalling and single-strand DNA exposure by uncoupling polymerase and helicase activities[30]. Aphidicolin induced a major increase in EPCAM⁺ and *Rhoj*-KO EPCAM⁻ cell death. By contrast, EPCAM⁻ tumour cells were relatively insensitive to aphidicolin treatment (Extended Data Fig. 7g). In the case of insufficient protection of stalled forks during replicative stress, the nascent DNA will be degraded by different nucleases (for example, DNA2 and MRE11), leading to fork collapse[31]. Treatment with mirin, a MRE11 inhibitor, after cisplatin/5FU administration decreased cell death in EPCAM⁺ and *Rhoj*-KO EPCAM⁻ cells but had no effect on EPCAM⁻ tumour cells (Extended Data Fig. 7h), suggesting that the absence of RHOJ leads to replicative stress after chemotherapy. There was an increase in the number of micronuclei in EPCAM⁺ and *Rhoj*-KO EPCAM⁻ cells compared with in EPCAM⁻ cells 12 h and 24 h after chemotherapy (Extended Data Fig. 8a,b), indicating unresolved replicative stress in the absence of RHOJ.

To understand how RHOJ promotes continued DNA replication (Fig. 4d,e) and DDR, we assessed the speed of replication forks and the activation of replication origins under basal conditions and after chemotherapy[32]. Using DNA fibre analysis after sequential pulse of CldU and IdU, we found that cisplatin/5FU treatment induced a global slowing of replication forks in all cell types, further demonstrating the activation of the S-phase checkpoint in all conditions (Fig. 4f,g). The activation of the S-phase checkpoint was the consequence of fork stalling, as shown by the increase in the asymmetry of fork progression in all cell types after chemotherapy (Extended Data Fig. 8c,d). By contrast, the number of origins was increased only in EPCAM⁻ cells as compared with EPCAM⁺ and *Rhoj*-KO EPCAM⁻ cells in response to cisplatin/5FU treatment, showing that chemotherapy induced the firing of new origins of replication in EPCAM⁻ cells (Fig. 4h). Dormant DNA replication origins are activated to resume replication in case of fork stalling during replicative stress. It has been demonstrated that the amount of MCM proteins affects the availability of dormant origins[33,34]. Our data show that EPCAM⁻ cells display a higher concentration of several MCM subunits on chromatin (Extended Data Fig. 8e), consistent with the ability of EPCAM⁻ tumour cells to promote the firing of dormant origins in response to chemotherapy.

## RHOJ regulates nuclear actin polymerization

To understand how RHOJ regulates DNA repair and the activation of origins of DNA replication after chemotherapy, we sought to identify the proteins that interact with RHOJ in EMT tumour cells. To this end, we performed affinity purification followed by mass spectrometry (AP–MS) using anti-HA pull-down of EPCAM⁻ cells expressing 3HA-tagged RHOJ under untreated conditions and 12 h after cisplatin/5FU administration. We identified 118 proteins that were significantly enriched in untreated HA-tagged-RHOJ-expressing cells and 98 proteins that were significantly enriched in treated HA-tagged-RHOJ-expressing cells. Among them, we found previously identified RHOJ-interacting proteins such as GPRC5A[21]. Notably, after chemotherapy treatment, there was an enrichment of proteins implicated in the regulation of actin filament dynamics (FLNB and TLN1) and IPO9—a protein that regulates nuclear import of actin[35] (Fig. 5a). Immunostaining of HA-tagged-RHOJ-expressing tumour cells showed the cytoplasmic, perinuclear and nuclear localization of RHOJ (Extended Data Fig. 9a). These findings suggest that RHOJ physically interacts with nuclear actin

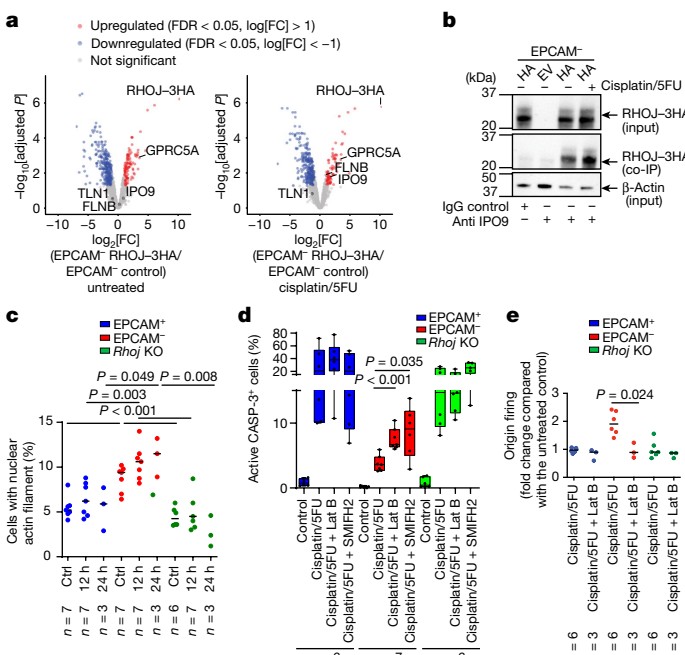

**Fig. 5 | RHOJ promotes cell survival and DNA repair by regulating nuclear actin polymerization. a**, Differential protein expression detected by AP–MS analysis of protein complexed with 3HA-tagged RHOJ in EPCAM⁻ cells. Volcano plot showing the statistical significance (−log₁₀-transformed *P*) and the fold change (log₂) of proteins identified by MS (false-discovery rate (FDR) < 0.05) in the untreated condition (left) and after 12 h of cisplatin/5FU treatment (right). Selected proteins enriched after immunoprecipitation of 3HA-tagged RHOJ are highlighted. **b**, Co-immunoprecipitation using anti-IPO9 or IgG control antibodies followed by western blotting using anti-HA antibodies, revealing the association between IPO9 and RHOJ–3HA protein in EPCAM⁻ cells 12 h after cisplatin/5FU treatment. *n* = 2. **c**, Quantification of the percentage of cells presenting nuclear actin filaments in EPCAM⁺, EPCAM⁻ and EPCAM⁻ *Rhoj*-KO cells expressing nuclear actin chromobody-GFP 12 h and 24 h after cisplatin/5FU treatment (>250 nuclei were quantified per replicate). Data are median. Statistical analysis was performed using ANOVA, with condition, experience and their interaction. *P* values were adjusted using two-way post hoc Sidak tests. *n* values indicate the number of replicates. **d**, FACS quantification of the percentage of activated caspase-3-positive cells in EPCAM⁺, EPCAM⁻ and EPCAM⁻ *Rhoj*-KO cells 24 h after treatment with cisplatin/5FU and latrunculin B (Lat B) or SMIFH2 inhibitor. For the box plots, the centre line shows the median, the box limits represent the 25th and 75th percentiles, and the whiskers show the minimum and maximum values. *n* values indicate the number of replicates. **e**, The relative proportion of origin firing in EPCAM⁺, EPCAM⁻ and EPCAM⁻ *Rhoj*-KO cells treated with cisplatin/5FU or a combination of cisplatin/5FU and latrunculin B for 12 h compared with the untreated condition. Data are median. *n* values indicate the number of independent experiments. For **d** and **e**, statistical analysis was performed using Kruskal–Wallis tests followed by two-sided Mann–Whitney *U*-tests. *P* values were adjusted for multiple comparisons using Bonferroni correction. Gel source data are provided in Supplementary Fig. 1.

regulators. Nuclear actin has been shown to regulate DNA replication and repair[36–41] and KD of *IPO9* has been shown to decrease nuclear actin filaments and impair DNA repair[37]. Co-immunoprecipitation experiments confirmed the presence of IPO9 in the RHOJ protein complex (Fig. 5b), suggesting that RHOJ regulates the response to chemotherapy through the modulation of nuclear actin function.

To test this possibility, we investigated the presence of actin fibres in the nucleus using transfection of nuclear actin chromobody in EPCAM⁺, EPCAM⁻ and *Rhoj*-KO tumour cells isolated from *Lgr5^creER^Kras^G12D^ Trp53^cKO^Rosa-tdTomato* mice. The proportion of cells with nuclear actin fibres was increased in EPCAM⁻ tumour cells compared with in EPCAM⁺

and *Rhoj*-KO tumour cells in the presence or absence of cisplatin/5FU, showing that RHOJ controls nuclear actin polymerization (Fig. 5c). Different patterns of nuclear actin filaments were identified. After chemotherapy, there was an increase in elongated actin filaments in EPCAM⁻ cells compared with the short and branched actin filaments observed in EPCAM⁺ cells. *Rhoj*-KO EPCAM⁻ cells were associated with short filaments with a dense and multipolar organization (Extended Data Fig. 9b,c). Immunostaining of nuclear actin with phosphorylated histone H3 and EdU showed that chemotherapy promoted the expression of nuclear actin in replicative cells in a RHOJ-dependent manner (Extended Data Fig. 10a–d).

To investigate whether nuclear actin polymerization promotes RHOJ-mediated replication stress tolerance, we assessed the effects of the inhibition of nuclear actin polymerization on the replicative-stress response. A combination of cisplatin/5FU treatment with the F-actin inhibitor latrunculin B significantly and specifically increased apoptosis in EPCAM⁻ cells (Fig. 5d and Extended Data Fig. 10e). Latrunculin B treatment inhibited the formation of new origins of replication and the disappearance of RPA2 foci after chemotherapy in EPCAM⁻ cells (Fig. 5e and Extended Data Fig. 10f), suggesting that RHOJ-mediated replication-stress tolerance is dependent on nuclear actin polymerization. Among different inhibitors of actin remodelling, only the formin inhibitor SMIFH2 increased apoptosis in EPCAM⁻ cells when combined with chemotherapy treatment (Fig. 5d and Extended Data Fig. 10g). Together, these data indicate that RHOJ controls DNA repair by regulating nuclear actin polymerization, which, in turn can modulate the processes of DNA replication and repair.

## Discussion

Here we identified a molecular mechanism by which EMT tumour cells resist a broad range of genotoxic and chemotherapeutic agents in a primary mouse model of skin SCC presenting spontaneous EMT. We found that RHOJ, a small Rho GTPase, is overexpressed in tumour cells presenting EMT compared with in epithelial tumour cells. We found, through genetic gain and loss of function studies in vitro and in vivo, the key role of RHOJ in regulating the resistance of EMT tumour cells to chemotherapy. Whereas small GTPases of the Rho family have been previously associated with DNA repair and response to therapy[42,43], the role of RHOJ in regulating resistance to therapy in EMT tumour cells was unclear.

RHOJ modulated the response to chemotherapy in melanoma cell lines by suppressing ATR and CHK1 activation and decreasing p53-mediated cell death[13]. By contrast, p53 is deleted in skin SCC presenting EMT, and inhibition of ATR does not affect the resistance of EMT tumour cells to chemotherapy, suggesting that RHOJ regulates other mechanisms to mediate resistance to therapy in EMT tumour cells.

RHOJ controls several aspects of tumorigenesis, including tumour initiation and tumour growth, by regulating the rate of tumour cell proliferation. Moreover, RHOJ inhibits cell death after the administration of distinct types of chemotherapy and DNA-damaging agents, and the cells are repaired by different pathways, suggesting that RHOJ does not regulate a specific DNA-repair pathway but, rather, controls a common mechanism shared by the different DDR pathways and mechanisms counteracting replication stress. Our proteomic analysis shows that RHOJ regulates the expression of many proteins that promote DNA replication and DNA repair. After DNA damage and replication stress induced by chemotherapy, both epithelial and EMT tumour cells decrease the speed of DNA replication by activating the intra-S checkpoint. However, only EMT tumour cells activate dormant origins in a RHOJ-dependent manner, which facilitates continued DNA replication despite the slowdown of replication forks and promotes cell survival after chemotherapy.

EMT is associated with remodelling of the actin cytoskeleton, which is essential to promote cell migration and invasion[44]. Although RHOJ was not previously associated with changes in actin remodelling occurring during EMT, it has been implicated in the regulation of actin polymerization, the formation of localized actin stress fibres, and the reduction in actomyosin contractility in fibroblasts and endothelial cells[16,19]. RHOJ also promotes TGF-β-induced breast cancer cell migration and invasion[15] and regulates melanoma cell migration through the regulation of the actin cytoskeleton[45]. RHOJ controls neurite extension by binding to and activating N-WASP to induce ARP2/3-complex-mediated actin polymerization[46]. RHOJ also interacts in a complex containing other actin regulators, including formin like 1 and WASPL/N-WASP[21].

Notably, recent studies have implicated nuclear actin in DNA repair and replication fork repair. Inactivation of ARP2/3, formin or nuclear actin polymerization decrease DNA repair[36,37,39,41,47] and formin has been shown to promote the initiation of DNA replication through the regulation of nuclear actin dynamics[40]. Our data demonstrate that RHOJ promotes resistance to therapy in EMT tumour cells by modulating DNA repair and activating dormant replication origins through the regulation of formin-dependent nuclear actin polymerization.

In conclusion, our study identified the key role of RHOJ and the mechanisms by which RHOJ promotes resistance to chemotherapy in EMT tumour cells, with important implications for the development of new strategies to overcome resistance to anticancer therapy.

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

## Methods

### Mice

*Rosa26-YFP*[48], *Rosa-tDTomato*[49], *K14*[creER], *Lgr5*[creER] (ref. [50]), *KrasLSL*[G12D] (ref. [51]) and *Trp53*[fl/fl] (ref. [52]) mice were obtained from the NCI mouse repository and Jackson Laboratories. *Rhoj*[fl/fl] mice[53] were a gift from A. Uemura. All mice used in this study were composed of males and females with mixed genetic background. Mouse colonies were maintained in a certified animal facility in accordance with the European guidelines. The room temperature ranged from 20 °C to 25 °C. The relative ambient humidity at the level of mouse cages was 55 ± 15%. Each cage was provided with food, water and two types of nesting material. A semi-natural light cycle of 12 h–12 h light–dark was used. All of the experiments were approved by the Ethical Committee for Animal Welfare (Commission d'Ethique et du Bien Etre Animal, CEBEA, Faculty of Medicine, Université Libre de Bruxelles, reference no. 434N and 663N). Sample sizes for experiments involving mice were determined according to protocols 434N and 663N, stating the number of mice used for experiments should be reduced to the minimum as soon as the result is reproducible within each experiment.

### *Kras*[G12D]*Trp53*[fl/fl]-induced skin tumours

Tamoxifen was diluted at 25 mg ml$^{-1}$ in sunflower oil (Sigma-Aldrich). Tamoxifen (2.5 mg) was administered intraperitoneally for 4 days to *Lgr5*[creER]*KrasLSL*[G12D]*Trp53*[fl/fl]*Rosa-YFP*[+/+] mice at postnatal day 28 as previously described[52,54].

### Monitoring of tumour growth

Tumour appearance and size were detected by daily observation and palpation. Mice were euthanized when the tumour size reached more than 1 cm of diameter or when the mice presented signs of distress or lost more than 15% of their initial weight as permitted by our IACUC. These limits were not exceeded in any of the experiments. Skin tumours were measured using precision callipers enabling us to discriminate size modifications of greater than 0.1 mm. Tumour volumes were measured on the first day of appearance of the tumour and then every week until the death of the animal using the formula $V = D \times d \times h \times \pi/6$, where $d$ is the minor tumour axis, $D$ is the major tumour axis and $h$ is the height. No macroscopic or microscopic intestinal tumours were observed in WT (*Lgr5*[creER]*Kras*[G12D]*p53*[cKO]*Rosa-YFP*) or *Rhoj*-KO (*Lgr5*[creER]*Kras*[G12D]*p53*[cKO]*Rhoj*[cKO]*Rosa-YFP*) mice harbouring skin SCC that could have influenced general health and chemotherapy response.

### Primary cell culture

FACS-isolated tumour YFP$^+$EPCAM$^+$ or YFP$^+$EPCAM$^-$ cells were plated on γ-irradiated 3T3 feeder cells in six-well plates. Cells were cultured in MEM medium supplemented with 10% fetal bovine serum (FBS), 0.4 mg ml$^{-1}$ hydrocortisone, $2 \times 10^{-9}$ M T3, 1% penicillin–streptomycin and 2 mM L-glutamine. The feeders were removed using PBS/EDTA (1 mM). Cells were incubated at 37 °C with 20% $O_2$ and 5% $CO_2$.

### Human cell culture

The MDA-MB-231 cell line (ATCC HTB-26) was grown in Dulbecco's modified Eagle's medium supplemented with 10% FBS and 1% penicillin–streptomycin. Cells were incubated at 37 °C with 20% $O_2$ and 5% $CO_2$. No commonly misidentified cell lines were used in this study according to ICLAC register version II. All cell lines have been tested negative for mycoplasma contamination.

### Chemotherapy treatments, irradiation and inhibitors

Mice were treated with 3.5 mg per kg *cis*-diammineplatinum dichloride (cisplatin) (Sigma-Aldrich, P4394) and 15 mg per kg 5FU (Sigma-Aldrich, F6627) administered intraperitoneally weekly for 4 weeks. For in vivo studies of primary mouse models, animals were selected according to their correct genotypes. The mice were induced with tamoxifen

injection 28–35 days after birth and developed tumours in 2–3 months, thus minimizing the difference in age of different animals used. When tumour sizes reached 2–5 mm$^3$, mice were treated with chemotherapy injected intraperitoneally; their tumours were compared with those developing in mice of the same genotype injected with physiological serum.

Mouse tumoural cells were plated on six-well plates. For the irradiation, 10 Gy of radiation was delivered from a $^{137}$Cs source, at a dose rate of 2.34 Gy min$^{-1}$. Cells were collected 24 h after the administration of the dose. For the chemotherapy, cells were incubated with 8.5 μM cisplatin (Sigma-Aldrich, P4394), 15 μM 5FU (Sigma-Aldrich, F6627), 0.4 μM gemcitabine (Sigma-Aldrich, G6423), 200 nM paclitaxel (Sigma-Aldrich, T7402) and 0.25 μM doxorubicin (Sigma-Aldrich, D1515), 1 μM topotecan (Sigma-Aldrich, T2705), 25 μM etoposide (Sigma-Aldrich, E1383). For the inhibitors, cells were incubated with 50 μM aphidicolin (Santa Cruz, sc-201535A), 50 μM Mirin (Selleckchem, S8096), 1 μM VE-821 inhibitor (Selleckchem, S8007), 5 μM KU60019 (Selleckchem, S1570), 400 nM latrunculin B (Sigma-Aldrich, L5288), 5 μM wiskostatin (Sigma-Aldrich, W2270), 50 μM CK666 (Sigma-Aldrich, SML0006) and 50 μM SMIFH2 (Sigma-Aldrich, S4826), as indicated in the figure legends. Investigators were blinded to mouse and cell line genotypes or treatment conditions during experiments and when performing sample analysis, imaging and quantification.

### Scratch assay

Scratch assays were performed to evaluate the effect of *Rhoj* KD on the migration of EPCAM$^-$ tumour cells. The cells were grown in a culture-insert well in 48-well plates. At confluency, cells were serum-starved (1% FBS medium) for 24 h before removing the culture-insert creating a cell-free gap. Next, cells were washed twice with PBS and reincubated in 1% FBS medium. To evaluate wound closure, pictures were processed using ImageJ.

### Cell growth assay (crystal violet assay)

For proliferation testing, cells were first seeded in a 96-well plate. After incubation, cells were washed twice with PBS and fixed with 1% glutaraldehyde at different timepoints as described in the figure legends. The staining was then performed using a 0.2% crystal violet solution, and cell were permeabilized using Triton X-100. The absorbance of each well was read at 570 nm on a microplate reader. The cell growth was measured by the ratio of the absorbance of the well at each timepoint to the average absorbance of the wells at day 0.

### Tumour transplantation assays

To test the long-term response of *Rhoj*-WT and *Rhoj*-KO EPCAM$^-$ tumour cells to chemotherapy, we injected subcutaneously 1,000 cells into NUDE mice after collection in 4 °C medium and resuspension in half medium/half Matrigel (E1270, 970 mg ml$^{-1}$; Sigma-Aldrich). When the mice developed palpable tumours, they were treated weekly with 3.5 mg per kg cisplatin and 15 mg per kg 5FU as described above. The mice were followed by daily observation and weight measurement, and the tumour size was monitored every week.

### Immunoprecipitation

Co-immunoprecipitation of HA-tagged RHOJ was performed using standard methods and adapting previously described protocols[55]. In brief, after three washes in ice-cold Tris-buffered saline, cells were collected on ice by scraping in lysis buffer (150 mM NaCl, 50 mM HEPES pH 7.5, 2 mM EDTA, 10% glycerol, 1 mM β-mercaptoethanol, 1% Triton X-100, protease inhibitor cocktail (11836170001, Roche) and phosphatase inhibitor cocktail 2 (Sigma-Aldrich, P5726)), vortexed 3 times for 30 s with a 2 min pause in between and then centrifuged for 15 min at 13,000 rpm. We evaluated the total protein content of each sample using the Bradford assay and 1 mg of protein was used in every immunoprecipitation. Antibodies (6 μg; rabbit IgG control Chip grade,

ab171870, Abcam; rabbit IPO9, A305-475A, Bethyl Lab) were incubated with 1 mg of precleared lysate at 4 °C under constant rotation overnight. Subsequently, 25 µl Dynabeads Protein G (10003D, Thermo Fisher Scientific) was added and rotated at 4 °C for 4 h. The Dynabeads were washed in 800 µl NETN buffer (20 mM Tris (pH 8), 1 mM EDTA, 900 mM NaCl, 0.5% CA-630) with rotation at 4 °C for 5 min. The washes were repeated five more times and finally the Dynabeads were rinsed with 1 ml dPBS and subsequently eluted with 40 µl 1× SDS gel-loading buffer at 95 °C for 5 min. 26 µl eluted samples were used to perform the western blot using mouse anti-HA antibodies (ab1424, Abcam).

For analyses using liquid chromatography coupled with tandem MS (LC–MS/MS), the lysates were not precleared and the Dynabeads were washed once in wash buffer (150 mM NaCl, 50 mM HEPES pH 7.5, 2 mM EDTA, 10% glycerol, 1 mM β-mercaptoethanol, 0.1% Triton X-100, protease inhibitor cocktail and phosphatase inhibitor cocktail) followed by three washes in MS-compatible buffer (20 mM Tris-HCl pH 8.0, 2 mM CaCl$_2$) before being frozen.

## Histology and immunostaining

For immunostaining of frozen sections, skin tumours were embedded in optimal cutting temperature compound (OCT, Sakura) and cut into 5 µm frozen sections using the CM3050S Leica Cryostat (Leica Microsystems) after being pre-fixed in 4% paraformaldehyde for 4 h at room temperature, rinsed in PBS and incubated overnight in 30% sucrose at 4 °C. The sections were blocked using blocking buffer for 1 h (PBS, 5% horse serum, 1% BSA, Triton 0.1%) and then incubated with primary antibodies diluted in blocking buffer overnight at 4 °C, washed three times with PBS for 5 min, and then incubated with Hoechst solution and secondary antibodies diluted in blocking buffer for 1 h at room temperature. Finally, the sections were washed three times with PBS 5 min at room temperature and mounted in DAKO mounting medium supplemented with 2.5% Dabco (Sigma-Aldrich). Haematoxillin and eosin stainings were performed on paraffin sections, 5 µm paraffin sections were deparaffinized and rehydrated. Slides were mounted using SafeMount (Labonord). For immunostaining of cultured cells, the cells were plated on a coverslip and fixed with 4% paraformaldehyde for 5 min 48 h after plating the cells. For DDR nuclear focus detection, the coverslips were fixed with 4% paraformaldehyde for 5 min and, after washing, were incubated either in 70% ethanol for 20 min at −20 °C for γ-H2AX and 53bp1 staining or in methanol for 20 min at −20 °C for RAD51 and PCNA staining.

Non-specific antibody binding was prevented by blocking with 5% horse serum, 1% BSA and 0.2% Triton X-100 for 1 h at room temperature. When mouse primary antibodies were used, non-specific antigen blocking was performed using the M.O.M. Basic kit reagent (Vector Laboratories) according to the manufacturer's instructions. Coverslips were then incubated overnight at 4 °C in the presence of the primary antibodies, followed by 1 h of incubation of the secondary antibodies and Hoechst solution at room temperature. Coverslips were mounted using Glycergel (Dako) supplemented with 2.5% DABCO (Sigma-Aldrich). For F-Actin immunofluorescence, the following modifications from the above protocol were made: the coverslips were incubated with blocking solution for 30 min, then incubated for 30 min with rhodamine Phalloidin and, after washes, were incubated with Hoechst solution for 20 min.

## Image acquisition and data analysis for immunofluorescence microscopy

Imaging of tumour tissue was performed on the Zeiss Axio Imager M1 (Thornwood) fluorescence microscope with the Zeiss Axiocam MRm camera for immunofluorescence and Zeiss Axiocam MRC5 camera for bright-field microscopy using Axiovision release 4.6 software. Quantification of Ki-67-positive tumour cells was performed on four different fields per tumour using ImageJ. Cell pictures were acquired using the Zeiss Axio Imager.M1 supplied by a ×100 objective (alpha plan-fluar 1.4 numerical aperture oil-immersion objective). z series were acquired at 0.3 µm intervals throughout the entire nucleus. Orthogonal projection was performed using Zen Blue 3.3 (ZEISS) to quantify the number of DNA-damage-induced repair foci. The number of foci in each cell was quantified using ImageJ. To determine the nuclear localization of RHOJ, z stacks were processed using the Huygens Professional 22.04 deconvolution software and the global intersection coefficient was used to characterize the degree of overlap between HA-tagged RHOJ and DAPI signals. Brightness, contrast and picture size were adjusted using Photoshop CS6 (Adobe).

## Western blot analysis

Total cell lysates were prepared using the radioimmunoprecipitation assay supplemented with phosphatase inhibitors (Cell Signaling, 5870) for 15 min on ice and then centrifuged for 15 min at 14,000 rpm. The protein concentrations of supernatants were measured using the Bradford assay. Total protein lysate (30 µg) was loaded on a gel. Gel electrophoresis was performed using 4–12% NuPAGE Bis-Tris gradient gels (Invitrogen), transferred to PVDF membranes. Membranes were blocked for 1 h in Tris-buffered saline, 0.1% Tween-20 (TBST) containing 3% bovine serum albumin (BSA) then incubated with primary antibodies in the blocking buffer overnight at 4 °C. After washing in TBST, membranes were incubated with secondary antibodies in the blocking buffer for 1 h. Proteins were detected by enhanced chemiluminescence (ECL) western blotting detection reagents (Amersham Biosciences).

## FACS isolation of EPCAM$^+$ and EPCAM$^-$ tumour cells

Tumours were dissected, minced and digested in collagenase I (Sigma-Aldrich) during 2 h at 37 °C on a rocking plate. Collagenase I activity was blocked by the addition of EDTA (5 mM) and the cells were then rinsed in PBS supplemented with 2% FBS. Before the staining, cells were blocked for 20 min at room temperature in PBS supplemented with 30% FBS. Cell suspensions were filtered through a 70 µm cell strainer (BD) then through a 40 µm cell strainer to ensure the elimination of cell debris and clumps of cells. Immunostaining was performed using PE-conjugated anti-CD45 (30F11, 1:100, eBioscience), PE-conjugated anti-CD31 (MEC13.3; 1:100, BD PharMingen) and APC-Cy7-conjugated anti-EPCAM (G8.8; 1:100, BioLegend) for 30 min at 4 °C on a rocking plate protected from light. Living tumour cells were selected by forward scatter, side scatter, doublets discrimination and by Hoechst dye exclusion. EPCAM$^+$ and EPCAM$^-$ tumour cells were selected on the basis of the expression of YFP and the exclusion of CD45, CD31 (Lin$^-$). For EPCAM$^-$ tumour cell subpopulation identification, brilliant violet stain buffer (BD Bioscience) was added (50 µl per sample) and the cells were incubated with PE-conjugated anti-CD51 (rat, RMV-7, BioLegend 104106, 1:50), BV421-conjugated anti-CD61 (Armenian hamster, 2C9.G2, BD Bioscience 553345, 1:50), biotin-conjugated anti-CD106 (rat, 429 (MVCAM.A), BD Bioscience, 553331, 1:50), BV711-conjugated anti-EPCAM (rat, G8.8, BD Bioscience, 563134, 1:100), PerCPCy5.5 conjugated anti-CD45 (rat, 30-F11, BD Bioscience, 550994, 1:100) and PerCPCy5.5 conjugated anti-CD31 (rat, MEC13.3, BD Bioscience, 562861, 1:100) for 30 min at 4 °C. Cells were washed with PBS supplemented with 2% FBS and incubated with streptavidin-BV786 (BD Bioscience, 563858, 1:400) for 30 min at 4 °C. FACS analysis was performed using FACSAria and FACSDiva software (BD Bioscience). Sorted cells were collected either in culture medium for cell culture experiments or into lysis buffer for RNA extraction.

## FACS analysis

After trypsinization and washing once in cold PBS, cells were incubated with APC-Cy7-conjugated anti-EPCAM (G8.8; 1:100, BioLegend) antibodies diluted in 200 µl PBS supplemented with 2% FBS for 30 min at 4 °C on a rocking plate protected from light before the fixation step required for the intracellular antigens staining. To detect cell apoptosis or double-stranded DNA breaks, cells were respectively labelled with PE anti-active caspase-3 (BD PharMingen, 550821, 1:25) and PE

anti-H2AX (pS139) (BD PharMingen, 562377, 1:20) using the active caspase-3 apoptosis kit (BD PharMingen, 550480) according to the manufacturer's protocol and resuspended in PBS supplemented with 2% FBS. For cell cycle distribution analysis, cells were incubated with 10 μM 5-bromo-2′-deoxyuridine (BrdU) for 45 min. Cells were labelled for BrdU incorporation with an Alexa Fluor 647 anti-BrdU (BD PharMingen, 560209, 1:50) antibody using the BrdU flow kit (BD Pharmigen, 55789) according to the manufacturer's protocol and resuspended in 20 μl 7-AAD for 5 min followed by 200 μl PBS supplemented with 2% FBS. For the chemotherapy sensitivity assay, cells were seeded at an equal density in a six-well plate. Then, 24 h after seeding, the cells were treated with chemotherapy as indicated in the figure legends. Next, 24 h and 48 h after the start of treatment, cells were collected by trypsinization and quantified by counting the number of living cells by FACS. Living cells were selected by forward and side scatter and by Hoechst dye exclusion. FACS analysis was performed using Fortessa, FACSDiva software and FlowJo (BD Bioscience).

## Antibodies

For immunostaining, the following primary antibodies were used: goat GFP (Abcam, ab6673, 1:500), chicken K14 (Thermo Fisher Scientific, MA5-11599, 1:2,000), rabbit vimentin (Abcam, ab92547, 1:500), rabbit active caspase-3 (R&D, AF835, 1:600), rabbit 53bp1 (Novus, NB100-304, 1:200), mouse phospho-histone H2A.X (Ser139) (Millipore, 05-636, 1:500), mouse RAD51 (Santa-Cruz Biotechnology, sc-398587, 1:50), rat RPA32/RPA2 (Cell Signaling, 2208, 1:500), Rhodamine Phalloidin (Thermo Fisher Scientific, Arg415, 1:400), rabbit HA (Abcam, ab9110, 1:1,000), rat phospho-histone H3 (S28) (Abcam, ab10543, 1:2,000), rabbit Ki-67 (Abcam, ab15580, 1:400).

For western blotting, the following primary antibodies were used: rabbit phospho-histone H2A.X (Ser139) (Cell Signaling, 2577, 1:800), rabbit histone H2A.X (Cell Signaling, 2595, 1:1,000), rabbit phosphorylated ATM/ATR substrate (Cell Signaling, 9607, 1:750), rabbit beta-actin (Abcam, ab8227, 1:2,000), mouse HA (Roche, 11583816001, 1:1,000), rabbit phosphorylated CDC2 (Tyr15) (Cell Signaling, 4539, 1:1,000), rabbit phosphorylated CDC2 (Thr161) (Cell Signaling, 9114, 1:1,000), rabbit CDC2 (Cell Signaling, 77055, 1:1,000), rabbit phosphorylated CDK2 (Thr160) (Cell Signaling, 2561, 1:1,000), rabbit CDK2 (Cell Signaling, 18048, 1:1,000), affinity-purified mouse monoclonal antibodies NB8-AD9 (WB/IP) raised against human phosphorylated CDK4 (Thr172)[56] (1:500), rabbit CDK4 (Abcam, ab199728, 1:1,000), rabbit CDK6 (Cell Signaling, 3136, 1:1,000), rabbit N-WASP (Cell Signaling, 4848, 1:1,000), rabbit MCM2 and rabbit MCM3[61], mouse POLD (Santa Cruz, sc-373731), mouse PCNA (Santa Cruz, sc-56), rat RPA32 (Cell Signalling, 2208), rabbit phosphorylated RPA32 S4/S8 (Bethyl, A300-245A), rabbit CTCF (Millipore, 07-729), rabbit H3 (Abcam, ab1791) and mouse α-tubulin (Sigma-Aldrich, T9026). The following secondary antibodies were used: ECL anti-rabbit, anti-rat anti-mouse IgG conjugated with horseradish peroxidase (GE Healthcare, 1:2,000 or 1:5,000).

## Nuclear actin Chromobody transfection

The Nuclear Actin ChromobodyTagGFP plasmid (pnACTagGFP) was purchased from Chromotek. Cells were plated on sterile cover slips and transfected 24 h later with Lipofectamine 3000 (Thermo Fischer Scientific). The chemotherapy agents (cisplatin/5FU) were added 36 h after transfection for 12 and 24 h. After PBS washes, samples were fixed with 4% paraformaldehyde for 5 min at room temperature, counterstained with Hoescht and then mounted with Glycergel (Dako) supplemented with 2.5% DABCO (Sigma-Aldrich). For co-immunostaining of EdU with p-H3 and nuclear actin chromobody–GFP, cells grown onto round coverslips were treated as described in the figures and incubated with 10 μM EdU for the last 45 min. The coverslips were then washed three times with PBS and fixed with 4% paraformaldehyde for 5 min at room temperature. Permeabilization steps and detection of EdU were performed using click-iT Plus EdU cell proliferation kit, Alexa

Fluor 647 dye (Thermo Fischer Scientific, C10640), according to the manufacturer's protocol. The coverslips were then stained with p-H3 and Hoescht before proceeding to mounting as previously described.

## Virus production, infection and selection

Stable KD cell lines were generated using lentiviral pLKO/PuroR vectors (Sigma-Aldrich) after puromycin selection. KD was confirmed by RT–qPCR. Three different shRNA were used at the same time to target the same gene. Overexpression cell lines were generated using pLX302-EF1a lentiviral vector. pLX302-EF1a was a gift from the Beronja laboratory (Fred Huchinson Cancer Research Center). N-terminal 3×HA-tagged *Rhoj* construct was cloned into pLX302-EF1a using Gateway Technology (Invitrogen) according to the manufacturer's instructions. For virus production, $5 \times 10^6$ HEK293T cells were seeded into 10 cm dishes and transfected with the vector of interest and appropriate packaging plasmids psPax2 and pMD2.G (12260 and 12259, respectively, Addgene). The medium was changed 24 h later and supernatants were next collected at 48 h, and passed through a 0.45 μm filter. Tumour epithelial cells or tumour mesenchymal cells were plated in six-well plates and incubated with 40 μl per ml viruses when they reach 50% of confluence, in the presence of polybrene (5 mg ml$^{-1}$). Medium was changed 24 h later and cells were selected with puromycin for at least 1 week.

## shRNA

The following shRNA were used (TRC ID, clone name): mRhoj: TRCN0000313500, NM_023275.2, 673s21c1; mRhoj: TRCN0000313499, NM_023275.2, 713s21c1; mRhoj: TRCN0000077567, NM_023275.1, 690s1c1; mRhoq: TRCN0000077513, NM_145491.2, 1732s1c1; mRhoq: TRCN0000077515, NM_145491.2, 892s1c1; mRhoq: TRCN0000077516, NM_145491.2, 661s1c1; hRHOJ: TRCN0000047603, NM_020663.2, 878s1c1; hRHOJ: TRCN0000047604, NM_020663.2, 1007s1c1; hRHOJ: TRCN0000047605, NM_020663.2, 647s1c1.

## RNA and DNA extraction and RT–qPCR

RNA extraction from FACS-isolated cells was performed using the RNeasy micro kit (QIAGEN) according to the manufacturer's recommendations with DNase treatment. After RNA quantification using the Nanodrop, the first-strand cDNA was synthesized using Superscript II (Invitrogen) and random hexamers (Roche) at a final volume of 50 μl. Control of genomic contaminations was measured for each sample by performing the same procedure with or without reverse transcriptase. qPCR assays were performed using 1 ng of cDNA as template, SYBRGreen mix (Applied Bioscience) and the Light Cycler 96 (Roche) real-time PCR system. The *Tbp* housekeeping gene was used for normalization of mouse RT–qPCR and the *POLR2A* housekeeping gene was used for normalization of human RT–qPCR. Primers were designed using Roche Universal ProbeLibrary Assay Design Center (https://lifescience.roche.com/webapp/wcs/stores/servlet/Category Display?tab=Assay+Design+Center&identifier=Universal+Probe+Lib rary&langId=-1). qPCR analysis was performed using the Light Cycler 96 (Roche) and the $\Delta\Delta C_t$ method with *Tbp* as a reference.

## Primers used for RT–qPCR

The sequences of the primers used for RT–qPCR were as follows: mRhoj For(5′-3′): ACCACTACGCAGTTACCGTG; mRhoj Rev(3′-5′): TGCAACACCATTCTCCGACC; mRhoq For(5′-3′): TTCGACCACT ACGCAGTCAG; mRhoq Rev(3′-5′): CCTGCAAACCGCGTATAAGG; mTBP For(5′-3′): TGTACCGCAGCTTCAAAATATTGTAT; mTBP Rev(3′-5′): AAATCAACGCAGTTGTCCGTG; mKrt14 For(5′-3′): GCCGCCCCTGG TGTGGAC; mKrt14 Rev(3′-5′): GTGCGCCGGAGCTCAGAAATC; mEpcam For(5′-3′): CATTTGCTCCAAACTGGCGT; mEpcam Rev(3′-5′): TTGTTCTGGATCGCCCCTTC; mVimFor(5′-3′): CCAACCTTTTCTTCCC TGAAC; mVimRev(3′-5′): TTGAGTGGGTGTCAACCAGA; mZeb1For(5′-3′): ATTCCCCAAGTGGCATACA; mZeb1 Rev(3′-5′): GAGCTAGTGTCT TGTCTTTCATCC; mZeb2 For(5′-3′): ACCGCATATGGCCTATACCTAC;

mZeb2 Rev(3′-5′): TGCTCCATCCAGCAAGTCT; mPrrx1 For(5′-3′): TGTTGATTCGAGCGGGAAGA; mPrrx1 Rev(3′-5′): TCTAGCAGGTGAC TGACGGA; mPdgfra For(5′-3′): TGGAAGCTTGGGGCTTACTT; mPdgfra Rev(3′-5′): CATAGCTCCTGAGACCTTCTCC; hRHOJ For(5′-3′): ACAA TGTCCAGGAGGAATGGG; hRHOJ Rev(3′-5′): TGTGCTCCGATCGCTTTTG; hRHOQ For (5′-3′): AAAGAGGAGTGGGTACCGGA; hRHOQ Rev (5′-3′): GCAGCATGCTCCTATCTCTT; hPOLR2A For(5′-3′): GCAAATTCACCA AGAGAGACG; and hPOLR2A Rev(3′-5′): CACGTCGACAGGAACATCAG.

## RNA-seq analysis

RNA quality was checked using the Bioanalyzer (Agilent). For RNA extracted from tumour epithelial cells or tumour mesenchymal cells, indexed cDNA libraries were obtained using the TruSeq Stranded mRNA Sample prep kit (Illumina) according to the manufacturer's recommendations. The multiplexed libraries (11 pM) were loaded and sequences were produced using a HiSeq PE Cluster Kit v4 and TruSeq SBS Kit v3-HS (250 cycles) on the HiSeq 1500 (Illumina) system. Approximately 8 million paired-end reads per sample were mapped against the mouse reference genome (GRCm38.p4/mm10) using STAR software to generate read alignments for each sample. Annotations Mus_mus-culus.GRCm38.84.gtf were obtained from https://ftp.ensembl.org/. After transcripts were assembled, gene-level counts were obtained using HTSeq and normalized to 20 million aligned reads. Fold change (FC) values were computed on these values between the conditions. The accession number for the RNA-seq reported in this paper is GEO: GSE205985.

## DNA fibre analysis

Exponentially growing cells were pulse-labelled with 50 µM CldU (20 min; Sigma-Aldrich, C6891) followed by 250 µM IdU (20 min; Sigma-Aldrich, I7125). Labelled cells were collected and DNA fibres were spread in buffer containing 0.5% SDS, 200 mM Tris pH 7.4 and 50 mM EDTA as described previously[57]. For immunodetection of labelled tracks, fibres were incubated with primary antibodies (for CldU, rat anti-BrdU, ab6326 Abcam; for IdU, mouse anti-BrdU, 347580 BD Bioscience) for 1 h at room temperature and developed with the corresponding secondary antibodies (anti-rat IgG AF594, A-11007; anti-mouse IgG1 AF488, A-21121; all from Molecular Probes) for 30 min at room temperature. Mouse anti-ssDNA antibody was used to assess fibre integrity (MAB3034, Millipore, secondary antibody anti-mouse IgG2a AF647, A-21241 Molecular Probes). Slides were examined with a Leica DM6000 B microscope, as described previously[58]. The conversion factor used was 1 µm = 2.59 kb (ref. [59]). In each assay, more than 250 tracks were measured to estimate the fork rate and more than 500 tracks were analysed to estimate the frequency of origin firing (first label origins—green-red-green—are shown as the percentage of all red (CldU-labelled) tracks)[60]. For estimating asymmetry, both forks emanating from an origin were measured (100 origins per condition in each experiment) and the ratio of long/short was calculated. An origin was considered to be asymmetric when the long/short fork ratio was greater than 1.4.

## Chromatin fractionation

For the analysis of chromatin-bound proteins, biochemical fractionations were performed as described previously[61]. In brief, cells were resuspended at $10^7$ cells per ml in buffer A (10 mM HEPES pH 7.9, 10 mM KCl, 1.5 mM MgCl$_2$, 0.34 M sucrose, 10% glycerol, 1 mM DTT, 1× protease inhibitor cocktail; bimake.com), and incubated on ice for 5 min in the presence of 0.1% Triton X-100. Low-speed centrifugation (for 4 min at 600$g$ and 4 °C) enabled the separation of the cytosolic fraction (supernatant) and nuclei (pellet). Nuclei were washed and subjected to hypotonic lysis in buffer B (3 mM EDTA, 0.2 mM EGTA, 1 mM DTT, 1× protease inhibitor cocktail) for 30 min on ice. The nucleoplasmic and chromatin fractions were separated after centrifugation (for 4 min at 600$g$ and 4 °C). Chromatin was resuspended in Laemmli sample buffer. Whole-cell extracts (WCE) were prepared by resuspension of cells in Laemli sample buffer ($10^7$ cells per ml). Both WCE and chromatin were sonicated twice for 15 s at 15% amplitude and boiled at 95 °C for 5 min. Then, 10 µl of WCE (equivalent to $10^5$ cells) and 20 µl of chromatin ($2 × 10^5$ cells) were loaded onto 4–20% custom-made gradient gels. Protein was transferred to nitrocellulose membranes, blocked with 5% non-fat milk in TBST for 1 h at room temperature and incubated with primary antibodies overnight at 4 °C. Incubation with secondary antibodies was performed for 1 h at room temperature and proteins were detected with WesternBright ECL (Advansa).

## EdU immunofluorescence

Cells grown onto round coverslips were treated as described in the figure and incubated with 25 µM EdU for the last 30 min. The coverslips were then washed twice with PBS and cells were pre-extracted with CSK buffer (10 mM PIPES pH 7, 0.1 M NaCl, 0.3 M sucrose, 3 mM MgCl$_2$) for 5 min 4 °C. After fixation with formaldehyde for 10 min at room temperature, a click reaction was performed on the coverslips (100 mM Tris pH 8, 10 mM CuSO$_4$, 2 mM Na-L-ascorbate, 50 µM biotin-azide-AF488) for 1.5 h at 37 °C. Then, the coverslips were washed with PBS and DNA was stained with DAPI (1 mg ml$^{-1}$ in PBS) for 10 min at room temperature, mounted with Prolong Gold Antifade (Thermo Fisher Scientific) and visualized under a fluorescence microscope (DM6000 B Leica microscope with a HCX PL APO 40 0.75 NA objective). EdU intensity was assessed in at least 250 EdU-positive nuclei per condition per experiment using CellProfiler (v.3.1.9).

## Sample preparation for proteomic analysis

To define the effect of RHOJ on protein expression in EMT tumour cells, a total of 24 samples was prepared for LC−MS/MS analyses, corresponding to 3 replicates of EPCAM$^+$ tumour cells, 3 replicates of EPCAM$^-$ tumour cells, 3 replicates of EPCAM$^-$ $Rhoj$ shRNA tumour cells and 3 replicates of EPCAM$^-$ control shRNA treated for 24 h with cisplatin/5FU as described above or with vehicle control (DMSO). Cells were collected by manual scraping in Tris-buffered saline on ice and were flash-frozen as a dry pellet. The cell pellets (15 million cells per pellet) were lysed in 1 ml urea lysis buffer containing 9 M urea, 20 mM HEPES pH 8.0 and PhosSTOP phosphatase inhibitor cocktail (Roche, 1 tablet per 10 ml buffer). The samples were sonicated with 3 pulses of 15 s at an amplitude of 20% using a 3 mm probe, with incubation on ice for 1 min between pulses. After centrifugation for 15 min at 20,000$g$ at room temperature to remove insoluble components, proteins were reduced by addition of 5 mM DTT and incubation for 30 min at 55° C and then alkylated by addition of 10 mM iodoacetamide and incubation for 15 min at room temperature in the dark. The protein concentration was measured using the Bradford assay (Bio-Rad) and, from each sample, 1 mg protein was used to continue the protocol. The samples were further diluted with 20 mM HEPES pH 8.0 to a final urea concentration of 4 M and proteins were digested with 10 µg LysC (Wako) (1/100, w/w) for 4 h at 37 °C. Samples were again diluted to 2 M urea and digested with 10 µg trypsin (Promega) (1/100, w/w) overnight at 37° C. The resulting peptide mixture was acidified by addition of 1% trifluoroacetic acid and, after 15 min incubation on ice, the samples were centrifuged for 15 min at 1,780$g$ at room temperature to remove insoluble components. Next, peptides were purified on SampliQ SPE C18 cartridges (Agilent). Columns were first washed with 1 ml 100% acetonitrile and pre-equilibrated with 3 ml of solvent A (0.1% trifluoroacetic acid in water/acetonitrile (98:2, v/v)) before the samples were loaded on the column. After peptide binding, the column was washed again with 2 ml of solvent A and peptides were eluted twice with 750 µl elution buffer (0.1% trifluoroacetic acid in water/acetonitrile (20:80, v/v)). Then, 75 µl of the eluate was dried completely in a SpeedVac vacuum concentrator for shotgun analysis.

## LC−MS/MS analysis

Purified peptides were redissolved in 30 µl solvent A and half of each sample was injected for LC−MS/MS analysis on the Ultimate 3000

RSLCnano system in-line connected to a Q Exactive HF mass spectrometer equipped with a Nanospray Flex Ion source (Thermo Fisher Scientific). Trapping was performed at 10 µl min⁻¹ for 4 min in loading solvent A on a 20 mm trapping column (custom made, 100 µm internal diameter, 5 µm beads, C18 Reprosil-HD, Dr Maisch) and the sample was loaded onto a reverse-phase column (custom made, 75 µm inner diameter × 400 mm, 1.9 µm beads C18 Reprosil-HD, Dr Maisch). The peptides were eluted by a nonlinear increase from 2% to 56% solvent B (0.1% formic acid in water/acetonitrile (20:80, v/v)) over 145 min at a constant flow rate of 250 nl min⁻¹, followed by a 10 min wash reaching 99% solvent B and re-equilibration with solvent A (0.1% formic acid in water). The column temperature was kept constant at 50 °C (CoControl v.3.3.05, Sonation).

The mass spectrometer was operated in data-dependent mode, automatically switching between MS and MS/MS acquisition for the 16 most abundant ion peaks per MS spectrum. Full-scan MS spectra (375 to 1,500 $m/z$) were acquired at a resolution of 60,000 in the Orbitrap analyzer after accumulation to a target value of 3,000,000. The 16 most intense ions above a threshold value of 13,000 were isolated (window of 1.5 Th) for fragmentation at a normalized collision energy of 28% after filling the trap at a target value of 100,000 for maximum 80 ms. MS/MS spectra (200 to 2,000 $m/z$) were acquired at a resolution of 15,000 in the orbitrap analyzer. The S-lens RF level was set at 55, and we excluded precursor ions with single and unassigned charge states from fragmentation selection. QCloud[62] was used to control instrument longitudinal performance during the project.

## Proteomic data analysis

Data analysis of the shotgun data was performed using MaxQuant (v.1.5.8.3) using the Andromeda search engine with the default search settings including a false-discovery rate set at 1% at the peptide and protein level. Spectra were searched against the mouse proteins in the Swiss-Prot database (database release version of September 2017 containing 16,931 mouse protein sequences; http://www.uniprot.org). The mass tolerance for precursor and fragment ions was set to 4.5 and 20 ppm, respectively, during the main search. Enzyme specificity was set as C terminal to arginine and lysine, also allowing cleavage at proline bonds with a maximum of two missed cleavages. Variable modifications were set to oxidation of methionine residues, acetylation of protein N termini, and phosphorylation of serine, threonine or tyrosine residues, while carbamidomethylation of cysteine residues was set as a fixed modification. Matching between runs was enabled with a matching time window of 0.7 min and an alignment time window of 20 min. Only proteins with at least one unique or razor peptide were retained leading to the identification of 5,354 proteins. Proteins were quantified by the MaxLFQ algorithm integrated in the MaxQuant software. A minimum ratio count of two unique or razor peptides was required for quantification.

Further data analysis of the shotgun results was performed using the Perseus software (v.1.5.5.3) after loading the protein groups file from MaxQuant. Reverse database hits, potential contaminants and hits only identified by site were removed, LFQ intensities were log₂-transformed and replicate samples were grouped. Proteins with less than three valid values in at least one group were removed and missing values were imputed from a normal distribution around the detection limit leading to a list of 4,239 quantified proteins that was used for further data analysis. To reveal proteins of which the expression level was significantly regulated between the different conditions, sample groups were defined on the basis of the treatment (control versus treated) and RHOJ expression (with versus without RHOJ) and a two-way ANOVA test was performed. For each protein, this test calculated a $P$ value (−log-transformed $P$ value) for treatment, a $P$ value for RHOJ expression and a $P$ value for the interaction between treatment and RHOJ expression. Proteins with $P < 0.05$ in at least one of these three conditions were considered to be significantly regulated. A total of 99

proteins of interest were selected, the log₂ transformed intensities of these proteins were Z-scored and these values were plotted in a heat map after non-supervised hierarchical clustering.

The MS proteomics data have been deposited at the ProteomeXchange Consortium through the PRIDE[63] partner repository under dataset identifier PXD025737.

## Sample preparation for AP–MS

To identify the proteins interacting with RHOJ, a total of 12 samples were prepared for LC–MS/MS analysis corresponding to 3 replicates of HA-tagged-RHOJ-transfected EPCAM⁻ tumour cells and 3 control empty vector EPCAM⁻ tumour cells treated for 12 h with cisplatin/5FU or with vehicle control (DMSO). Immunoprecipitation of HA-tagged RHOJ was performed as follows. After three washes in ice-cold Tris-buffered saline, cells were collected on ice by scraping in lysis buffer (150 mM NaCl, 50 mM HEPES pH 7.5, 2 mM EDTA, 10% glycerol, 1 mM β-mercaptoethanol, 1% Triton X-100, protease inhibitor cocktail (11836170001, Roche) and phosphatase inhibitor cocktail 2 (Sigma-Aldrich, P5726)), vortexed three times for 30 s with a 2 min pause in between and then centrifuged for 15 min at 13,000 rpm. The total protein content of each sample was evaluated using the Bradford assay and 1 mg of protein was used in every immunoprecipitation. A total of 6 µg of antibodies (rabbit HA tag Chip grade, ab9110, Abcam) were incubated with 1 mg of lysate at 4 °C under constant rotation overnight. Subsequently, 25 µl Dynabeads Protein G (10003D, Thermo Fisher Scientific) was added and rotated at 4 °C for 4 h. The Dynabeads were washed once in wash buffer (150 mM NaCl, 50 mM HEPES pH 7.5, 2 mM EDTA, 10% glycerol, 1 mM β-mercaptoethanol, 0.1% Triton X-100, protease inhibitor cocktail and phosphatase inhibitor cocktail) followed by three washes in MS-compatible buffer (20 mM Tris-HCl pH 8.0, 2 mM CaCl₂). Washed beads were resuspended in 150 µl trypsin digestion buffer and incubated for 4 h with 1 µg trypsin (Promega) at 37° C. Beads were removed, another 1 µg of trypsin was added and proteins were further digested overnight at 37° C. Peptides were purified on Omix C18 tips (Agilent) and dried completely in a rotary evaporator.

## LC–MS/MS analysis

Peptides were redissolved in 20 µl loading solvent A (0.1% trifluoroacetic acid in water/acetonitrile (98:2, v/v)) of which 2 µl was injected for LC–MS/MS analysis on the Ultimate 3000 RSLCnano system in-line connected to a Q Exactive HF mass spectrometer (Thermo Fisher Scientific). Trapping was performed at 10 µl min⁻¹ for 2 min in loading solvent A on a 5 mm trapping column (Thermo Fisher Scientific, 300 µm internal diameter, 5 µm beads). The peptides were separated on a 250 mm Waters nanoEase M/Z HSS T3 Column, 100 Å, 1.8 µm, 75 µm inner diameter (Waters Corporation) kept at a constant temperature of 45 °C. Peptides were eluted by a non-linear gradient starting at 1% MS solvent B reaching 33% MS solvent B (0.1% formic acid in water/acetonitrile (2:8, v/v)) in 60 min, 55% MS solvent B (0.1% formic acid in water/acetonitrile (2:8, v/v)) in 75 min, 99% MS solvent B in 90 min followed by a 10 min wash at 99% MS solvent B and re-equilibration with MS solvent A (0.1% formic acid in water). The mass spectrometer was operated in data-dependent mode, automatically switching between MS and MS/MS acquisition for the 12 most abundant ion peaks per MS spectrum. Full-scan MS spectra (375–1,500 $m/z$) were acquired at a resolution of 60,000 in the Orbitrap analyzer after accumulation to a target value of 3,000,000. The 12 most intense ions above a threshold value of 15,000 were isolated with a width of 1.5 $m/z$ for fragmentation at a normalized collision energy of 30% after filling the trap at a target value of 100,000 for maximum 80 ms. MS/MS spectra (200–2,000 $m/z$) were acquired at a resolution of 15,000 in the Orbitrap analyzer.

## Data analysis

Analysis of the MS data was performed in MaxQuant (v.2.0.3.0) with mainly the default search settings, including a false-discovery rate set

at 1% at the peptide-to-spectrum match, peptide and protein level. Spectra were searched against the mouse proteins in the Reference proteins database (UP000000589, database release version of January 2022 containing 21,986 mouse protein sequences; http://www.uniprot.org). The mass tolerance for precursor and fragment ions was set to 4.5 and 20 ppm, respectively, during the main search. Enzyme specificity was set as C terminal to arginine and lysine, also allowing cleavage at proline bonds with a maximum of two missed cleavages. Variable modifications were set to oxidation of methionine residues, acetylation of protein N termini. Matching between runs was enabled with a matching time window of 0.7 min and an alignment time window of 20 min. Only proteins with at least one unique or razor peptide were retained. Proteins were quantified using the MaxLFQ algorithm integrated in the MaxQuant software. A minimum ratio count of two unique or razor peptides was required for quantification. A total of 126,976 peptide-to-spectrum matches was performed, resulting in 14,501 identified unique peptides, corresponding to 2,091 identified proteins. Further data analysis of the AP−MS results was performed using a custom R script, using the proteinGroups output table from MaxQuant. Reverse database hits were removed, LFQ intensities were $\log_2$-transformed and the replicate samples were grouped. Proteins with less than three valid values in at least one group were removed and missing values were imputed from a normal distribution centred around the detection limit (package DEP[64]), leading to a list of 1,381 quantified proteins in the experiment, used for further data analysis. To compare protein abundance between pairs of sample groups (RhoJHAuntreatedIP versus EVuntreatedIP, RhoJHAtreated12hchemoIP versus EVtreated12hchemoIP sample groups), statistical testing for differences between two group means was performed, using the package limma[65]. Statistical significance for differential regulation was set to a false-discovery rate of <0.05 and a fold change of >2-fold or <0.5-fold ($|\log_2FC| = 1$). The MS proteomics data have been deposited at the ProteomeXchange Consortium through the PRIDE partner repository under dataset identifier PXD038278.

## Statistics

Statistical and graphical data analyses were performed using IBM-SPSS v.28.0 (IBM, Released 2021; IBM SPSS Statistics for Windows, v.28.0), Medcalc v.20 (MedCalc Statistical Software v.20.109; MedCalc Software; https://www.medcalc.org; 2022) and Prism 5 (GraphPad). Continuous variables are summarized by their means and their s.e.m., and qualitative variables as numbers and percentages. Differences in continuous variables were compared between groups using nonparametric tests in the case of small sample sizes ($n$ of less than around 30), including Mann−Whitney tests in the case of two groups and Kruskal−Wallis tests followed by Mann−Whitney tests corrected for multiple comparisons with Bonferroni correction when there were more than two groups. When sample sizes were sufficiently large ($n$ of more than around 30), parametric tests were used, including classical Student's $t$-tests or Welch's $t$-tests in the case of variance inequality when there were two samples and ANOVA followed by Sidak test or Dunnett T3 test in the case of variance heterogeneity when more than two samples had to be compared. $P < 0.05$ was considered to be statistically significant. All statistical tests were two-sided. Each test used is mentioned in the legend of the respective figure.

## Reporting summary

Further information on research design is available in the Nature Portfolio Reporting Summary linked to this article.

## Data availability

The MS proteomics data have been deposited to the ProteomeXchange Consortium through the PRIDE[63] partner repository under dataset identifiers PXD025737 and PXD038278. RNA-seq data reported in this paper are available at the GEO (GSE205985). All other data are available from the corresponding author on reasonable request. Source data are provided with this paper.

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

**Acknowledgements** We thank G. Lapouge and the staff at the ULB animal facility; the staff at the ULB genomic core facility (F. Libert and A. Lefort); M. Martens and J.-M. Vanderwinden and the members of the LiMiF (www.limif.ulb.ac.be) for help with microscopy; B. Beck, O. Serçin, A. Karambelas and Blanpain laboratory members for discussions and suggestions. We thank Y. Pommier for helpful insights. M.D. was supported by FNRS doctoral fellowship (Aspirant) and Association Vésale. M.Z. and M.-A.P. were supported by FNRS fellowship and the TELEVIE. J.M. is supported by the Ministry of Science and Innovation, Spain. The Center for Microscopy and Molecular Imaging (CMMI) is supported by the European Regional Development Fund and the Walloon Region. Work in the Mendez laboratory was supported by grant PID2019-106707-RB from the Spanish Ministry of Science and Innovation, co-sponsored by ERDF funds from the EU. C.B. is supported by WELBIO, FNRS, the TELEVIE, Julie and Françoise Drion Fondation, Fondation Contre le Cancer, ULB Foundation, Foundation Baillet Latour, FNRS/FWO EOS (40007513) and the European Research Council (AdvGrant 885093).

**Author contributions** M.D. and C.B. designed the experiments, performed most of the data analysis and wrote the manuscript. M.D., M.-A.P. and J.B. performed most of the biological experiments. M.Z. contributed to performing gain-of-function experiments and sample collection for proteomic analysis. M.L. contributed to performing shRNA gene KD experiments. S.R.-A. and J.M. conducted DNA fibre analysis, identification of replication factors on the chromatin fraction, EdU incorporation analysis and western blot validation of proteins identified by proteomic analysis, and acquired, analysed and interpreted the data. K.C. performed the western blot to study the activation of different CDKs. Y.S. performed bioinformatic analysis. V.d.M. performed the statistical analysis. C.D. performed FACS. V.M. provided technical support. F.I., D.V.H. and S.D. performed proteomic analysis. A.U. generated *Rhoj*-cKO mice. P.A.S. contributed to the research design and supervised experiments. All of the authors read and approved the final manuscript.

**Competing interests** The authors declare no competing interests.

**Additional information**
**Correspondence and requests for materials** should be addressed to Cédric Blanpain.

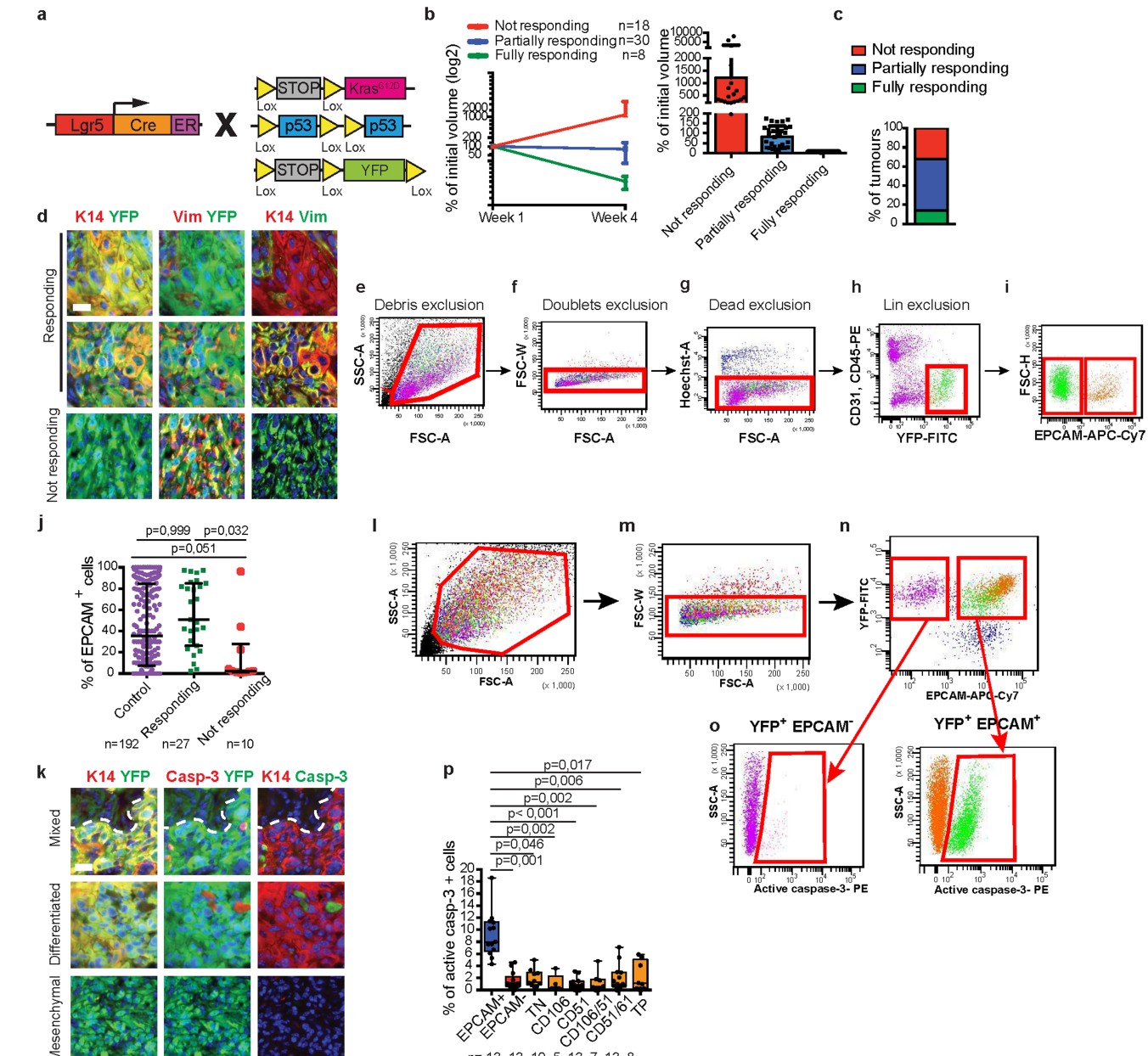

**Extended Data Fig. 1 | Mouse skin SCCs with EMT are resistant to chemotherapy. a**, Scheme of the genetic strategy to induce *Kras*<sup>G12D</sup> and YFP expression and *Trp53* deletion in the mouse model of skin SCCs. After tumour appearance, mice are treated for 4 weeks with cisplatin/5FU. **b**, Evolution of tumour size of treated primary SCC. Relative tumour volumes (Mean ± S.D.) are shown for not responding (n = 18), partially responding (n = 30) and fully responding (n = 8) tumours after 4 weeks of cisplatin/5FU treatment. **c**, Proportion of not responding, partially responding and fully responding tumours after 4 weeks of cisplatin/5FU treatment. **d**, Coimmunostaining of YFP and Keratin 14 (K14) or Vimentin (Vim) in partially responding and non-responding tumours treated for 4 weeks with cisplatin/5FU treatment (n = 5; scale bar, 20µm). **e-i**, FACS plots showing the gating strategy used to FACS-isolate or to analyse the proportion of YFP+ EPCAM⁺ and EPCAM⁻ tumour cells from *Lgr5*<sup>creER</sup>*Kras*<sup>G12D</sup>*Trp53*<sup>cKO</sup>*Rosa-YFP* and *Lgr5*<sup>creER</sup>*Kras*<sup>G12D</sup>*Trp53*<sup>cKO</sup>*Rhoj*<sup>cKO</sup>*Rosa-YFP* skin SCCs presented on Fig. 2h and Extended Data Fig. 1j. Cells were stained for endothelial and immune markers (Lin+) (CD31, CD45) in PE, EPCAM in APC-Cy7 and were gated to eliminate debris (e) and to discard the doublets (f). The living cells were gated by Hoescht dye exclusion (g) and the non-epithelial Lin+ cells

were discarded (h). EPCAM expression was studied in Lin-YFP+ (i). **j**, FACS quantification of the percentage of EPCAM⁺ cells in the different SCCs subtypes. (Medians and interquartile range (IQR) are shown, two-sided Mann-Whitney *U*-test, n represents the number of SCCs). **k**, Co-immunostaining of YFP and activated Caspase-3 or K14 in differentiated, mixed and mesenchymal tumours after 24h of cisplatin/5FU treatment (n = 10; scale bar, 20µm). Nuclei are stained with DAPI (blue). **l-o**, FACS plots showing the gating strategy to quantify the percentage of activated caspase-3 positive cells in YFP+ EPCAM⁺ and EPCAM⁻ cells after eliminating debris and discarding the doublets (Fig. 1a, d; 2c, e, i, j; 5d and Extended Data Fig. 2d, 7g, h, 8d, e, 10e, g). The same strategy (**l-n**) was used to quantify the percentage of cells expressing γH2AX and cell cycle analysis (Fig. 4c–e) in YFP+EPCAM⁺, EPCAM⁻ and *Rhoj* KO EPCAM⁻ cells. **p**, FACS quantification of the percentage of activated caspase-3 positive cells in YFP+/EPCAM⁺ tumour cells, YFP+/EPCAM⁻ tumour cells and in the six different subpopulations of EPCAM⁻ tumour cells based on CD106/51/61 markers 24h after cisplatin/5FU administration to mice harbouring SCCs (Medians (IQR) are shown, Kruskal-Wallis test followed by two-sided Mann Whitney tests, n represents the number of SCCs).

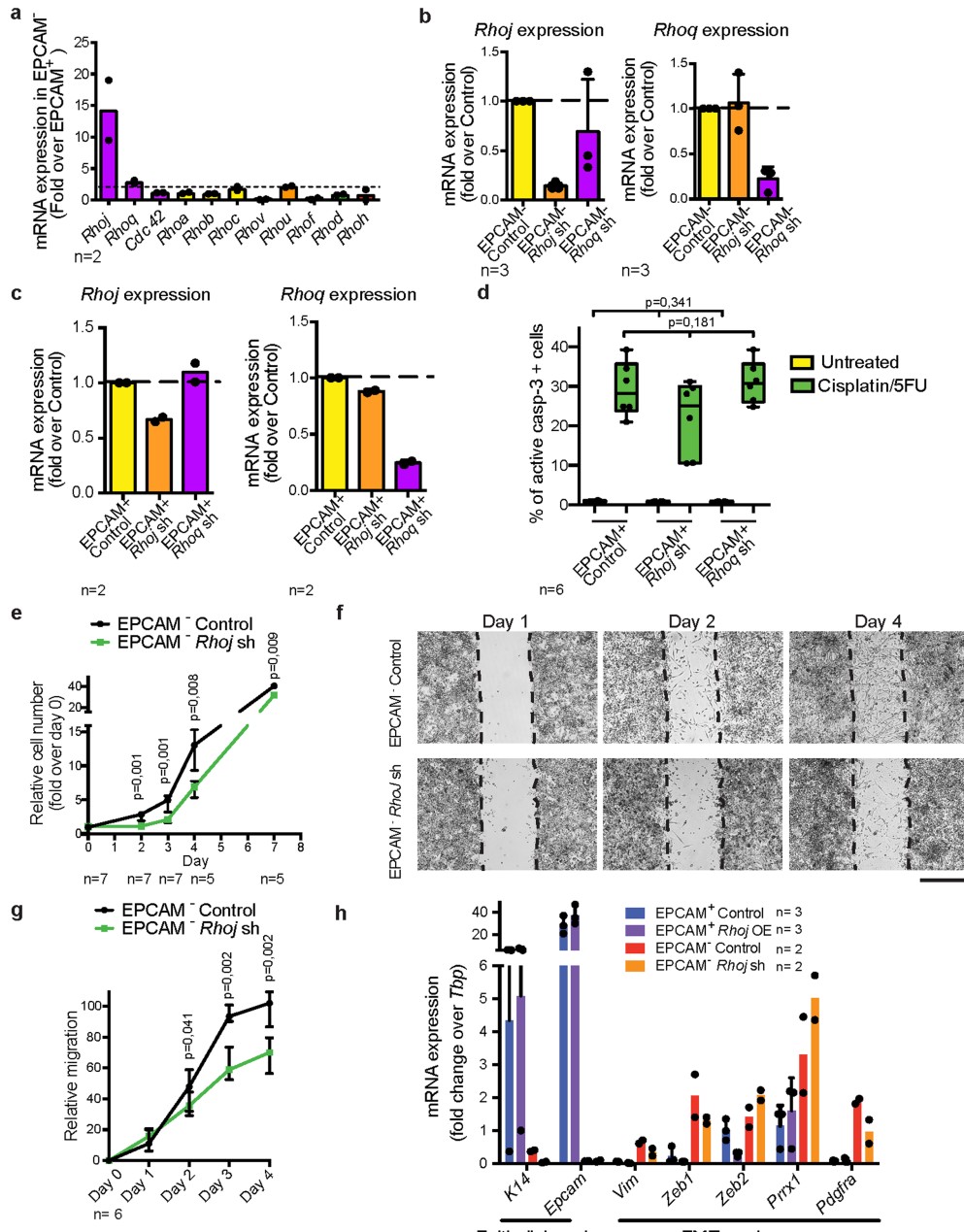

**Extended Data Fig. 2 | RHOJ knockdown in EMT tumour cells decreases cell survival, growth and migration without affecting the expression of EMT or epithelial markers. a**, Relative mRNA expression of *Rhoj* and *Rhoq* by RNA seq in FACS isolated EPCAM+ and- EPCAM− cells (Bar chart indicates mean; n = 2 SCCs). **b-c**, qRT-PCR of *Rhoj* and *Rhoq* expression validating the specific downregulation of *Rhoj* and *Rhoq* after shRNA knockdown in EPCAM− cells as compared to control shRNA EPCAM− cells (b) and after shRNA knockdown in EPCAM+ cells as compared to control shRNA EPCAM+ cells (c) (data are normalized using housekeeping gene *Tbp*, Bar chart indicates mean± SD, n represents the number of independent primary cell lines). **d**, FACS quantification of the percentage of activated caspase-3 positive cells in control EPCAM+, *Rhoj* shRNA knockdown EPCAM+ and *Rhoq* shRNA knockdown EPCAM+ cells 24h after cisplatin/5FU administration (box boundaries represent 25th and 75th percentiles; whiskers, minimum and maximum; centre line, median, Kruskal-Wallis test followed by two-sided Mann Whitney

tests, n represents the numbers of replicates from 2 independent cell lines). **e**, Relative cell number as compared to day 0 in control EPCAM− and *Rhoj* shRNA knockdown cells cultured in vitro and quantified by crystal violet assay (Medians and range are shown, two-sided Mann-Whitney *U*-test, n represents replicates of three biological replicates). **f-g**, Cell Migration. Representative images (f) and quantification (g) of cell migration following scratch-wound assay in EPCAM− control and *Rhoj* shRNA knockdown cells (scale bar, 500µm; data represent the percentage of wound closure, medians with range are shown, two-sided Mann-Whitney *U*-test, n = 6 biological replicates, 3–4 pictures per well). **h**, mRNA expression of selected epithelial and EMT markers quantified by qRT-PCR in EPCAM− control cells, *Rhoj* shRNA knockdown cells, EPCAM+ control cells and *Rhoj* overexpressing cells (data are normalized using housekeeping gene TBP. Bars represent mean± SD, n represents the number of independent primary cell lines).

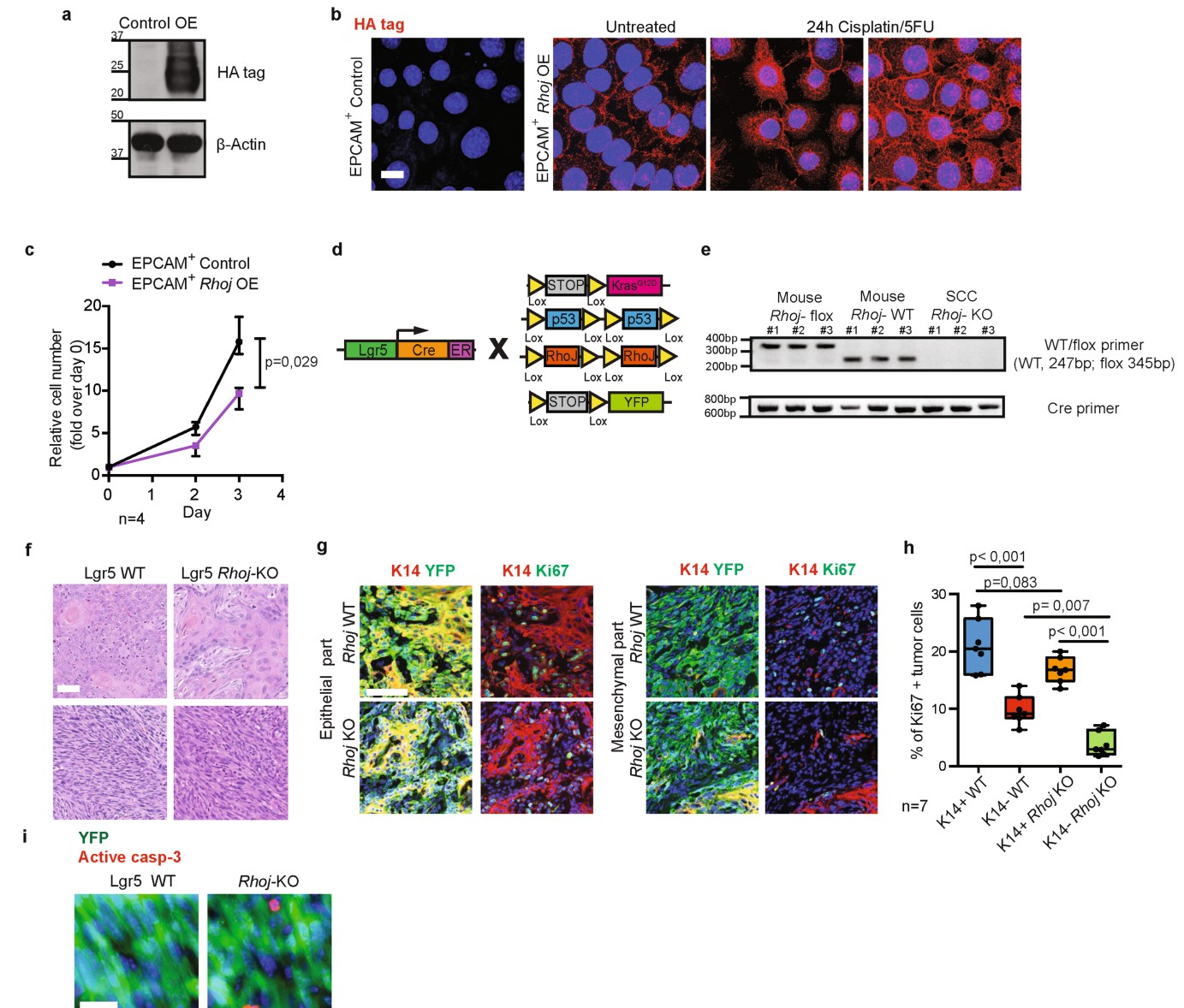

**Extended Data Fig. 3 | RHOJ overexpression in EPCAM⁺ tumour cells in vitro and RHOJ deletion in tumour cells in vivo impair cell proliferation.**
**a-b**, HA-tagged RHOJ expression measured by western blotting (a) and immunofluorescence (b) validating the overexpression (OE) of RHOJ in EPCAM⁺ cells as compared to control empty vector transduced EPCAM⁺ cells. Molecular weights (kDa) are indicated on the left side of the blots (n = 3 biological replicates; scale bar, 20μm). **c**, Relative cell number of YFP+ EPCAM⁺ control and RHOJ OE cells quantified by crystal violet assay (Medians and range are shown, two-sided Mann-Whitney *U*-test, n represents duplicates of two biological replicates). **d**, Scheme of the genetic strategy to induce *Kras^{G12D}* and YFP expression, *TrpS3* and *Rhoj* deletion in the mouse model of skin SCCs. **e**, PCR analysis of *Cre* expression and *Rhoj* floxed allele recombination in *Lgr5^{creER}Kras^{G12D}TrpS3^{cKO}Rosa-YFP* mice (n = 3), *Lgr5^{creER}Kras^{G12D}TrpS3^{cKO}Rhoj^{cKO}*

*Rosa-YFP* (*Rhoj* KO) mice (n = 3) and *Rhoj* KO tumours (n = 3) confirming the deletion of the floxed alleles in *Rhoj*-KO SCCs arising after tamoxifen administration. **f**, Hematoxylin and Eosin staining of WT and *Rhoj*-KO SCCs. (Scale bar, 50μm; n = 15 SCCs). **g**, Co-immunostaining of YFP (tumour cells), K14 and Ki67 (proliferating cells) in epithelial and mesenchymal part of *Rhoj* WT and *Rhoj*-KO SCCs (Scale bar, 100 μm). **h**, Quantification of the percentage of proliferating cells in epithelial (YFP+K14+) and mesenchymal (YFP+K14-) tumour cells from *Rhoj*-WT and *Rhoj*-KO SCCs (n = 7 independent tumour samples, Kruskal-Wallis test followed by two-sided Mann-Whitney *U*-tests, p-values adjusted by Bonferroni correction; box boundaries represent 25th and 75th percentiles; whiskers, minimum and maximum; centre line, median). **i**, Co-Immunostaining of active caspase-3 and YFP in primary tumours from WT and *Rhoj*-KO SCCs treated with cisplatin/5FU for 24h (n = 5; scale bar, 20μm).

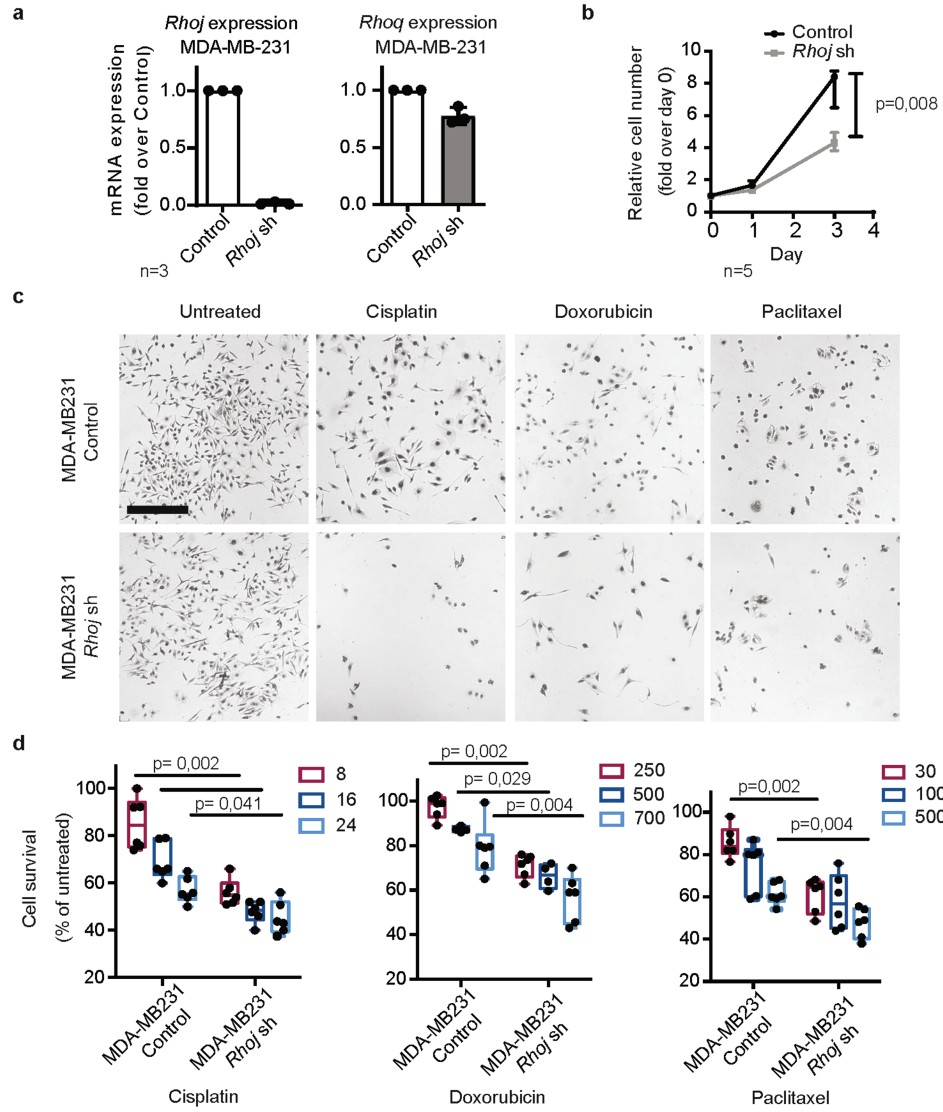

**Extended Data Fig. 4 | RHOJ mediates resistance to chemotherapy associated with EMT in human cancer cells. a**, qRT-PCR of *Rhoj* and *Rhoq* expression validating the specific downregulation of *Rhoj* after shRNA knockdown (*Rhoj* sh) in human breast cancer cell line MDA-MB-231 as compared to control cells. (n = 3 independent experiments, data are normalized using housekeeping gene *Polr2a*; Control shRNA (Control) is an empty shRNA vector). **b**, Relative cell number as compared to day 0 in MDA-MB-231 control and RhoJ shRNA knockdown cultured cells in vitro and quantified by performing crystal violet assay (Medians and range are shown, two-sided Mann-Whitney *U*-test, n represents triplicates from two biological replicates). **c-d**, Representative images (c) and cell survival analysis (d) after 48h of treatment with 8, 16, 24 µM cisplatin; 30, 100, 300 nM paclitaxel; 250, 500, 700 nM doxorubicin of MDA-MB-231 control cells and *Rhoj* shRNA knockdown (scale bar, 500 µm; box boundaries represent 25th and 75th percentiles; whiskers, minimum and maximum; centre line, median, two-sided Mann-Whitney U-test, n represents replicates from two biological replicates).

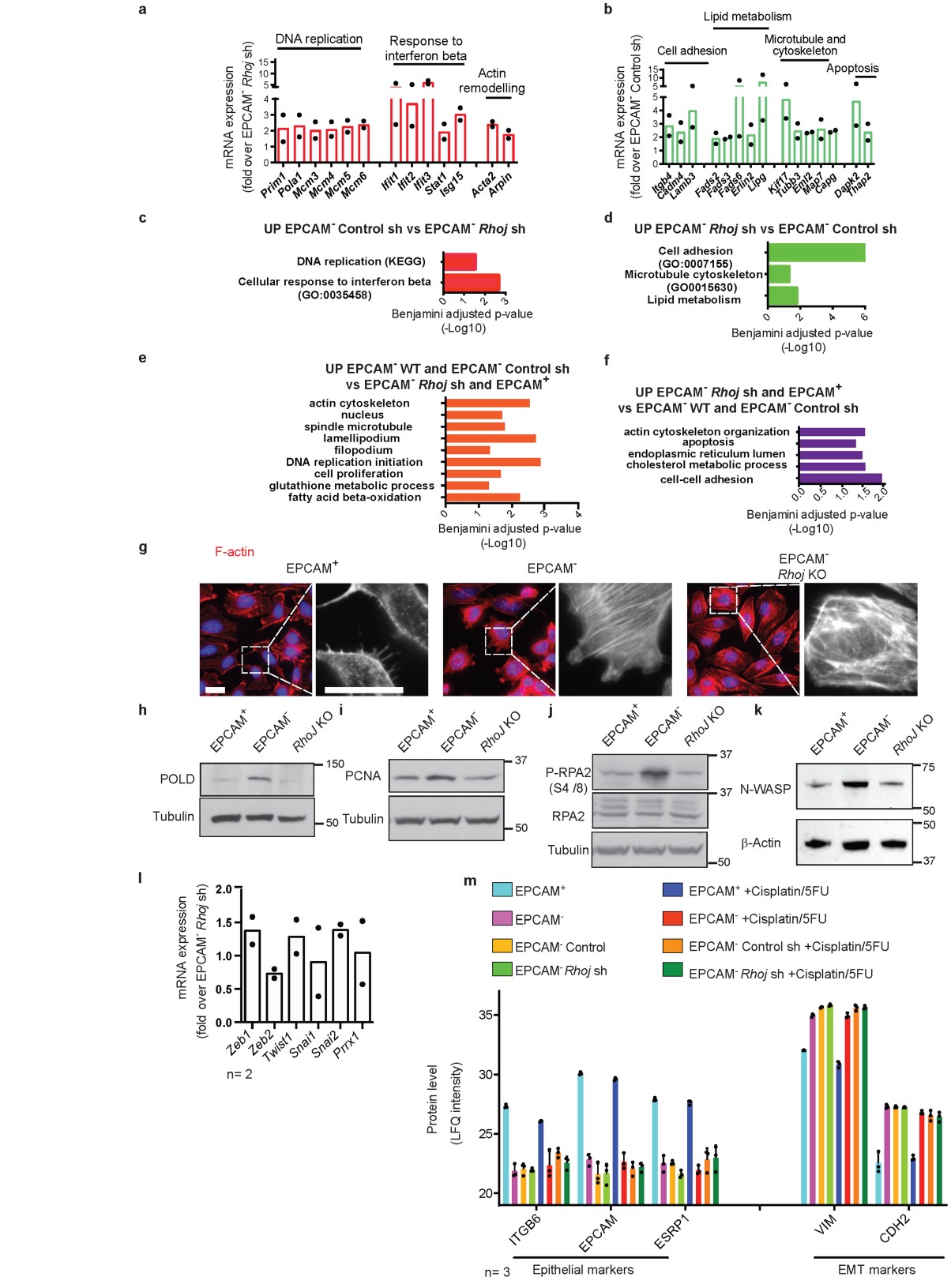

**Extended Data Fig. 5 |** See next page for caption.

**Extended Data Fig. 5 | Transcriptomic and proteomic characterization of EMT tumour cells after *Rhoj* deletion. a**, **b**, mRNA expression of genes upregulated (a) or downregulated (b) in EPCAM⁻ Control shRNA compared to EPCAM⁻ *Rhoj* sh KD measured by RNA seq. (Means are shown, n = 2 independent primary cultured cell lines). **c**,**d**, Gene Ontology analysis of genes that are upregulated (c) or downregulated (d) in EPCAM⁻ control cells compared to *Rhoj* Sh KD EPCAM⁻ cells (c), showing categories of genes that are significantly enriched. **e-f**, Gene Ontology analysis corresponding to the proteins significantly upregulated in EPCAM⁻ WT and EPCAM⁻ Control sh (e) and in EPCAM⁺ and EPCAM⁻ *Rhoj* sh (f). p value is calculated according to the Benjamini–Hochberg method for multiple hypothesis testing. **g**, Immunofluorescence of Phalloidin (red) in EPCAM⁺, EPCAM⁻ and EPCAM⁻ *Rhoj* KO cells. Nuclei are counterstained with DAPI (blue) (n = 3 biological replicates; scale bar, 20µm). **h-k**, Western blot showing the expression of POLD (h), PCNA (i), phospho-RPA2 (S4/8), total RPA2 (j) and N-WASP (k) in EPCAM⁺, EPCAM⁻ and *Rhoj* KO cells. Tubulin or β-Actin loading controls (n = 2, molecular weights (kDa) are indicated to the right side of the blots). **l**, mRNA expression of EMT transcription factors measured by RNA-sequencing in EPCAM⁻ control cells compared to EPCAM⁻ *Rhoj* sh KD cells (Means are shown, n represents the number of independent primary cell lines). **m**, Protein expression of selected epithelial and EMT markers in EPCAM⁺, EPCAM⁻ WT, EPCAM⁻ Control sh, EPCAM⁻ *Rhoj* sh cells untreated and treated for 24h with cisplatin/5FU (n represents the number of technical replicates, Means + S.D. are shown).

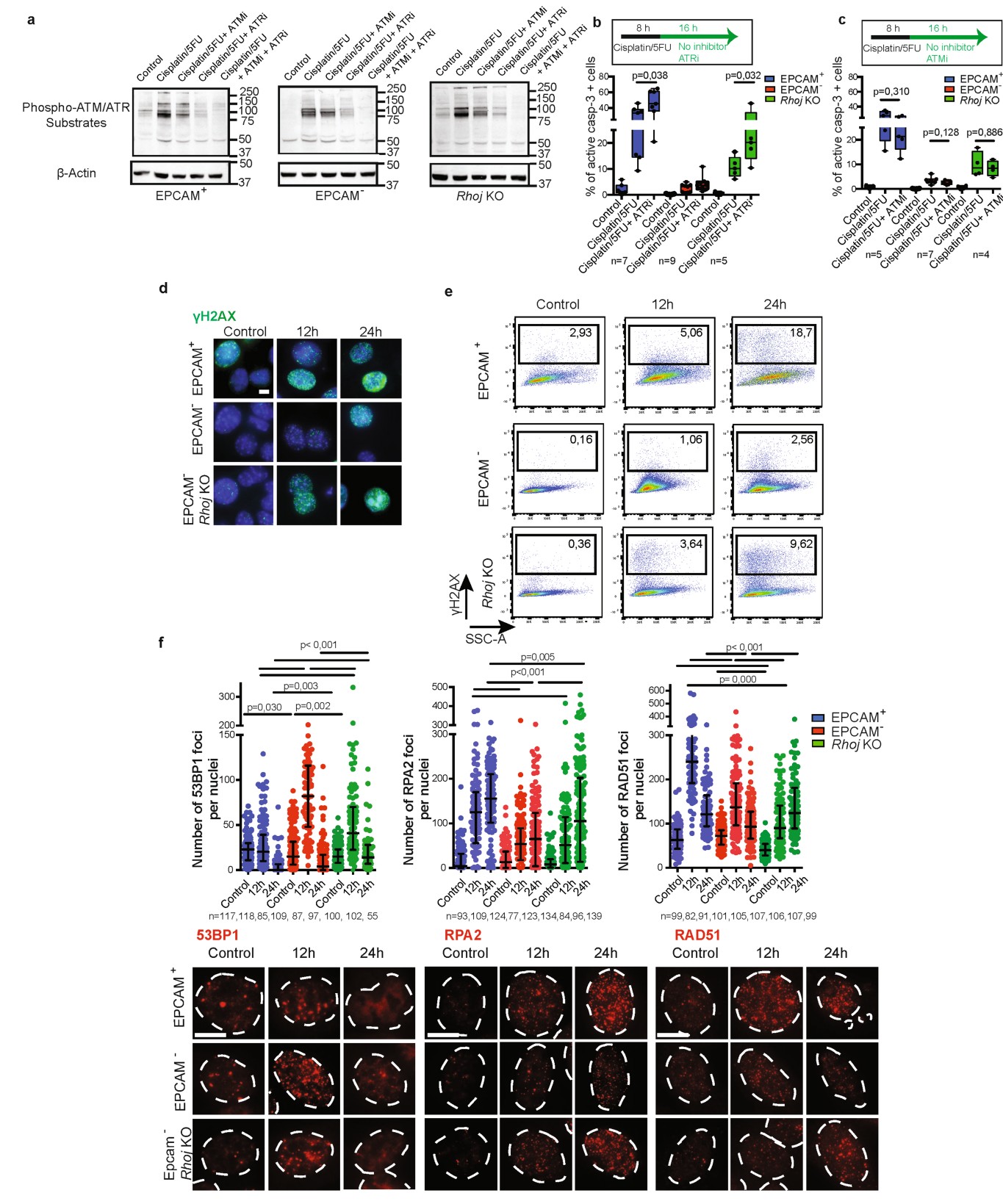

**Extended Data Fig. 6** | See next page for caption.

**Extended Data Fig. 6 | RHOJ promotes DNA repair in EMT tumour cells independently of ATM and ATR. a**, Western blot analysis of phospho-ATM/ATR substrates (S*Q) in EPCAM$^+$, EPCAM$^-$ and EPCAM$^-$ *Rhoj* KO cells untreated, treated with cisplatin/5FU for 8 h and treated with cisplatin/5FU for 8 h followed by 16 h of ATM inhibitor (KU60019, 5µM) or ATR inhibitor (VE-821, 1µM) or a combination of ATM and ATR inhibitor showing the specific decrease in the level of phosphorylated ATM/ATR substrates upon combination of their respective inhibitors with chemotherapy (n = 2 independent experiments, β-Actin loading control, molecular weights (kDa) are indicated to the right side of the blots). **b**, **c**, FACS quantification of the percentage of activated caspase-3 positive cells treated with cisplatin/5FU for 8h followed by treatment with ATR inhibitor (b) or ATM inhibitor (c) for 16h. Top panel shows drug treatment scheme. (two-sided Mann-Whitney *U*-test, n represents the number of replicates using 4 different primary cultured cell lines, box boundaries represent 25th and 75th percentiles; whiskers, minimum and maximum; centre line, median). **d**, **e**, Representative immunofluorescence (d), and FACS plots (e) of the cells expressing γH2AX in EPCAM$^+$, EPCAM$^-$ and EPCAM$^-$ *Rhoj* KO cells, 12h and 24h after cisplatin/5FU administration. (n = 3 independent primary cultured cell lines; scale bar, 10µm). **f**, Representative immunofluorescence and quantification of 53BP1, RPA2 and RAD51 focus formation (in red) in EPCAM$^+$, EPCAM$^-$ and EPCAM$^-$ *Rhoj* KO cells treated with cisplatin for 12h and 24h. (Scale bar, 10µm. Medians (IQR) are shown, Kruskal-Wallis test, p-values adjusted for multiple comparisons by Bonferroni correction, n represents the number of analysed nuclei from two independent primary cultured cell lines and experiments).

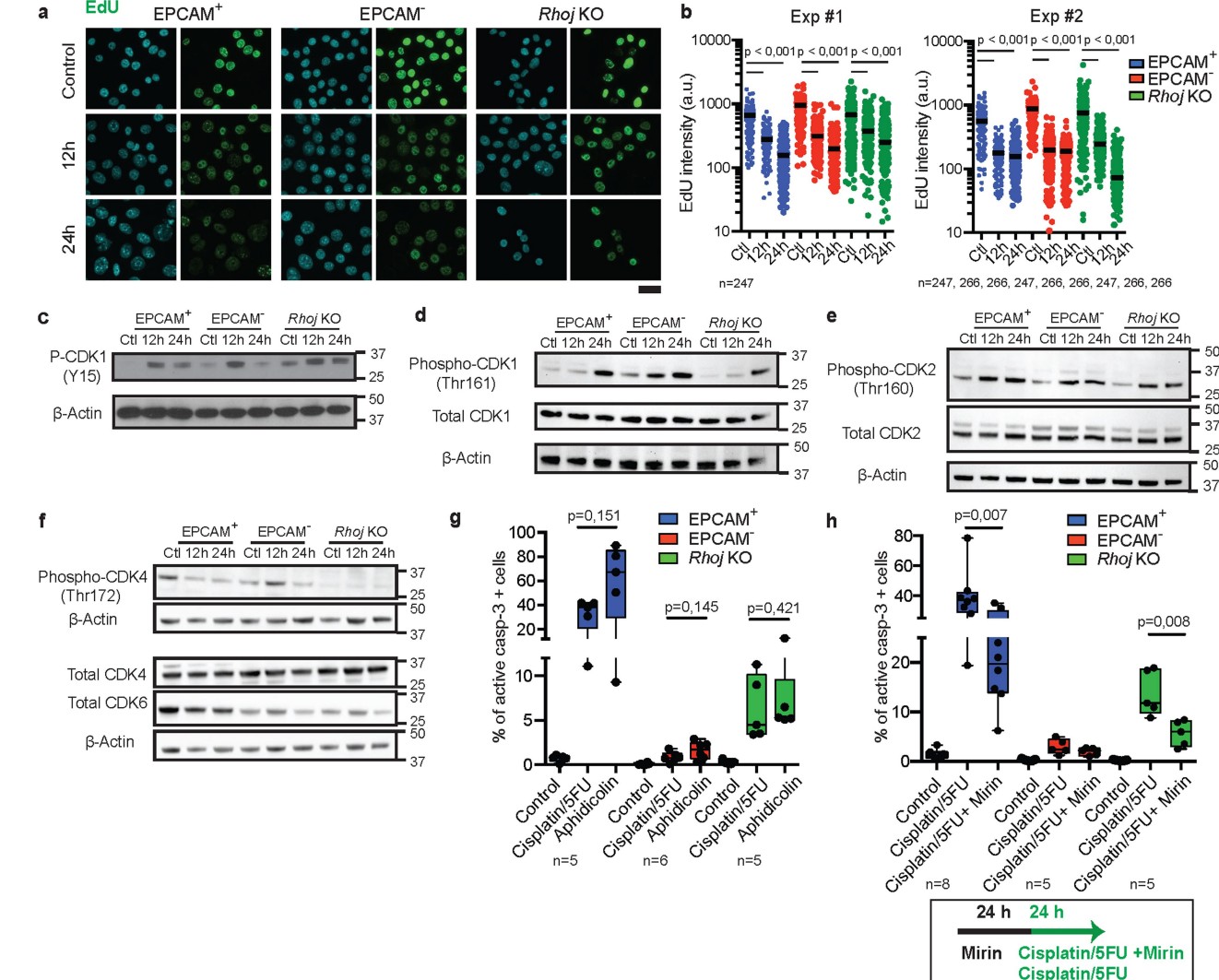

**Extended Data Fig. 7 | RHOJ promotes tolerance to replicative stress in EMT tumour cells following chemotherapy. a**, Immunofluorescence of EdU incorporation (green) in EPCAM⁺, EPCAM⁻ and EPCAM⁻ *Rhoj* KO cells treated with cisplatin/5FU for 12 and 24h. EdU pulse is given 30 min before sampling. Nuclei are counterstained with DAPI (blue) (Scale bar, 25µm). **b**, EdU intensity measured with Cell profiler software in EPCAM⁺, EPCAM⁻ and EPCAM⁻ *Rhoj* KO cells treated with cisplatin/5FU for 12 and 24h. (n represents the number of nuclei analysed from 2 independent experiments ; a.u., arbitrary unit, Medians are shown, Kruskal-Wallis test) **c-f**, Western blot of p-CDK1 at Y15 (c), at Thr161 and total CDK1 (d), p-CDK2 at Thr160 and total CDK2 (e), p-CDK4 at Thr172, total CDK4 and CDK6 (f) and corresponding β-Actin in EPCAM⁺, EPCAM⁻ and *Rhoj* KO cells exposed to cisplatin/5FU treatment showing the differential activation of the CDKs in response to chemotherapy in the different tumour cell types. Activation of CDK1 was observed in EPCAM⁻ cells as shown by transient level of inhibitory phosphorylation and earlier appearance of activating phosphorylation of CDK1 in response to chemotherapy as compared to EPCAM⁺ and *Rhoj* KO Epcam- cells. Same levels of activating CDK2 phosphorylation was found in all cell types and sustained activating CDK4 phosphorylation was observed in EPCAM⁻ cells 12h after chemotherapy compared to EPCAM⁺ and *Rhoj* KO EPCAM⁻ cells suggesting that EPCAM⁻ cells are allowed to progress in the cell cycle following chemotherapy. Molecular weights (kDa) are indicated on the right side of the blots (n = 2). **g**, FACS quantification of the percentage of activated caspase-3 positive cells in YFP+ EPCAM⁺, EPCAM⁻ and EPCAM⁻ *Rhoj* KO in response to 24h aphidicolin (50µM) and cisplatin/5FU. **h**, FACS quantification of the percentage of activated caspase-3 positive cells in response to 24h combination of cisplatin/5FU with MRE11 inhibitor (Mirin 50µM). Lower panel shows drug treatment scheme. (n represents the number of independent primary cultured cell lines). In **g** and **h**, Kruskal-Wallis test followed by Mann-Whitney *U*-tests. p-values adjusted for multiple comparisons by Bonferroni correction, box boundaries represent 25th and 75th percentiles; whiskers, minimum and maximum; centre line, median.

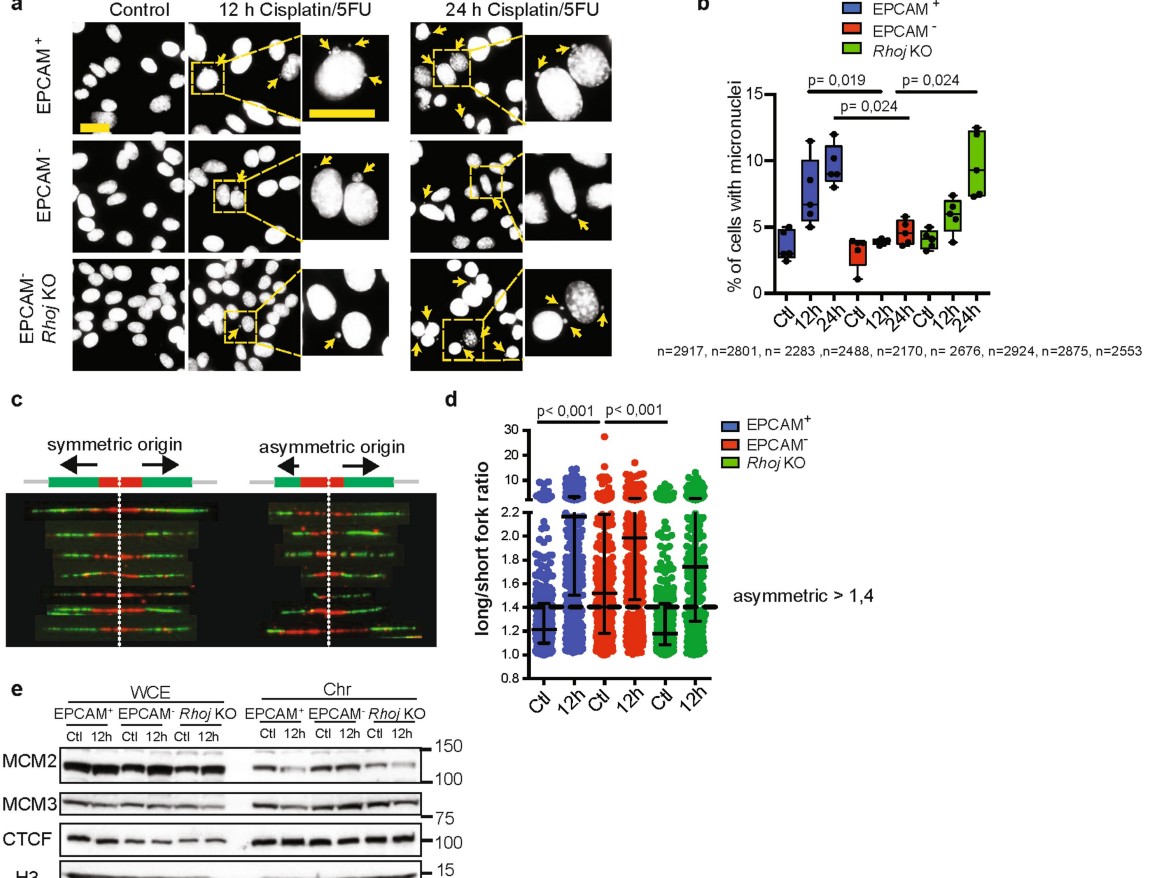

n=2917, n=2801, n= 2283 ,n=2488, n=2170, n= 2676, n=2924, n=2875, n=2553

asymmetric > 1,4

**Extended Data Fig. 8 | Replicative stress in EMT tumour cells is associated with the activation of dormant origins in a RHOJ-dependent manner that prevents micronuclei formation. a-b**, Representative images (a) and quantification (b) of micronuclei in EPCAM+, EPCAM− and EPCAM− *Rhoj* KO cells untreated and treated with cisplatin/5FU for 12 and 24h. Nuclei are stained with DAPI. Yellow arrowheads indicate micronuclei (a) (Scale bar, 25μm, n represents the number of nuclei pooled from five independent experiments, Kruskal-Wallis test followed by two-sided Mann-Whitney paired comparisons tests. p-values adjusted for multiple comparisons by Bonferroni correction, box boundaries represent 25th and 75th percentiles; whiskers, minimum and maximum; centre line, median). **c-d**, Representative images (c) and

quantification (d) of fork asymmetry to measure fork stalling in EPCAM+, EPCAM− and EPCAM− *Rhoj* KO cells untreated and treated with cisplatin/5FU for 12 h. The degree of symmetry around the replication origin was calculated as long/short fork ratio (100 individual forks were measured for each cell line in three replicates, medians with interquartile range are shown, ANOVA with condition, experience and their interaction, p-values after two-way post-hoc Sidak tests). **e**, Western blot analysis of the indicated replication factors and loading control in the whole cell extract and loaded on chromatin in untreated conditions and 12h after cisplatin/5FU administration (n = 2, molecular weights (kDa) are indicated on the right side of the blots).

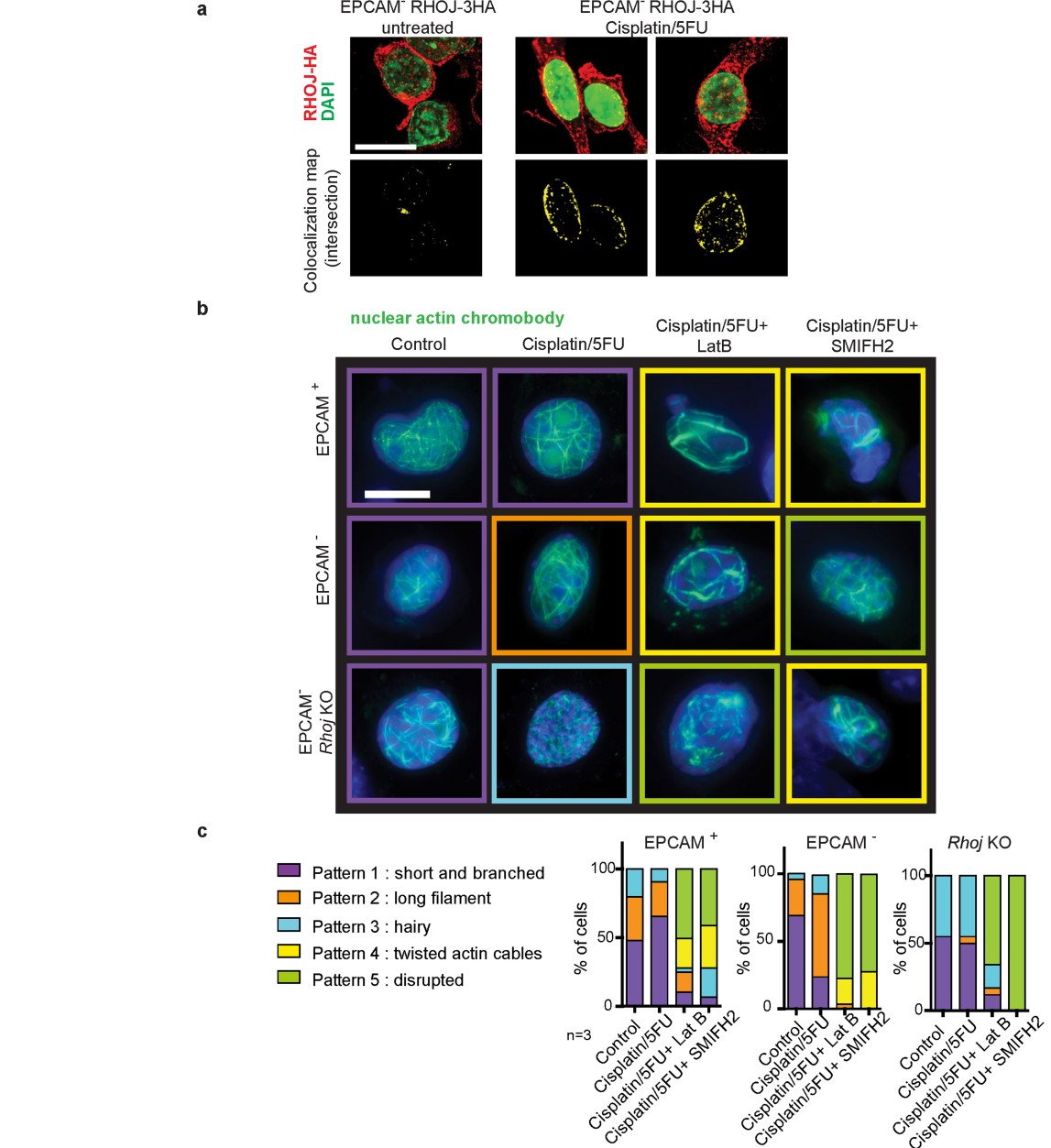

**Extended Data Fig. 9 | Patterns of nuclear actin filaments in response to chemotherapy, RHOJ deletion and actin polymerization inhibitor.**
**a**, Representative images of EPCAM⁻ cells transfected with HA-tagged RHOJ untreated and 12h after cisplatin/5FU treatment. Maximum intensity projections are shown (n = 3 biological replicates; anti-HA in red; nuclei are counterstained with DAPI in green; scale bar, 20μm) (top). Corresponding colocalization map of the overlap between HA-tagged RHOJ and DAPI signals showing peri- and intra-nuclear localization of RHOJ. Global intersection coefficient is represented (bottom). **b**, Representative images of the different nuclear actin patterns found in EPCAM⁺, EPCAM⁻ and EPCAM⁻ *Rhoj* KO cells expressing nuclear actin chromobody-GFP treated with cisplatin/5FU and a combination of cisplatin/5FU + Latrunculin B (LatB, 400nM) or SMIFH2 (50 μM) for 12h. Nuclei are counterstained with DAPI (blue) (n = 3; scale bar, 10μm). **c**, Quantification of nuclear actin patterns in EPCAM⁺, EPCAM⁻ and EPCAM⁻ *Rhoj* KO cells treated

with cisplatin/5FU and a combination of cisplatin/5FU + Latrunculin B or SMIFH2 for 12h (Means are shown, n represents the number of independent primary cultured cell lines). Several classes of nuclear actin filaments are observed (b). Pattern 1, thin, short and branched filaments, is found in all cell types as shown in panel highlighted by a purple border ; pattern 2, elongated filaments, is mainly induced in EPCAM⁻ cells upon chemotherapy treatment as shown in panel highlighted by an orange border; pattern 3, hairy filaments which are short filaments with a dense and multipolar organization, is detected in EPCAM⁻ *Rhoj* KO cells, as shown in panel highlighted by a blue border; pattern 4, thick and twisted actin filaments as shown in panel highlighted by a yellow border and pattern 5, severe disruption of actin filaments as shown in panel highlighted by a green border, are found when chemotherapy was combined with F-actin inhibitors.

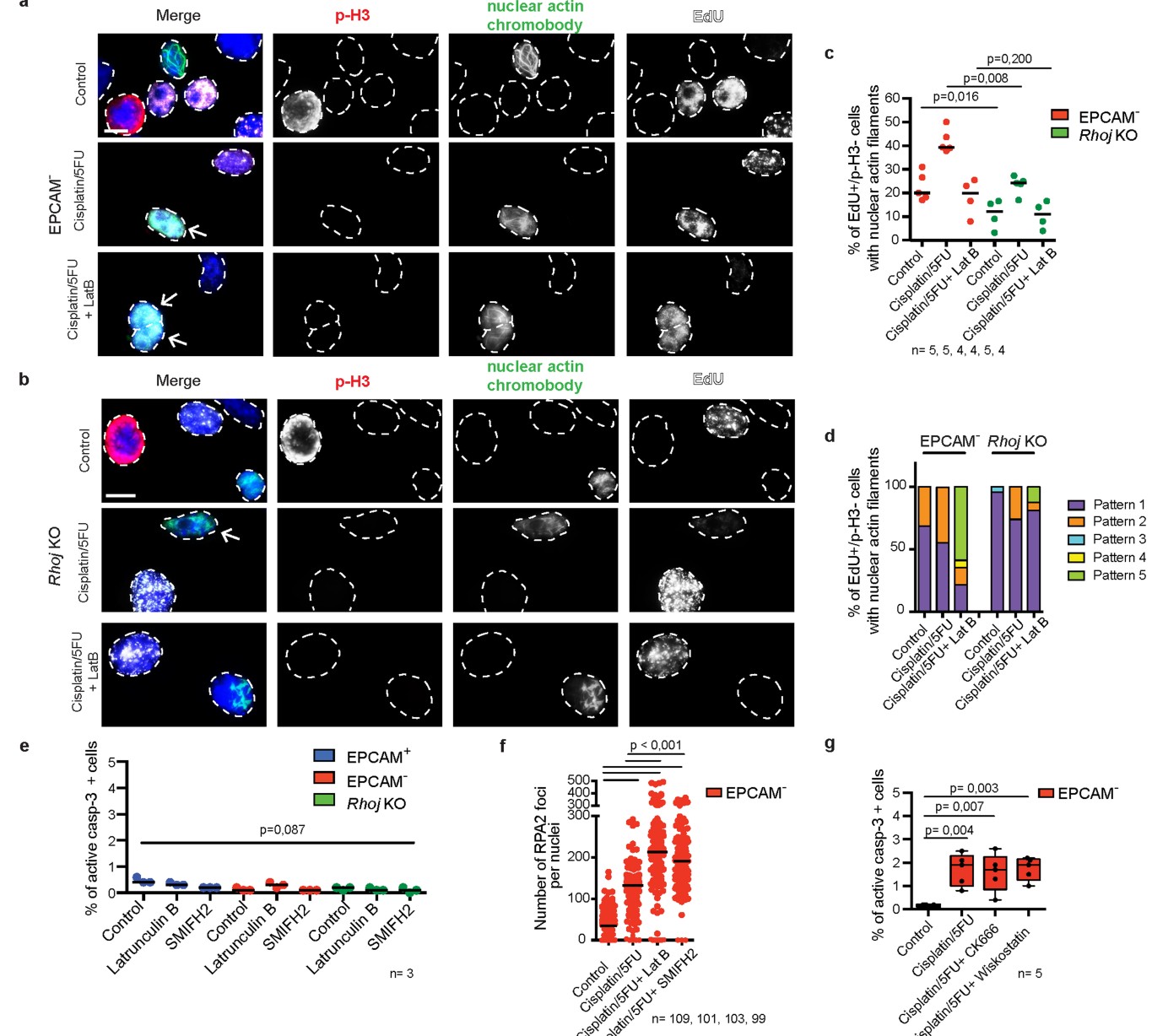

**Extended Data Fig. 10 | Nuclear actin filament formation occurs in replicative cells and participates in DNA repair and cell survival in response to chemotherapy. a-b**, Representative immunofluorescence of phospho-histone H3 (p-H3) (red) and EdU (white) in EPCAM⁻ (a) and EPCAM⁻ *Rhoj* KO (b) cells expressing nuclear actin chromobody-GFP treated with cisplatin/5FU or a combination of cisplatin/5FU and Latrunculin B for 12h compared to untreated conditions. Nuclei are counterstained with DAPI (blue). Arrowhead indicates a nucleus presenting co-localization of nuclear actin filament and EdU (Scale bar, 10µm). **c**, Quantification of EdU positive/ p-H3 negative cells in EPCAM⁻ and EPCAM⁻ *Rhoj* KO tumour cells presenting nuclear actin filaments of (a) and (b) (>100 cells counted per condition; n represents the number of replicates from 2 independent cell lines, medians are shown, two-sided Mann-Whitney U-tests). **d**, Quantification of the proportion of each nuclear actin patterns in EPCAM⁻ and EPCAM⁻ *Rhoj* KO cells presenting nuclear actin filaments of (a) and (b) (Means are shown). **e**, FACS quantification

of the percentage of activated caspase-3 positive cells in EPCAM⁺, EPCAM⁻ and EPCAM⁻ *Rhoj* KO in response to Latrunculin B or SMIFH2 for 24h. (Medians are shown, Kruskal-Wallis test, n represents the number of independent primary cultured cell lines). **f**, Quantification of RPA2 foci in EPCAM⁻ cells treated with a combination of cisplatin/5FU with Latrunculin B or SMIFH2 inhibitor for 12h. (Medians are shown, Kruskal-Wallis test, p-values adjusted for multiple comparisons by Bonferroni correction, n represents the number of analysed nuclei pooled from two independent primary cultured cell lines). **g**, FACS quantification of the percentage of activated caspase-3 positive cells in EPCAM⁻ tumour cells in response to cisplatin/5FU combined with ARP2/3 inhibitor (CK666, 50 µM) or N-WASP inhibitor (Wiskostatin, 5 µM) for 24h (box boundaries represent 25th and 75th percentiles; whiskers, minimum and maximum; centre line, median; n represents the number of replicates from 3 independent cell lines, Kruskal-Wallis test followed by two- sided Mann Whitney tests). For gel source data, see Supplementary Figure 1 and 2.

# Reporting Summary

## Statistics

For all statistical analyses, confirm that the following items are present in the figure legend, table legend, main text, or Methods section.

| n/a | Confirmed | |
|---|---|---|
| ☐ | ☒ | The exact sample size (*n*) for each experimental group/condition, given as a discrete number and unit of measurement |
| ☐ | ☒ | A statement on whether measurements were taken from distinct samples or whether the same sample was measured repeatedly |
| ☐ | ☒ | The statistical test(s) used AND whether they are one- or two-sided *Only common tests should be described solely by name; describe more complex techniques in the Methods section.* |
| ☒ | ☐ | A description of all covariates tested |
| ☐ | ☒ | A description of any assumptions or corrections, such as tests of normality and adjustment for multiple comparisons |
| ☐ | ☒ | A full description of the statistical parameters including central tendency (e.g. means) or other basic estimates (e.g. regression coefficient) AND variation (e.g. standard deviation) or associated estimates of uncertainty (e.g. confidence intervals) |
| ☐ | ☒ | For null hypothesis testing, the test statistic (e.g. *F, t, r*) with confidence intervals, effect sizes, degrees of freedom and *P* value noted *Give P values as exact values whenever suitable.* |
| ☒ | ☐ | For Bayesian analysis, information on the choice of priors and Markov chain Monte Carlo settings |
| ☒ | ☐ | For hierarchical and complex designs, identification of the appropriate level for tests and full reporting of outcomes |
| ☒ | ☐ | Estimates of effect sizes (e.g. Cohen's *d*, Pearson's *r*), indicating how they were calculated |

*Our web collection on statistics for biologists contains articles on many of the points above.*

## Software and code

Policy information about availability of computer code

| Data collection | FACS ARIA III<br>Axiovision release 4.6<br>Zen Blue 3.3 (ZEISS) |
|---|---|
| Data analysis | STAR software (2.4.2a)<br>Prism 5 (Graphpad)<br>FACSDiva  Software 8.0.1<br>FlowJo v10<br>MaxQuant (version 1.5.8.3) and (version 2.0.3.0)<br>Perseus Software (version 1.5.5.3)<br>Adobe Photoshop and Illustrator CS6 (Figure preparation)<br>R software (3.2.3)<br>ImageJ 1.53j<br>IBM-SPSS V28.0<br>Medcalc  V 20<br>Huygens Professional 22.04 deconvolution software<br>CellProfiler 3.1.9 software |

For manuscripts utilizing custom algorithms or software that are central to the research but not yet described in published literature, software must be made available to editors and reviewers. We strongly encourage code deposition in a community repository (e.g. GitHub). See the Nature Portfolio guidelines for submitting code & software for further information.

## Data

Policy information about [availability of data](availability of data)

All manuscripts must include a [data availability statement](data availability statement). This statement should provide the following information, where applicable:
- Accession codes, unique identifiers, or web links for publicly available datasets
- A description of any restrictions on data availability
- For clinical datasets or third party data, please ensure that the statement adheres to our [policy](policy)

The mass spectrometry proteomics data have been deposited to the ProteomeXchange Consortium via the PRIDE83 partner repository with the dataset identifier PXD025737 and the dataset identifier PXD038278.
The accession number for the RNA sequencing reported in this paper is GEO: GSE205985.

# Field-specific reporting

Please select the one below that is the best fit for your research. If you are not sure, read the appropriate sections before making your selection.

☒ Life sciences      ☐ Behavioural & social sciences      ☐ Ecological, evolutionary & environmental sciences

For a reference copy of the document with all sections, see [nature.com/documents/nr-reporting-summary-flat.pdf](nature.com/documents/nr-reporting-summary-flat.pdf)

# Life sciences study design

All studies must disclose on these points even when the disclosure is negative.

| | |
|---|---|
| Sample size | Samples sizes are indicated in the figures for each experiment. Samples sizes for experiments involving mice was determined accordingly to the protocols 434N and 663N, stating the number of mice used for experiments , should be reduced to the minimum as soon as the result is reproducible within each experiment. For in vitro experiments, samples sizes were determined based on previous experience in the lab and previously published studies of a similar nature (Latil Cell Stem Cell 2017, Pastushenko Nature 2020, Ibarra PNAS 2008, Lamm Nature Cell Biology 2020, Murai Molecular Cell 2018, Schrank Nature 2018)  . No statistical methods were used to predetermine sample size. |
| Data exclusions | No data were excluded from the analysis |
| Replication | Each experiment was repeated at least three times to confirm reproducibility of findings, except for RNAseq analysis for which experiment was repeated twice. n is described in figure legends. |
| Randomization | For experiments involving cell culture, all tumor cell types (Epcam+, Epcam- and RhoJ KO Epcam- tumor cells and human MDA-MB231) were treated with all the drugs so no allocation in groups or randomization was required. For in vivo studies on primary mouse models , the animals were selected according to their correct genotypes.  The mice were induced with Tamoxifen injection 28-35 days after birth and the mice developed tumors in 2-3 months thus minimizing the difference in age of different animals used. When tumor size reached between 2 and 5 mm3, mice were treated with chemotherapy injected intraperitoneally and were compared with tumors developed in mice of the same genotype injected with physiological serum. |
| Blinding | Investigators were blinded to mouse and cell line genotypes  or treatment conditions during experiments, for performing sample analysis, imaging and quantification. |

# Reporting for specific materials, systems and methods

We require information from authors about some types of materials, experimental systems and methods used in many studies. Here, indicate whether each material, system or method listed is relevant to your study. If you are not sure if a list item applies to your research, read the appropriate section before selecting a response.

## Materials & experimental systems

| n/a | Involved in the study |
|---|---|
| ☐ | ☒ Antibodies |
| ☐ | ☒ Eukaryotic cell lines |
| ☒ | ☐ Palaeontology and archaeology |
| ☐ | ☒ Animals and other organisms |
| ☒ | ☐ Human research participants |
| ☒ | ☐ Clinical data |
| ☒ | ☐ Dual use research of concern |

## Methods

| n/a | Involved in the study |
|---|---|
| ☒ | ☐ ChIP-seq |
| ☐ | ☒ Flow cytometry |
| ☒ | ☐ MRI-based neuroimaging |

# Antibodies

| | |
|---|---|
| Antibodies used | For immunostaining, the following primary antibodies were used: goat GFP (Abcam, ab6673, 1:500), chicken K14 (Thermo Fisher Scientific, #MA5-11599, 1:2000), rabbit Vimentin (Abcam, ab92547, 1:500), rabbit active caspase-3 (R&D, AF835, 1:600), rabbit 53bp1 (Novus, NB100-304, 1:200), mouse phospho-histone H2A.X (Ser139) (Millipore, 05-636, 1:500), mouse RAD51 (Santa-Cruz Biotechnology, sc-398587, 1:50), rat RPA32/RPA2 (Cell Signaling, #2208, 1:500), Rhodamine Phalloidin (Thermo Fisher Scientific, R415, 1:400), rabbit HA (Abcam, ab9110, 1:1000), mouse PCNA (Abcam, ab29, 1:1000), rat phospho-histone H3 (S28) (Abcam, ab10543, 1:2000), rabbit Ki67 (Abcam, ab15580, 1:400). Validated for IF by manufacturer.<br><br>For western blotting, the following primary antibodies were used: rabbit phospho-histone H2A.X (Ser139) (Cell Signaling, #2577, 1:800), rabbit histone H2A.X (Cell Signaling, #2595, 1:1000), rabbit phospho-ATM/ATR substrate (Cell Signaling, #9607, 1:750), rabbit beta-Actin (Abcam, ab8227, 1:2000), mouse HA (Roche, #11583816001, 1:1000), rabbit phospho -CDC2 (Tyr15) (Cell Signaling, #4539, 1:1000), ), rabbit phospho-CDC2 (Thr161) (Cell Signaling, #9114, 1:1000), rabbit CDC2 (Cell Signaling, #77055, 1:1000), rabbit phospho-CDK2 (Thr160) (Cell Signaling, #2561, 1:1000), rabbit CDK2 (Cell Signaling, #18048, 1:1000), rabbit CDK4 (Abcam, ab199728, 1:1000), rabbit CDK6 (Cell Signaling, #3136, 1:1000), rabbit N-WASP (Cell Signaling, #4848, 1:1000), mouse POLD (Santa Cruz #sc-373731), mouse PCNA (Santa Cruz #sc-56), rat RPA32 (Cell Signalling #2208), rabbit phospho-RPA32 S4/S8 (Bethyl #A300-245A), rabbit CTCF (Millipore #07-729), mouse alpha-tubulin (Sigma #T9026), rabbit H3 (Abcam #ab1791). The following secondary antibodies were used: ECL anti-Rabbit, anti-rat or anti-mouse IgG conjugated with horseradish peroxidase (#NA934, #NA935, #NA931,GE Healthcare, 1:2000 or 1:5000). Validated for WB by manufacturer. Affinity-purified mouse monoclonal antibodies NB8-AD9 (WB/IP) raised against human phospho-CDK4 (Thr172) have been validated for WB by Coulonval et al. 2022(1:500). Rabbit MCM2 and rabbit MCM3 have been validated for WB and described by Mendez and Stillman, 2000.<br><br>For immunodetection of labeled tracks, the primary antibodies used are :for CldU, rat anti-BrdU # ab6326 Abcam; for IdU, mouse anti-BrdU #347580 BD Bioscience, the corresponding secondary antibodies used are anti-rat IgG AF594, #A-11007; anti-mouse IgG1 AF488, #A-21121; all from Molecular Probes. Mouse anti-ssDNA antibody was used to assess fiber integrity (#MAB3034 Millipore, secondary antibody anti-mouse IgG2a AF647, #A-21241 Molecular Probes).<br><br>Immunostaining for FACS analysis was performed using PE-conjugated anti-CD45 (clone 30F11,#103114, 1:100, eBioscience), PE-conjugated anti CD31 (clone MEC13.3; #102508, 1:100, BD PharMingen), and APC-Cy7-conjugated anti-EpCAM (clone G8.8;#118218, 1:100, Biolegend), PE-conjugated anti-CD51 (rat, clone RMV-7, Biolegend #104106, 1:50), BV421-conjugated anti-CD61 (Armenian hamster, clone 2C9.G2, BD Bioscience #553345, 1:50),  biotin-conjugated anti-CD106 (rat, clone 429 (MVCAM.A), BD Bioscience, #553331, 1:50), BV711- conjugated anti-Epcam (rat, clone, G8.8, BD Bioscience, #563134, 1:100, PerCPCy5.5 conjugated anti-CD45 (rat, clone 30-F11, BD Bioscience, #550994, 1:100), and PerCPCy5.5 conjugated anti-CD31 (rat, clone MEC13.3, BD Bioscience, #562861, 1:100), PE Anti- Active Caspase-3 (BD Pharmingen #550821, 1:25) and PE Anti- H2AX (pS139) (BD pharmingen #562377, 1:20), Alexa Fluor 647 anti-BrdU (BD Pharmingen #560209, 1:50). Validated for flow cytometry by manufacturer.<br><br>Immunoprecipitation were performed using 6µg of rabbit Igg control Chip grade (Abcam, ab171870) ; rabbit IPO9 (A305-475A , Bethyl Lab). Validated for immunoprecipitation by manufacturer. |
| Validation | Affinity-purified mouse monoclonal antibodies NB8-AD9 raised against human phospho-CDK4 (Thr172) was characterized in Coulonval et al. (2022) 10.1080/15384101.2021.1984663 and allow the direct detection of endogenous phospho-CDK4 from human and mouse cells by a variety of techniques including immunoblotting, immunoprecipitation.<br>The other antibodies used are commercially available and were validated by the provider. We used protocols and recommandations of the manufacturer on validated species. |

# Eukaryotic cell lines

Policy information about cell lines

| | |
|---|---|
| Cell line source(s) | Primary mouse skin SCC cell lines isolated from Lgr5/Kras/p53 RhoJ WT and RhoJ cKO skin SCC, human MDA-MB-231 cell line (ATCC HTB-26) |
| Authentication | the cell lines have not been authenticated |
| Mycoplasma contamination | All cell lines have been tested negative for mycoplasma contamination |
| Commonly misidentified lines (See ICLAC register) | No commonly misidentified cell lines were used in this study according to ICLAC register version II. |

# Animals and other organisms

Policy information about studies involving animals; ARRIVE guidelines recommended for reporting animal research

| | |
|---|---|
| Laboratory animals | Rosa26-YFP, Rosa-tDTomato, K14CreER , Lgr5CreER, KRasLSL-G12D and p53fl/fl mice have been imported from the NCI mouse repository and the Jackson Laboratories. RhoJ fl/fl mice were a kind gift from A. Uemura (Department of Retinal Vascular Biology, Nagoya City University Graduate School of Medical Sciences, Japan).  All mice used in this study were composed of males and females with mixed genetic background. Mouse colonies were maintained in a certified animal facility in accordance with the European guidelines. All the experiments were approved by the Ethical Committee for Animal Welfare (Commission d' Ethique et du Bien Etre Animal, CEBEA, Faculty of Medicine, Université Libre de Bruxelles, reference no. 434N and 663N).<br>Lgr5CreER/Kras/p53/RYFP, Lgr5CreER/Kras/p53/RHOJ KO/RYFP, Lgr5CreER/Kras/p53/tdTomato and Lgr5CreER/Kras/p53/RHOJ KO/tdTomato were induced with tamoxifen at 28-35 days after birth. Tumor appearance and size were detected by daily observation and palpation. Mice were euthanized when tumor size reached 1cm3 or when mice presented signs of distress or lost >20% of its initial weight. For grafting experiments, NUDE mice were used with age ranging from 4 to 8 weeks. |
| Wild animals | No wild animals were used in this study |

| Field-collected samples | No field-collected samples were used in this study |
|---|---|
| Ethics oversight | Mice colonies were maintained in a certified animal facility in accordance with European guidelines.The experiments were approved by the local ethical committee (CEBEA) under protocols 434N and 663N. The study is compliant with all relevant ethical regulations regarding animal research. |

Note that full information on the approval of the study protocol must also be provided in the manuscript.

# Flow Cytometry

## Plots

Confirm that:

☒ The axis labels state the marker and fluorochrome used (e.g. CD4-FITC).

☒ The axis scales are clearly visible. Include numbers along axes only for bottom left plot of group (a 'group' is an analysis of identical markers).

☒ All plots are contour plots with outliers or pseudocolor plots.

☒ A numerical value for number of cells or percentage (with statistics) is provided.

## Methodology

| Sample preparation | Tumors were dissected, minced and digested in collagenase I (Sigma) during 2 hr at 37°C on a rocking plate. Collagenase I activity was blocked by the addition of EDTA (5 mM) and then the cells were rinsed in PBS supplemented with 2% FBS. Before the staining, cells were blocked during 20 min at room temperature in PBS supplemented with 30% FBS. Cell suspensions were filtered through a 70 µm cell strainers (BD) then through a 40 µm cell strainer to ensure the elimination of cell debris and clumps of cells. |
|---|---|
| Instrument | FACSaria III and LSRFortessa (BD Bioscience) |
| Software | FACSDiva Softwara 8.0.1 (BD Bioscience) and FlowJo v10 |
| Cell population abundance | The proportion of YFP+ tumor cells in Lin- population varied from 20 to 95%. The proportion of tumor cell subpopulations within YFP+ tumor cells varied depending on the tumor type and between individual tumors. |
| Gating strategy | Living cells were selected by forward scatter, side scatter, doublets discrimination and by Hoescht dye exclusion. Tumor cell subpopulations were selected based on the expression of YFP or Tomato and the exclusion of CD45 and CD31 |

☒ Tick this box to confirm that a figure exemplifying the gating strategy is provided in the Supplementary Information.

