## [Peer Review File · Nature]

Manuscript Title: RhoJ controls EMT associated resistance to chemotherapy

Reviewer Comments & Author Rebuttals

Reviewer Reports on the Initial Version:

Referee expertise:

Referee #1: Rho GTPases, cytoskeleton

Referee #2: DNA repair, replication stress, genomic instability

Referee #3: EMT

Referee #4: skin cancer models

Referees' comments:

Referee #1 (Remarks to the Author):

This is an interesting paper that describes a role for the small GTPase RhoJ in resistance to chemotherapy. Previous work has shown that RhoJ promotes resistance to chemotherapy in melanoma cells by targeting ATM signalling and so reducing apoptosis in response to DNA damage. Here the authors discover what seems to be an additional pathway in squamous cell carcinoma, whereby RhoJ upregulation controls additional elements of the DNA damage response. The findings are important; however, some of the key findings need to be put on a firmer footing.

Specific points

In giving the background to what is known about RhoJ and cancer, the authors should reference the work of Chen et al. (Front Cell Dev Biol 8:832).

The Methods state that the statistical analyses are described in the Figure Legends, but the Figure legends give only partial information; often just stating the post-test used. Some of the statistical analysis is inappropriate; for example, the use of a t-test for the multiple comparisons in Figure 2o. That Figure legend also describes having n=12 replicates from 2 independent experiments, which sounds like an n=2 experiment is being analysed like an n=12 experiment. The authors need to give a full account of how all of the statistical analysis was done. It is possible that this will reveal the need for additional experimental work.

Figure 1. This is a small point, but the graph in Figure 1b would be better as a histogram and one that shows the individual tumor sizes as datapoints. We don't really care about the average tumour size of these arbitrary groupings, so the s.e.m. is meaningless, but it would be useful to see the range of

sizes in each group.

Figure 2. The authors perform RNAseq to identify genes that change in SSC cells that have undergone EMT. The raw data need to be presented as a Supplementary file.

Figure 2. The data in Figures 2a and 2b are critical to the paper. These two figures are the only evidence for increased RhoJ expression in the SSC cells that have undergone EMT. Figure 1a shows the average of RNAseq measurements of RhoJ mRNA from two independent samples of sorted cells. Figure 1b shows (I think) n=5 qPCR measurements from either 5 independent sets of sorted cells, or 5 measurements on replicates of the same cells – this isn't clear. In either case, there isn't enough work to be able to accurately estimate the frequency or degree of RhoJ upregulation in SSC. This is the foundational observation of the paper, and the authors need a much bigger dataset and more detailed analysis.

In Figure 2d, the authors report differences in cell number in isolated cancer cells depleted of RhoJ. The data do not reach statistical significance and the authors need to either increase the power or change how they describe this result. The same is true for Figure 2f.

Figure 2n, o. The authors look briefly at RhoJ MDA-MB-231 breast cancer cells. They do not show if MDA-MB-231 cells upregulate RhoJ, or if RhoJ levels change with EMT in breast cancer. The effects of silencing RhoJ are slight and there are issues with the statistical analysis (see first point). I don't think these panels add anything to the paper and they should be removed.

Figure 3. The authors use RNAseq to look for genes regulated by RhoJ in the cells undergoing EMT. They also use iPOND to look for changes in protein recruitment to replication forks. These screens produce a lot of hits, and these are detailed in the text. But there is no validation of any of the candidates. I return to this point later. The authors need to provide the raw data for each screen in the Supplementary material.

Figure 4d. The authors need to revise their interpretation a little. The overall results back up the hypothesis, but it isn't as straightforward as described in the text. For example, the recruitment of RAD51 goes down with RhoJ overexpression at 12h. The experimental work is fine, but the description needs to match the data better.

Figure 5. The quantified effects of RhoJ on nuclear actin are interesting, but the images don't need to be in the main Figures. And it would be better if they included representative images from the key conditions.

Figure 6. The final Figure examines how RhoJ affects cell cycle arrest and replication after DNA damage. The SSC cells that have undergone EMT are more resistant to replication arrest and this depends on RhoJ. This is an interesting finding, but it leads to the biggest issue with the manuscript, which is that we don't really know how RhoJ is acting here. The authors look in a number of places – we know that RhoJ changes expression of DNA repair enzymes, that it changes recruitment of proteins to replication forks, that it can regulate nuclear actin and that it can help maintain replication after DNA damage. Some of these readouts might be connected, but the problem is that

we are left with a number of partial possible solutions to how RhoJ is acting, but no clear answer. For publication at this level, the authors need to work out how RhoJ is doing this.

In summary, this is an interesting and potentially important paper, but the key findings are not fully resolved or fully supported. It is important that the authors properly determine the frequency and extent of RhoJ overexpression in SSC. And it is also critical that the authors find an explanation for the role of RhoJ in chemoresistance. Currently they present a number of possibilities, but none of these is taken to a conclusive answer.

Referee #2 (Remarks to the Author):

In their manuscript, Debaugnies et al. identify and characterize the small Rho GTPase RhoJ as a potential driver of chemotherapy resistance in cancer cells that have undergone Epithelial-to-Mesenchymal Transition (EMT) using a skin squamous cell carcinoma (SCC) mouse model. The authors first show that RhoJ is overexpressed in EMT SCC cells that are resistant to standard SCC chemotherapy and that loss of RhoJ re-sensitizes these cells to treatment. They then try to dissect the molecular mechanism underlying RhoJ-mediated chemotherapy resistance, concluding that RhoJ upregulates DNA damage response pathways and confers resistance to replication stress.

While several groups have shown previously that GTPases of the Rho family, including RhoJ, are connected to the EMT process in various tumour types, Debaugnies et al. demonstrate that RhoJ expression status is specifically linked to therapy resistance in cells that have undergone EMT, at least in their mouse SCC model and also in a human breast cancer cell line. While this finding has important implications for tumor therapy, particularly with regards to patient stratification and prognosis prior to therapy of metastasized cancers, its novelty is somewhat dampened by a previous report, which showed that RhoJ is overexpressed in metastatic melanoma (DOI: 10.1158/0008-5472.CAN-12-0775; the authors briefly discuss this paper in the text). As to the link between RhoJ overexpression and a stunted replication stress response, the evidence provided in the manuscript remains superficial, and the authors do not provide a convincing explanation of how RhoJ promotes replication stress tolerance on a molecular level. Overall, I feel that the manuscript, while conceptually intriguing, contains limited new insights into the roles of RhoJ in either EMT or the DNA damage response.

Major points/ suggestions:

My main concerns all center around the mechanism of the replication stress tolerance in Epcam-cells:

- The conclusion that Epcam- cells tolerate higher levels of replication stress is mostly based on the finding that origin firing is increased in these cells. However, is this accompanied by higher levels of fork breakage or the formation of micronuclei, or do the cells only display fork stalling? This is important for the model. The nature of the replication stress needs to be better characterized.
- In the same vein, the authors argue that “Epcam- resistance to therapy is independent of ATR-

mediated checkpoint" signalling. However, have they excluded that ATR expression or activation is downregulated in Epcam- cells? The lower levels of RPA focus formation could be explained by a lack of ATR. Is the intra-S checkpoint activated at all?

- And what would be the nature of an ATR-independent resistance mechanism? Is lesion bypass upregulated? Or if there is increased DNA breakage (as suggested by elevated 53BP1 levels), how do cells deal with these lesions and avoid cell death? Increased efficiency of DSB repair? Is the ATM checkpoint active?
- The Te Riele lab has shown that inactivation of p53 can suppress replication stress-induced DNA damage (DOI: 10.7554/eLife.37868). The authors need to rule out lack of p53, in the context of RhoJ overexpression, has a similar effect in their SCC model. Why did the authors look at CDK1 activation but not CDK2, the latter of which is an important component of the intra-S-checkpoint?
- Figure 5: this figure is totally out of context. I did not understand how it relates to either the preceding or the following figure. In addition, it is poorly labelled and annotated. How does nuclear actin polymerization promote RhoJ-mediated replication stress tolerance? While the authors speculate about this point in the discussion, the impact of the paper would be much higher if they could experimentally connect RhoJ expression levels and changes in actin polymerization to the activity of components of the replication stress machinery in their model system. Do Arp2/3 localize to stalled or collapsed replication forks in Epcam- cells? Is their activity important for origin firing in Epcam- cells? Does inhibition of nuclear actin polymerization re-sensitize Epcam- cells to chemotherapy?

Other points:

- Figure 1I: The resistance of Epcam- cells to Doxorubicin, Paclitaxel and IR appears to be minor compared to CDDP/5FU or Gemcitabine. This difference should be commented on in the text. The text discussing this figure is misleading in its current form.
- Figure 2D: Epcam+ control is missing. No statistical analysis.
- Figure 2C: Epcam+ control is missing.
- Figure 2F: No statistical analysis.
- Page 5: "In the absence of chemotherapy, we found that the majority of tumors (62%) were mixed tumors presenting epithelial (Epcam+) and mesenchymal (Epcam-) TCs, 29% were fully mesenchymal tumors with more than 90% of Epcam- TCs, and 9% were well-differentiated epithelial tumors with more than 90% of Epcam+ TCs." – Which figure do these numbers refer to? This part of the paper was very confusing to read and hard to understand.
- Figure 3C: The Epcam- cells transfected with control hairpins show a clear reduction in expression levels relative to Epcam- WT cells (particularly in the upper part of cluster 1). Wouldn't this suggest that the shRNA transduction alone already causes changes in gene expression?
- Figure 6C: total CDK1 is missing.
- Data and methodology: Overall, the scientific approach is valid. The methods section is reasonably detailed although it is not clear how DNA damage foci were quantified in ImageJ. Was this an automated analysis or was it done by hand? Have the authors taken into account different nucleus sizes? In Figure 4D, nuclear sizes appear quite variable – this would obviously affect the total number of foci per cell.
- Statistics: Mostly appropriate. However, there are several graphs that lack statistical analysis. In addition, is there a reason why the authors summarize their data in many figures with SEM rather

than SD? Wouldn't they want to indicate the variability within their samples rather the proximity to the population mean?

- Clarity and context: Overall, the manuscript is well structured. However, the authors tend to overstate but not explain certain important ideas. For example, what are the "implications for the development of new strategies to overcome resistance to anticancer therapy" (abstract and last sentence of the discussion)? Given the difficulties to develop clinically relevant Rho inhibitors, this is a bold statement and, at the minimum, warrants further discussion in the text. Also, the statement that "cellular effectors of DDR are not often activated at the mRNA level" is simply not true and this is also as such not stated in the cited review by Polo and Jackson. Please amend. Readability of the manuscript could be further improved 1) by using fewer acronyms (do "tumor cells" really need to be abbreviated as TC?), 2) by making sure to define all acronyms (for example, what does "Epcam Co sh" mean?! I assume it refers to the control hairpin, but I shouldn't have to guess) and 3) by removing the many typos.

o Page 5: They should briefly explain that Vimentin is used as a marker for mesenchymal cells and K14 is a marker for epithelial cells to make it easier to understand the experimental set up and the connection to EMT.

- References: Overall, comprehensive and correct. The authors might want to consider adding two recent publications that further link RhoJ expression to EMT (DOI: 10.1016/j.bbrc.2021.06.004 and DOI: 10.1007/s13311-020-00910-w).

Referee #3 (Remarks to the Author):

Debaugnies et al. aim to unravel mechanisms underlying the EMT-associated therapy resistance in cancer. A causal link of therapy resistance (to various modalities, including chemotherapy, radiotherapy and targeted therapy) and EMT is evident since many years. However the underlying molecular mechanisms are not well understood. Known resistance mechanisms include increased expression of drug efflux pumps and drug-metabolizing enzymes, as well as general inhibition of apoptotic pathways due to an EMT-associated stemness state. The authors used a genetic mouse model of skin squamous carcinoma (SCC), which is well established in the group, isolated an epithelial fraction of tumor cells (Epcam+ TCs), which is highly sensitive to chemotherapy (Cisplatin and 5-FU) and a partially mesenchymal fraction (Epcam- TCs), which is highly resistant. Using this pair of cell types they identified the small GTPase RhoJ to be preferentially expressed in Epcam- TCs. By further applying proteomics, transcriptomics, overexpression and depletion studies (both in vitro and in vivo) they demonstrate that RhoJ limits chemotherapy-induced tumor cell killing by promoting DNA damage response and the new formation of replication forks, thereby attenuating replicative stress and allowing cell cycle progression and proliferation under chemotherapy. Since therapy resistance, often linked to an EMT-phenotype, is a major cause of cancer-associated death, the topic and the results are of high relevance and potential translational impact in cancer research and also related fields (cell biology and cell cycle regulation/DNA damage response, EMT in cancer). If validated in other systems and entities the results would be interesting for the development of novel therapeutic strategies and drug development against therapy resistance.

This is a well-presented study on a relevant topic by leading experts in the field. The outstanding

result is the description of a factor (RhoJ), which is overexpressed in the EMT state and (partially) responsible for EMT-associated therapy resistance and thus could become a promising therapeutic target. The study is well performed including proper statistical analyses.

However, open questions, discrepancies and concerns, particularly to the underlying mechanisms of the described effects, remain to be addressed:

1. The major critics concerns data presented in Fig.5: Data, particularly shown in Fig. 6, support the claim of the authors that RhoJ promotes DNA damage response and the formation of replication forks, thereby attenuating replicative stress and allowing cell cycle progression. However, the molecular link of RhoJ to these effects is likely indirect. Here the recently published role of nuclear actin in cell cycle regulation, DNA damage response and spreading of replication forks (ref. 43) comes into play. The authors describe (Fig.5), a two fold increase in nuclear actin filaments in Epcam- TCs, which can be inhibited by RhoJ KO. This is an interesting correlation, but deeper understanding if and how RhoJ is directly involved in nuclear actin (pattern) formation is lacking. Since general effects of RhoJ on DNA damage response are already described (e.g. ref.25,25) a deeper understanding of such a causal connection would strongly increase the impact and novelty of the paper. E.g. what is the role of actin polymerising/remodelling factors described to be dependent on RhoJ expression (e.g. Arp2/3 and others, see fig. 3a, c, d). In addition, since only 5-10% of cells have nuclear actin filaments, what about the 90-95% of the rest of cells? Is the expression of nuclear actin as well as the formation of different patterns depending on the cell cycle state? Is there a role/function of the different expression nuclear actin patterns (depending on RhoJ expression) on DNA damage response and therapy resistance (e.g. as described in Figs. 1 and 6)?

2. The authors state that `the proportion of Epcam+/Epcam- TCs was unchanged in WT and RhoJ KO tumors (Fig. 2i-j), showing that RhoJ does not control EMT per se. Although not being significant, there seems to be a loss of tumors with a low share of Epcam- TCs (Fig. 2j), indicating that more TCs underwent EMT in RhoJ KO tumors. This would indicate that RhoJ is inhibiting EMT, whereas in the other data RhoJ effects seem to be coupled to the (partial) EMT phenotype. This is contradictory. What is the effect of RhoJ on EMT eventually? In order to clarify this, the RhoJ KO tumors could be separated into groups (E-M-mixed) according to their differentiation state by histology and then the Epcam +/- cells grouped accordingly (as in Figure 1g). Moreover, the influence of RhoJ overexpression (Fig. 2f) on EMT marker expression and migration/invasion should be addressed in TCs in vitro in order to strengthen the authors' claim. In summary, if RhoJ would have no effect on the EMT state, but is a crucial (and sufficient! as claimed from Fig. 2f) factor for therapy resistance, why do all resistant cells undergo an EMT (as shown e.g. in Fig. 1e)? This should also be discussed/explained.

3. Fig. 2n,o: The relevance for human cancer has only been addressed in one cell line (MDA-MB-231). Here, concentrations of cyclophosphamide are very high (Molar range!), arguing for unspecific toxicity. In fact, cyclophosphamide is a pro-drug that should be metabolized in vivo by CytP450 (usually in the liver). The active, anti-proliferative and cytotoxic substances may eventually differ in vitro and in vivo. To circumvent these problems, the impact of RhoJ loss on sensitivity to other clinically relevant drugs (such as those used in Fig. 1i) should be used in more than one human cell line. In addition, it could be clarified, whether there is also a clinical impact of RhoJ expression in patients on survival and metastasis in non-melanoma patients.

4. The authors put lot of emphasis on chemoresistance in the manuscript and state that `Altogether these data demonstrate that RhoJ is both necessary and sufficient to confer resistance to

chemotherapy in both mouse and human cancers presenting EMT`. However, the conclusion that RhoJ promotes resistance is not justified by the data provided, as only a short-term (initial) cell death response had been analyzed in vitro as well as in vivo. This conclusion should be weakened or tumor-bearing mice (Ctrls and RhoJ KO) should be treated for longer periods (e.g. 4 weeks, as in Fig. 1b) to demonstrate absence/shrinkage of the non-responding fraction of tumors.

5. The role of RhoJ in cell cycle progression and DDR remains unclear. Specific DNA damage responses are tightly interlinked with distinct cell cycle phases and progression under unperturbed conditions and under genotoxic stress, such as chemotherapy. For example, RAD51 and RPA foci formation are restricted to S and G2 phases of the cell cycle. However, the authors incompletely reconcile their DDR, fiber assay and cell cycle data. Several critical points and discrepancies have to be addressed:

a) The authors state that “No difference in cell cycle distribution was observed between Epcam+, Epcam- and RhoJ KO TCs in untreated condition.” However, the representative flow plots in Fig. 6a, show that RhoJ KO Epcam-negative TCs display more G1 and less G2 than EPCAM-negative TCs. Moreover, there seem to be less early and late S-phase cells in RhoJ KO Epcam-negative TCs than in both other cell types, implying that only mid-S phase cells (with highest replication rates) are unaffected by loss of RhoJ. Complete cell cycle profiles with statistical analyses (as in Fig. 6b) have to be provided to justify the abovementioned statement.

b) Fig. 3c shows that RhoJ KD TCs contain decreased amount of DNA replication proteins under unchallenged conditions (green pattern in heat map), which is partially restored upon chemotherapy. This implies that the major effect of RhoJ-KD may be in unchallenged S-phase of EMT-TCs rather than being a specific consequence of chemotherapy. Consistently, RhoJ-KO tumor formation is strongly impaired (Fig. 2h). Thus, the data argue for a chemo-independent role of RhoJ in cell cycle progression that may well affect DNA replication. The cell cycle profile does however not show a dramatic change in the fraction of pan S-phase cells. How can this be explained (see also below)? It is in general important to clarify, whether RhoJ KO/KD and OE has any effect on viability, proliferation and cell cycle progression under unchallenged conditions. Does RhoJ also play a role in mitosis or cytokinesis?

c) Chemotherapy with CDDP/5-FU induces DNA replication stress, triggering a DDR to activate the intra-S checkpoint, causing slowing of DNA synthesis or even its complete abolishment by replication fork stalling and inhibition of (late and backup) origin firing. For the cell cycle analysis in Fig. 6 a and b, the scheme shows that the BrdU pulse given to assess BrdU incorporation (hence DNA synthesis), is given 30 mins prior to sampling. The data shows that chemotherapy does not even slightly downregulate BrdU incorporation rate in any of the cell types, but even increases the fraction of BrdU+ cells within 12 hours (particularly in EPCAM-TCs). This means there must be massive global DNA synthesis maintained (even induced) in EPCAM-negative TCs at 12, and 24 hours of chemotherapy. Actually, at 12 hours into chemotherapy, the EPCAM- TCs accumulate cells that incorporate BrdU within 30 minutes (as compared to untreated cells). How can this stunning effect be explained? BrdU pulsing 30 mins prior to sampling should cause S-phase cells with strongly reduced DNA synthesis rates, particularly those with chemotherapy-induced (global) replication fork stalling, to become BrdU negative by falling below detection limit, thus eventually being gated as G1/G2. The authors are kindly asked to comment and to verify the treatment scheme in Fig. 6b.

d) The data on CDK1 Y15 phosphorylation (Fig. 6c) is intriguing, but cannot explain the cell cycle profiles of RhoJ KO TCs. The precise function of the different CDKs are complex-dependent and partially redundant. CDK1 is a strong driver of G2-to-M transition promoting mitotic entry, with a

smaller impact on G1-to-S transition. An increase in CDK1 Y15 would thus be expected to block entry into mitosis and thus a G2 arrest. RhoJ KO cells contain higher inhibitory Y15-CDK1 after 24 hours (Fig. 6c), but seem to undergo G1 arrest in response to chemotherapy (Fig. 6a). This is controversial and should be clarified. It would imply a more prominent role of CDK1 in G1-S transition (over mitotic entry) in the chemotherapeutic response of the TCs used in this study. To gain more insights into these essential mechanistic aspects, the activity of CDK 2,4,6 as predominant G1-S-transition drivers, should be analyzed to clarify the cell cycle progression phenotype of RhoJ KO TCs and the chemo-sensitization.

e) On page 15, the authors state that RhoJ is `activating both homologous recombination and non-homologous end-joining DNA repair mechanisms. This conclusion is not justified by the provided data and seems to be wrong. The presented data in figure 4d document reduced 53BP1 foci in the presence of induced RPA and RAD51 foci in RhoJ KO cells, indicating that non-homologous end joining (NHEJ) is inhibited in the RhoJ KO, but homology-directed repair (HDR) increased. Along that line, 53BP1 foci are peaking at 12h, and decrease to ctrl levels after 24h in all cell lines, suggesting transient engagement of NHEJ. Notably, the peak of 53BP1 foci is highest in EPCAM- cells, arguing for preferential NHEJ usage. This altogether suggests that RhoJ may play a role in double strand break repair pathway choice (e.g. by controlling end-resection) favoring NHEJ over HDR, despite the relative enrichment of DNA replication factors and DNA synthesis. What is the precise function of RhoJ here? This must be adequately discussed, notably in conjunction with the cell cycle profiles.

Minor points:

1) Complete lists of differential genes or proteins transcriptome and mass spectrometry data should be provided. In addition, showing comprehensive (e.g. GSEA, GO/KEGG, pathway enrichments analyses) may support the authors conclusions. For example, it is difficult to evaluate, how the modest downregulation of factors linked to actin reorganization justify the conclusion that there is a strong biological impact on actin cytoskeleton attributable to the loss of RhoJ?

2) The authors state that skin tumorigenesis in *Lgr5CreER/KRASG12D/p53cKO/Rosa-YFP* mice is induced by intraperitoneal injection of Tamoxifen. *LGR5* is strongly expressed in (tumor-initiating) stem cells in the gastrointestinal system. *KRAS* mutation and *p53* loss in these cells could cause tumor formation in the gut, and thus influence general health and chemotherapy response.

Therefore, it should be clarified/stated, whether there is gastrointestinal tumor formation in the used mice and, whether this may be changed upon RhoJ KO upon systemic Tamoxifen exposure.

3) It is unclear how many different shRNAs against RhoJ have been used, as one shRNA is shown in the figures, whereas several of them are provided in the material section. It should be clarified, how many, and which shRNA have actually been used.

4) iPOND-MS has not been performed, but data from a published study have been used. This is legitimate, but in its current wording in the main text it is misleading, "...following replicative stress, as determined by iPOND..." Although the reference is provided in the end, it should be clarified in the text as well to avoid confusion.

5) The authors state that "More importantly, the proteins differentially regulated by RhoJ KD were strongly enriched on newly replicated DNA or recruited to origin of replications following replicative stress, as determined by iPOND." It is important to avoid confusion here, as iPOND as such (without special treatments of the cells) cannot clearly distinguish between origins of replication and sites of ongoing replication.

Referee #4 (Remarks to the Author):

The Blanpain group has previously identified key steps in epithelial to mesenchymal transitions within the generation of cutaneous squamous cell carcinoma. They used the same methods here to identify populations of cells that are resistant to chemotherapy, namely the Epcam⁻ cells in the nascent tumor. They started by inducing tumorigenesis with activation of Kras and deletion of p53, and then treated with 5FU chemotherapy. They quantified the fraction of Epcam⁺ cells in tumors that responded to chemo versus those that did not, and found a tight correlation with Epcam positivity. They identified RhoJ specifically as highly upregulated in resistant cells as opposed to all other Rho isoforms. Experimentally they then deleted RhoJ while inducing tumorigenesis and found an abrogation of tumor appearance in terms of number of SCC and fraction of tumor-free mice in the absence of RhoJ. While there is numerous studies implicating RhoJ in a variety of tumors, cutaneous SCC has not been linked to RhoJ as far as this reviewer can discern.

Next, they profiled tumor samples with and without RhoJ expression and identified a number of pathways differentially expressed. Pathways enriched included many related to cell division, while those suppressed were related to DNA repair and apoptosis. This second finding led the authors to explore a role in DNA repair for RhoJ, looking at DNA damage with a variety of methods in Epcam⁺ vs Epcam⁻ in RhoJ ko tissues.

Finally, the authors attempt to link well-established roles for Rho proteins in cytoskeletal regulation to DNA-repair. They looked at the organization of nuclear actin filaments in the tumor cell types described above, and found a correlation between nuclear actin filament and loss of RhoJ expression, and suggested that RhoJ acts to regulate resistance to chemotherapy by regulating DNA-repair through control of nuclear filaments.

While a role for RhoJ has been described in many other studies in non-SCC cancers, only one previous study has implicated RhoJ and CDK in regulation of DNA-repair in cancer progression. The observations made here are interesting and well-documented, but there are some additional experiments that could improve the study.

1, throughout most of the manuscript, it is surprising that the authors did not appear to measure cell proliferation. Tumor expansion occurs upon induction of proliferation, or inhibition of apoptosis, or both. In figure 1, they measured apoptosis but not proliferation, nor did they measure tumor size across the various conditions, instead just described proportions of cells. Without a measure of proliferation or tumor size, the differences shown in proportions lack context.

2, in figure 2 the authors describe the # of SCCs and % of tumor free mice. This cancer model typically produces papilloma and scc, which have different attributes beyond just maglinant vs benign. It would be important to understand whether RhoJ ko cells fail to produce scc and/or papilloma. In addition, the authors did not show any data to prove the RhoJ ko was actually successful. The showed data for shRNA in cell lines, but not the genetic ko. It would also be helpful

to see where RhoJ is expressed in tumors vs normal skin. Also, from 2J, there are some tumors with Epcam+ cells, what happens to these tumors? are they more or less cancerous? And finally, again, what happened with proliferation in these tumors, can the phenotype only be explained by measuring apoptosis?

3, Do these tumors produce metastasis? if so, what is the effect of the various chemotherapeutics? what is effect of RhoJ ko?

4, the authors performed gene expression profiling, so they could in fact use these data to look at proliferation. Indeed, the major difference described in Fig 3 was elevated expression of genes related to proliferation in the WT suggesting the RhoJ expressing cells are much more proliferative, beyond the effect of apoptosis shown in other figures.

5, in Figure 4, the authors explore dna repair. The quantification of panel C seems to incongruent with the facs plots in terms of the numbers shown in the boxes versus plotted on the graph. Also, the patterns of dna damage between C and D appear somewhat different, is there anything to be learned about the nature of the dna damage in the cells from this discrepancy?

6, Figure 5 has some interesting images, but is not well labeled. Which is wt? ko? This interesting observation that warrants significant further exploration. Where is RhoJ located in the nucleus to regulate this? What is the pattern in Epcam+ vs Epcam-? what does the pattern mean for DNA repair? if one uses orthogonal methods to abrogate nuclear fibers, does that influence dna repair?

7, the authors express most of the results as Epcam+ vs Epcam- vs RhoJ-Epcam-. But if there are Epcam+ cells in RhoJ ko tumors, what is the effect of loss of RhoJ on these cells? These cells have been left out of all the quantifications. Since they started by showing that RhoJ was induced in Epcam- cells, perhaps there is no effect, but this would be a nice confirmation of that idea.

8, in figure 6, there is again discrepancy between the facs plots and the quantification, perhaps the reviewer just doesnt understand how this was carried out. Regardless, here the authors look at cell division and see no difference between genotypes in homeostatic conditions. If this is the case in tumors, then why was the overall phenotype so different? In vivo, is there significant genotoxic stress as has been setup in the experimental conditions here?

Author Rebuttals to Initial Comments:

We thank the referees for their fair and positive assessment of our manuscript.

We appreciate their thoughtful comments and suggestions, which help us to strengthen and improve our revised manuscript.

We have performed a lot of novel experiments to address their questions, comments, and suggestions.

As requested by the reviewers, we have

1. cited the suggested references
2. performed robust statistical analyses of each data set
3. quantified cell death upon sh-RNA deletion of RhoJ and RhoQ in Epcam+ tumor cells
4. analyzed proliferation upon RhoJ deletion in Epcam- cells and RhoJ overexpression in Epcam+ cells
5. performed qPCR analysis of EMT and epithelial marker in Epcam- cells upon deletion of RhoJ
6. analyzed RhoJ expression in Epcam+ and Epcam- tumors in a larger cohort of tumors *in vitro* and *in vivo*
7. analyzed the response to different chemotherapy in human cell line
8. provided complete lists of differential genes or selected proteins transcriptome and mass spectrometry data. Raw datasets are now accessible online.
9. validated some of our candidate genes and proteins differentially regulated by RhoJ KD
10. described the types of nuclear actin filament found in the different conditions
11. performed IP/MS to identify proteins that interact with RhoJ, confirmed by further IP/western blot
12. studied cell survival upon inhibition of nuclear actin polymerization and chemotherapy treatment
13. analyzed the activation of new origins of replication following chemotherapy combined with nuclear actin polymerization inhibitor in EMT tumor cells
14. quantified the formation of RPA2 foci upon inhibition of nuclear actin polymerization and chemotherapy treatment in EMT tumor cells
15. quantified micronuclei formation upon chemotherapy treatment in the different tumor cell populations
16. studied the response to a combination of ATR/ATM inhibitors and chemotherapy treatment
17. assessed intra-S checkpoint activation in response to chemotherapy by comparing the cell cycle profile in Epcam+, Epcam- and Epcam- RhoJ KD tumor cells
18. studied the long-term response to Cisplatin/5FU treatment of tumor-bearing mice (Control and RhoJ KO).
19. studied the activation of different CDKs by western blot
20. assessed the fork asymmetry upon chemotherapy
21. studied EdU incorporation upon chemotherapy treatment
22. assessed the role of RhoJ in cell migration
23. performed PCR Validation of RhoJ deletion in tumors from Lgr5CreER/KRAS^{G12D}/p53^{cKO}/RhoJ^{cKO}/Rosa-YFP mice.

24. performed Gene Ontology analysis of genes and proteins differentially regulated upon RhoJ deletion

Referees' comments:

Referee #1 (Remarks to the Author):

This is an interesting paper that describes a role for the small GTPase RhoJ in resistance to chemotherapy. Previous work has shown that RhoJ promotes resistance to chemotherapy in melanoma cells by targeting ATM signalling and so reducing apoptosis in response to DNA damage. Here the authors discover what seems to be an additional pathway in squamous cell carcinoma, whereby RhoJ upregulation controls additional elements of the DNA damage response. The findings are important; however, some of the key findings need to be put on a firmer footing.

We thank the reviewer for his/her positive comments and important suggestions.

Specific points

In giving the background to what is known about RhoJ and cancer, the authors should reference the work of Chen et al. (Front Cell Dev Biol 8:832).

We thank the reviewer for pointing out this interesting paper that could be relevant for our study as TGF- β is promoting EMT in multiple cancers. We have now cited this paper in the background about RhoJ as suggested by the reviewer.

The Methods state that the statistical analyses are described in the Figure Legends, but the Figure legends give only partial information; often just stating the post-test used. Some of the statistical analysis is inappropriate; for example, the use of a t-test for the multiple comparisons in Figure 2o. That Figure legend also describes having n=12 replicates from 2 independent experiments, which sounds like an n=2 experiment is being analysed like an n=12 experiment. The authors need to give a full account of how all of the statistical analysis was done. It is possible that this will reveal the need for additional experimental work.

We agree with the reviewer comment. We are sorry for not being clear in the presentation of our biological replicates and statistical analysis we had initially performed. We have now collaborated with a statistician and used appropriate tests for each analysis described below. Differences in continuous variables were compared between groups with non-parametric tests in case of small sample sizes ($n < \sim 30$). Those were Mann-Whitney tests in case of two groups and Kruskal-Wallis followed by Mann-Whitney tests corrected for multiple comparisons with Bonferroni corrections in case of more than two groups. When sample sizes were sufficiently large ($n > \sim 30$), parametric tests were used. Those were classical Student t-tests or Welch's t-tests in case of variance inequality in case of two samples and ANOVA followed by Sidak tests or Dunnett T3 tests in case of variances heterogeneity when more than two samples had

to be compared. Statistical significance was considered when p was <0.05 . All statistical tests were two-sided.

Figure 1. This is a small point, but the graph in Figure 1b would be better as a histogram and one that shows the individual tumor sizes as datapoints. We don't really care about the average tumor size of these arbitrary groupings, so the s.e.m. is meaningless, but it would be useful to see the range of sizes in each group.

We are sorry for not being clear in the presentation of our results. We have now added in Figure 1b histograms representing the evolution of tumor size of treated primary SCC. As suggested by the reviewer, we showed the Mean \pm SD as well as individual tumor sizes in response to 4 weeks of Cisplatin/5FU relative to the size at which the treatment started which allows to clearly highlight the difference in tumor response upon chemotherapy treatment. To add clarity, histograms have been separated according to the three categories not responding ($n=18$), partially responding ($n=30$) and fully responding ($n=8$) tumors.

As requested by the reviewer we also represented the range sizes in each group by performing histograms, enclosed, representing the Mean \pm SD and individual tumor sizes at the start of the application of the treatment and at the end of the treatment. These two representations of the data lead to the same conclusion and we think that the first is clearer. If the reviewer thinks these data add information to the current paper, we can easily add them in the supplemental information.

Figure 2. The authors perform RNAseq to identify genes that change in SSC cells that have undergone EMT. The raw data need to be presented as a Supplementary file.

We agree with the reviewer. As suggested by the reviewer, we have now made the raw sequencing data accessible as download file from GEO with the following accession number GEO: GSE205985. In addition, we have added in the supplementary file an excel file with the normalized dataset (Sourcedata_Fig2a_Fig3ab_RNAseq).

Figure 2. The data in Figures 2a and 2b are critical to the paper. These two figures are the only evidence for increased RhoJ expression in the SSC cells that have undergone EMT. Figure 1a shows the average of RNAseq measurements of RhoJ mRNA from two independent samples of sorted cells. Figure 1b shows (I think) n=5 qPCR measurements from either 5 independent sets of sorted cells, or 5 measurements on replicates of the same cells – this isn't clear. In either case, there isn't enough work to be able to accurately estimate the frequency or degree of RhoJ upregulation in SSC. This is the foundational observation of the paper, and the authors need a much bigger dataset and more detailed analysis.

We agree with the reviewer comment. As suggested by the reviewer, we have now analyzed RhoJ expression in a much larger cohort of tumors that have undergone EMT. Using qRT-PCR, we confirmed that RhoJ transcripts are enriched in FACS isolated Epcam- tumor cells from 10 independent SCCs as compared to Epcam+ tumor cells FACS isolated from 7 independent tumors. The higher expression of RhoJ *in vitro* in Epcam- EMT tumor cells was found across 13 independent primary culture established from FACS isolated Epcam- tumor cells from different tumors compared to Epcam+ tumor cells isolated from 11 independent tumors.

These data are now shown in the revised manuscript (Fig. 2a and 2b).

In Figure 2d, the authors report differences in cell number in isolated cancer cells depleted of RhoJ. The data do not reach statistical significance and the authors need to either increase the power or change how they describe this result. The same is true for Figure 2f.

We agree with the referee. We increased the power by increasing the sample size and therefore we used appropriate statistical analysis (Mann-Whitney U test). We confirmed that RhoJ knockdown in Epcam- tumor cells leads to a significant decrease in living tumor cells 48 hours post-chemotherapy (n=5). Likewise, RhoJ overexpression in Epcam+ tumor cells significantly increases cell survival 48 hours after chemotherapy (n=6).

These data are now shown in the revised manuscript (Fig. 2d and 2f).

Figure 2n, o. The authors look briefly at RhoJ MDA-MB-231 breast cancer cells. They do not show if MDA-MB-231 cells upregulate RhoJ, or if RhoJ levels change with EMT in breast cancer. The effects of silencing RhoJ are slight and there are issues with the statistical analysis (see first point). I don't think these panels add anything to the paper and they should be removed.

We agree that the data on MDA-MB-231 were not extensively developed in the initial submission. As suggested, by the reviewer, we have now removed these data from the revised manuscript.

Figure 3. The authors use RNAseq to look for genes regulated by RhoJ in the cells undergoing EMT. They also use iPOND to look for changes in protein recruitment to replication forks. These screens produce a lot of hits, and these are detailed in the text. But there is no validation of any of the candidates. I return to this point later. The authors need to provide the raw data for each screen in the Supplementary material.

We thank the reviewer for this interesting suggestion. We have now provided all the raw data of our transcriptional and proteomic screens. The accession number for the RNA sequencing is GEO: GSE205985. The mass spectrometry proteomics data have been deposited to the ProteomeXchange Consortium via the PRIDE partner repository with the dataset identifier PXD025737. (login username: reviewer_pxd025737@ebi.ac.uk and password: GKfBF90U). In addition, we have added in the supplementary file an excel file with the normalized dataset for the RNA sequencing and an excel file with the selected proteomic hits identified with the proteomic screens.

As suggested by the reviewers, we have validated some of our candidate genes and proteins differentially regulated by RhoJ KD. Using western blot, we confirmed the increase expression of the actin regulator N-WASP, replication machinery components POLD, PCNA and DNA repair activating phosphorylation of RPA2 as well as higher expression of several MCM subunits on chromatin in Epcam- cells compared to Epcam+ and RhoJ KO cells.

These data are now shown in the revised manuscript (Fig. 3d-g and Fig. 5i).

Figure 4d. The authors need to revise their interpretation a little. The overall results back up the hypothesis, but it isn't as straightforward as described in the text. For example, the recruitment of RAD51 goes down with RhoJ overexpression at 12h. The experimental work is fine, but the description needs to match the data better.

We are sorry for being unclear. We have now revised our interpretation concerning RAD51 foci formation and detailed our observations as follows: While RAD51 nuclear foci formation was induced in all cell types upon Cisplatin treatment, several differences were observed. 12 hours after Cisplatin administration, the number of RAD51 foci was higher in Epcam+ cells. At 24 hours, the number of RAD51 nuclear foci was lower in Epcam- tumor cells as compared to Epcam+ and Epcam- RhoJ KD cells (Fig. 4d). These results are consistent with an increase of DNA repair and a decrease of replicative stress in the presence of RhoJ in Epcam- tumor cells, in good accordance with the γ -H2AX data supporting a faster DNA repair in Epcam- tumor cells in a RhoJ dependent manner.

We have now corrected this sentence in the text.

Figure 5. The quantified effects of RhoJ on nuclear actin are interesting, but the images don't need to be in the main Figures. And it would be better if they included representative images from the key conditions.

We thank the reviewer for the suggestions. As suggested by the reviewer, we have now shown the representative images of the nuclear actin from the key conditions in Figure 6d.

Figure 6. The final Figure examines how RhoJ affects cell cycle arrest and replication after DNA damage. The SSC cells that have undergone EMT are more resistant to replication arrest and this depends on RhoJ. This is an interesting finding, but it leads to the biggest issue with the manuscript, which is that we don't really know how RhoJ is acting here. The authors look in a number of places – we know that RhoJ changes expression of DNA repair enzymes, that it changes recruitment of proteins to replication forks, that it can regulate nuclear actin and that it can help maintain replication after DNA damage. Some of these readouts might be connected, but the problem is that we are left with a number of partial possible solutions to

how RhoJ is acting, but no clear answer. For publication at this level, the authors need to work out how RhoJ is doing this.

We thank the reviewer for this interesting question. As suggested by the reviewer, we have now performed many new experiments to decipher the mechanisms by which RhoJ mediates the resistance to chemotherapy in EMT tumor cells. To understand how RhoJ regulates chemoresistance, we sought to identify the proteins interacting with RhoJ. To this end, we performed affinity-purification followed by mass spectrometry (AP-MS) using HA pull-down of Epcam- cells expressing 3HA-tagged RhoJ in untreated conditions and 12 hours after Cisplatin/5FU administration. We identified 118 proteins that were significantly enriched in untreated HA-tagged RhoJ expressing cells and 98 proteins that were significantly enriched in treated HA-tagged RhoJ expressing cells. Among them we found previously identified RhoJ interacting proteins such as GPRC5A (Bagci et al., Nat Cell Biol, 2020). Interestingly we identified, upon chemotherapy treatment, an enrichment of protein implicated in the regulation of actin filament dynamics (FLNB and TLN1) and IPO9, a protein that regulates nuclear import of actin (Dopie et al., PNAS, 2012) in RhoJ protein complex. KD of IPO9 has been shown to decrease nuclear actin filament and impairs DNA repair (Belin et al., elife 2015). These findings suggest that RhoJ physically interacts with actin regulator IPO9. To confirm this possibility, we performed co-immunoprecipitation using antibodies against endogenous IPO9 protein, followed by western blot revealing the presence of RhoJ-3HA protein. This Co-IP experiment confirmed that IPO9 is found in a protein complex with RhoJ (Fig. 6b), further suggesting that RhoJ regulates the response to chemotherapy through the regulation of nuclear actin, which have previously been shown to control DNA repair (Andrin et al., Nucleus, 2012; Belin et al., Elife, 2015; Caridi et al, Nature, 2018; Caridi et al., Nat Cell Biol, 2019; Schrank et al., Nature 2018; Hurst et al., Trends Cell Biol, 2019; Hyrskyluoto et al., Curr Opin Cell Biol, 2020; Lamm et al., Nat Cell Biol, 2020). We have then assessed whether inhibition of actin polymerization can regulate cell death following chemotherapy. We found that actin polymerization regulates cell survival upon chemotherapy in a RhoJ dependent manner (Fig. 6f). Finally, we have showed that actin polymerization regulates the DNA repair capacity and the activation of new origins of replication following chemotherapy in EMT tumor cells in a RhoJ dependent manner (Fig.6g-h). These data now comprehensively connect all the previous and novel data together and identify a clear mechanism by which RhoJ regulates resistance to chemotherapy in EMT tumor cells.

We have now presented and discussed these new data in the revised manuscript.

In summary, this is an interesting and potentially important paper, but the key findings are not fully resolved or fully supported. It is important that the authors properly determine the frequency and extent of RhoJ overexpression in SSC. And it is also critical that the authors find an explanation for the role of RhoJ in chemoresistance. Currently they present a number of possibilities, but none of these is taken to a conclusive answer.

We thank the reviewer for this important question of how the different RhoJ functions are connected to promote chemoresistance. As mentioned in the previous point, we have now performed many new experiments to decipher the mechanisms by which RhoJ mediates the resistance to chemotherapy in EMT tumor cells. To understand how RhoJ regulates chemoresistance, we sought to identify the proteins interacting with RhoJ. To this end, we performed affinity-purification followed by mass spectrometry (AP-MS) using HA pull-down

of Epcam- cells expressing 3HA-tagged RhoJ in untreated conditions and 12 hours after Cisplatin/5FU administration. We identified 118 proteins that were significantly enriched in untreated HA-tagged RhoJ expressing cells and 98 proteins that were significantly enriched in treated HA-tagged RhoJ expressing cells. Among them we found previously identified RhoJ interacting proteins such as GPRC5A (Bagci et al., Nat Cell Biol, 2020). Interestingly we identified, upon chemotherapy treatment, an enrichment of protein implicated in the regulation of actin filament dynamics (FLNB and TLN1) and IPO9, a protein that regulates nuclear import of actin (Dopie et al., PNAS, 2012). These findings suggest that RhoJ physically interacts with actin regulator IPO9. To confirm this possibility, we performed co-immunoprecipitation using antibodies against endogenous IPO9, followed by western blot revealing the presence of RhoJ-3HA protein. These Co-IP experiments confirmed that IPO9 is indeed found in a protein complex with RhoJ (Fig. 6b), further suggesting that RhoJ regulates response to chemotherapy through the regulation of nuclear actin, which have previously been shown to control DNA repair (Andrin et al., Nucleus, 2012; Belin et al., Elife, 2015; Caridi et al, Nature, 2018; Caridi et al., Nat Cell Biol, 2019; Schrank et al., Nature 2018; Hurst et al., Trends Cell Biol, 2019; Hyrskyluoto et al., Curr Opin Cell Biol, 2020; Lamm et al., Nat Cell Biol, 2020). We have tested whether the regulation of nuclear actin by RhoJ regulates DNA replication and DNA repair. Using actin polymerization inhibitor, we found that inhibiting actin decreased the survival of Epcam- tumor cells but not the Epcam+ and RhoJ KD Epcam- (Fig 6f), showing that actin polymerization acts downstream of RhoJ in regulating cell survival following chemotherapy. We also found that inhibition of actin polymerization inhibited the activation of new origin of DNA replication and the DNA repair capacity of Epcam- tumor cells in a RhoJ dependent manner (Fig 6g and 6h). These new data now allow to connect all the data together and identify a clear mechanism by which RhoJ regulates resistance to chemotherapy in EMT tumor cells.

These new data are presented and discussed in the revised manuscript (Fig 6-P18-21).

Referee #2 (Remarks to the Author):

In their manuscript, Debaugnies et al. identify and characterize the small Rho GTPase RhoJ as a potential driver of chemotherapy resistance in cancer cells that have undergone Epithelial-to-Mesenchymal Transition (EMT) using a skin squamous cell carcinoma (SCC) mouse model. The authors first show that RhoJ is overexpressed in EMT SCC cells that are resistant to standard SCC chemotherapy and that loss of RhoJ re-sensitizes these cells to treatment. They then try to dissect the molecular mechanism underlying RhoJ-mediated chemotherapy resistance, concluding that RhoJ upregulates DNA damage response pathways and confers resistance to replication stress.

While several groups have shown previously that GTPases of the Rho family, including RhoJ, are connected to the EMT process in various tumor types, Debaugnies et al. demonstrate that RhoJ expression status is specifically linked to therapy resistance in cells that have undergone EMT, at least in their mouse SCC model and also in a human breast cancer cell line. While this finding has important implications for tumor therapy, particularly with regards to patient stratification and prognosis prior to therapy of metastasized cancers, its novelty is somewhat dampened by a previous report, which showed that RhoJ is overexpressed in metastatic melanoma (DOI: 10.1158/0008-5472.CAN-12-0775; the authors briefly discuss this paper in the text). As to the link between RhoJ overexpression and a stunted replication stress response, the evidence provided in the manuscript remains superficial, and the authors do not provide a convincing explanation of how RhoJ promotes replication stress tolerance on a molecular level. Overall, I feel that the manuscript, while conceptually intriguing, contains limited new insights into the roles of RhoJ in either EMT or the DNA damage response.

We thank the reviewer saying that the manuscript is conceptually intriguing. We have now performed a lot of new experiments to provide new mechanistic insights into the mechanisms by which RhoJ controls cell survival and DNA damage response following chemotherapy in EMT tumor cells.

Major points/ suggestions:

My main concerns all center around the mechanism of the replication stress tolerance in Epcam- cells:

- The conclusion that Epcam- cells tolerate higher levels of replication stress is mostly based on the finding that origin firing is increased in these cells. However, is this accompanied by higher levels of fork breakage or the formation of micronuclei, or do the cells only display fork stalling? This is important for the model. The nature of the replication stress needs to be better characterized.

We thank the reviewer for these interesting questions.

We agree with the reviewer that our claim that Epcam⁻ cells tolerate higher levels of replication stress was partially based on the finding that origin firing is increased in these cells (Fig. 5h). We have performed several other experiments to reinforce this point. To test whether the higher resistance of Epcam⁻ cells tumor cells to chemotherapy is related to the higher ability to cope with replicative stress, we evaluated the sensitivity of Epcam⁺, Epcam⁻ and RhoJ KO tumor cells to the replicative stress-inducing drug aphidicolin. Aphidicolin is a selective inhibitor of DNA polymerases that induces fork stalling and ssDNA exposure by uncoupling polymerase and helicase activities. Aphidicolin induced a major increase of Epcam⁺ cell death. In contrast, Epcam⁻ tumor cells were relatively insensitive to aphidicolin treatment, whereas RhoJ KO Epcam⁻ cells were much more sensitive and presented a strong increase in apoptosis as compared to Epcam⁻ cells (Fig. 5c). In case of insufficient protection of stalled forks during replicative stress, the nascent DNA will be degraded by different nucleases (f.e. DNA2, MRE11) leading to fork collapse. Treatment with mirin, a MRE11 inhibitor, following Cisplatin/5FU administration decreased cell death in Epcam⁺ and Epcam⁻ RhoJ KO cells but had no effect on Epcam⁻ tumor cells (Fig. 5d), reinforcing the notion that RhoJ decreases replicative stress following chemotherapy in Epcam⁻ cells.

To understand the nature of the replicative stress, we assessed the level of fork stalling, fork breakage and the formation of micronuclei in Epcam⁺, Epcam⁻ and Epcam⁻ RhoJ KO cells treated with Cisplatin/5FU. To evaluate the level of fork stalling, fork asymmetry measurement was performed in Epcam⁺, Epcam⁻ and Epcam⁻ RhoJ KO cells untreated and treated with Cisplatin/5FU for 12 hours. Asymmetry of DNA replication fork progression was increased in all cell types in response to chemotherapy reflecting fork stalling (Extended data Fig. 6e-f). To evaluate the level of fork collapse, we investigated the level of histone 2A.X phosphorylated at S139 (γ -H2AX), a mark of DNA double strand breaks following Cisplatin/5FU treatment. Western blot, immunofluorescence and FACS quantification showed that Epcam⁻ tumor cells presented a significant reduction in γ -H2AX as compared to Epcam⁺ and Epcam⁻ RhoJ KO tumor cells after Cisplatin/5FU administration (Fig. 4b-c), suggesting higher level of fork breakage in the absence of RhoJ. An increase in the number of micronuclei was found in Epcam⁺ and RhoJ KO Epcam⁻ cells compared to Epcam⁻ cells 12 hours and 24 hours after chemotherapy (Extended Data Fig. 6a-b) indicating unresolved replicative stress in the absence of RhoJ. Altogether these data indicate that Epcam⁻ cells tolerate higher levels of replication stress.

- In the same vein, the authors argue that “Epcam⁻ resistance to therapy is independent of ATR-mediated checkpoint” signalling. However, have they excluded that ATR expression or activation is downregulated in Epcam⁻ cells? The lower levels of RPA focus formation could be explained by a lack of ATR. Is the intra-S checkpoint activated at all?

We thank the referee for these interesting questions. To understand whether ATR was activated in Epcam⁻ cells in response to chemotherapy, we compared the level of phosphorylated substrates of ATM and ATR induced upon chemotherapy treatment combined with ATR inhibitor, ATM inhibitor or a combination of both inhibitors. Upon chemotherapy, Epcam⁺, Epcam⁻, and Epcam⁻ RhoJ KD tumor cells showed a major increase in the phosphorylation of ATM/ATR substrates in the different cell populations (Extended data Fig. 6c). When chemotherapy was combined with ATR and/or ATM inhibitor, there was a

decrease in the level of ATM/ATR substrate phosphorylation, and when both ATM and ATR inhibitors were administered together, there was almost complete inhibition of phosphorylation, showing that upon DNA damage, ATR and ATM are both activated in Epcam⁺ and Epcam⁻ cells, and RhoJ KD does not regulate ATM and ATR activation in response to chemotherapy (Extended data Fig. 6c). Treating Epcam⁺ and RhoJ KO Epcam⁻ cells with a combination of chemotherapy and ATR inhibitor increased cell death suggesting that Epcam⁺ and RhoJ KO Epcam⁻ cells rely on ATR during replicative stress (Fig 5e). In contrast, treating Epcam⁻ cells with a combination of chemotherapy and ATR inhibitor did not increase cell death indicating that Epcam⁻ resistance to therapy is independent of ATR-mediated checkpoint.

To determine whether intra-S checkpoint activation occurs in response to chemotherapy, we compare the cell cycle profile in Epcam⁺, Epcam⁻ and Epcam⁻ RhoJ KD tumor cells in response to chemotherapy using BrdU/7AAD FACS analysis. At 12 hours following Cisplatin/5FU treatment, the proportion of BrdU positive cells increased in all conditions with a predominant accumulation of the cells in early-S stages in Epcam⁻ cells suggesting that the progression through S- phase was dampened and intra-S phase checkpoint activated (Fig. 5a-b). Engagement of intra- S checkpoint was further suggested by the global slowing of replication forks induced with Cisplatin/5FU treatment in all cell types as determined by performing DNA fiber analysis (Fig. 5g).

- And what would be the nature of an ATR-independent resistance mechanism? Is lesion bypass upregulated? Or if there is increased DNA breakage (as suggested by elevated 53BP1 levels), how do cells deal with these lesions and avoid cell death? Increased efficiency of DSB repair? Is the ATM checkpoint active?

We thank the referee for these interesting questions.

To understand whether ATM was activated in Epcam⁻ cells in response to chemotherapy, we compared the level of phosphorylated substrates of ATM and ATR induced upon chemotherapy treatment combined with ATR inhibitor, ATM inhibitor or a combination of both inhibitors. Upon chemotherapy, Epcam⁺, Epcam⁻, and Epcam⁻ RhoJ KD tumor cells showed a major increase in the phosphorylation of ATM/ATR substrates in the different cell populations. When chemotherapy was combined with ATR and/or ATM inhibitor, there was a decrease in the level of phosphorylated ATM/ATR substrates, and when both ATM and ATR substrates was administered together, there was almost complete inhibition of phosphorylation, showing that upon DNA damage, ATR and ATM are both activated in Epcam⁺ and Epcam⁻ cells, and RhoJ KD does not regulate ATM and ATR activation in response to chemotherapy (Extended data Fig. 6c). Treating Epcam⁻, Epcam⁺ and RhoJ KO Epcam⁻ cells with a combination of chemotherapy and ATM inhibitor had no significant effect indicating a minor impact of ATM activation in chemotherapy response in our model (Extended data Fig. 6d).

To understand how the cells deal with replicative stress-induced lesions, we measured fork asymmetry in Epcam⁺, Epcam⁻ and Epcam⁻ RhoJ KO cells untreated and treated with Cisplatin/5FU for 12 hours. Asymmetry of DNA replication fork progression was increased in

all cell types in response to chemotherapy reflecting fork stalling. Interestingly, under unchallenged conditions, the basal level of asymmetry in Epcam⁻ cells was already increased compared to Epcam⁺ and RhoJ KO Epcam⁻ cells which could in part explain why Epcam⁻ cells present increased tolerance to DNA damage (Extended data Fig. 6e-f).

These new data are presented and discussed in the revised manuscript (Extended Data Fig. 6-P17-18).

- The Te Riele lab has shown that inactivation of p53 can suppress replication stress-induced DNA damage (DOI: 10.7554/eLife.37868). The authors need to rule out lack of p53, in the context of RhoJ overexpression, has a similar effect in their SCC model. Why did the authors look at CDK1 activation but not CDK2, the latter of which is an important component of the intra-S-checkpoint?

We thank the reviewer to draw our attention to this paper. The genetic mouse model used in our study to induce SCC presents p53 deletion (See controls in our original paper Lapouge et al. PNAS 2011 and Latil et al. Cell Stem Cell 2017). Therefore, all the cell types studied, Epcam⁺, Epcam⁻ and RhoJ KO cells lack p53. However only Epcam⁻ cells displays increased levels of origin firing and reduced DSB formation suggesting a role of RhoJ in suppressing replication-stress-induced DNA breakage. We agree with the reviewer that the cell specific response of Epcam⁻ cells in the presence of RhoJ expression is reminiscent to the DDR described in mouse embryonic fibroblasts KO for p53 and overexpressing Bcl2 by the Te Riele lab. We have now referred to this paper in our discussion.

We agree with the reviewer that CDK2 is an important component of the intra-S-checkpoint. We initially investigated CDK1 posttranslational modifications as CDK1 plays an essential role in S-phase progression including in the regulation of origin firing programs (Berthet et al., Cell Div , 2006; Hochegger et al., J Cell Biol, 2007; Nakanishi et al., Chromosome Res, 2010; Santamaria et al., Nature, 2007). As requested by this and other reviewers, we assessed whether the regulation of cell cycle induced by chemotherapy was associated with modulation in CDK1 and CDK2 activity in Epcam⁺, Epcam⁻ and Epcam⁻ RhoJ KD tumor cells (Extended Data Fig. 5c-d). Using western blot, we studied the level of the inhibitory CDK1 phosphorylation on residue Y15, the activating CDK1 phosphorylation on residue T161 and the activating CDK2 phosphorylation on residue T160. CDK1 phosphorylation on residue Y15 was transient in Epcam⁻ tumor cells peaking at 12 hours and decreasing thereafter, while it was sustained in Epcam⁺ and Epcam⁻ RhoJ KO tumor cells. CDK1 phosphorylation on residue T161 was induced upon chemotherapy treatment in all cell types but appeared earlier in Epcam⁻ cells and was already detected at 12h post chemotherapy suggesting that RhoJ negatively regulates CDK1 inhibition and allows S-phase progression and mitosis following chemotherapy in Epcam⁻ cells. CDK2 phosphorylation on residue T160 was induced in the same way in all cell types upon chemotherapy treatment allowing S-phase entry.

These new data are presented and discussed in the revised manuscript (Extended Data Fig. 5c-d-P15-16).

• Figure 5: this figure is totally out of context. I did not understand how it relates to either the preceding or the following figure. In addition, it is poorly labelled and annotated. How does nuclear actin polymerization promote RhoJ-mediated replication stress tolerance? While the authors speculate about this point in the discussion, the impact of the paper would be much higher if they could experimentally connect RhoJ expression levels and changes in actin polymerization to the activity of components of the replication stress machinery in their model system. Do Arp2/3 localize to stalled or collapsed replication forks in Epcam- cells? Is their activity important for origin firing in Epcam- cells? Does inhibition of nuclear actin polymerization re-sensitize Epcam- cells to chemotherapy?

We apologize for the poorly labelled and annotated figure. We have now reformatted the figure showing representative images of the different nuclear actin filament patterns in Epcam+, Epcam- and RhoJ KO Epcam- cells in control and treated conditions (Fig. 6d-e).

We thank the referee for this very interesting question. As suggested by the reviewer as well as the other referees, we investigated whether nuclear actin polymerization promotes RhoJ-mediated replicative stress tolerance. To this end, we assessed the effects of the inhibition of nuclear actin polymerization on replicative stress response in Epcam- cells, Epcam+ cells and Epcam- RhoJ KO cells. Combination of Cisplatin/5FU treatment with F-actin inhibitor (Latrunculin B), significantly increased apoptosis in Epcam- cells (Fig. 6f). Latrunculin B treatment also resulted in the reduction of the recruitment of new origin of replication following chemotherapy in Epcam- cells (Fig. 6g). Finally, inhibition of actin polymerization increased the formation of RPA2 foci (Fig. 6h) in Epcam- cells following chemotherapy suggesting that RhoJ-mediated replication stress tolerance is dependent on nuclear actin polymerization. To define which subset of the F-actin networks was relevant for the increased resistance of RhoJ expressing cells to chemotherapy, we treated the cells with different inhibitors: ARP2/3 inhibitor (CK666) that prevents the formation of F-actin branched-network, N-Wasp inhibitor (Wiskostatin) which prevents the activation of ARP2/3 complex and formin inhibitor (SMIFH2) which preferentially inhibits the formation of F-actin bundles. Among the three inhibitors, only SMIFH2 and Cisplatin/5FU increased the level of cell death in Epcam- cells similarly to the combination of Cisplatin/5FU with Latrunculin B (Fig. 6f and Extended Data Fig.7c) suggesting that RhoJ-mediated nuclear actin polymerization in response to replicative stress response was dependent on formin. Treating Epcam- cells with Latrunculin B or SMIFH2 alone in the absence of chemotherapy did not induce cell death suggesting a specific role of nuclear actin polymerization in response to replicative stress (Extended Data Fig. 7d).

These new data are presented and discussed in the revised manuscript (Fig. 6 and Extended Data Fig. 7-P19-21).

Other points:

• Figure 11: The resistance of Epcam- cells to Doxorubicin, Paclitaxel and IR appears to be minor compared to Cisplatin/5FU or Gemcitabine. This difference should be commented on in the text. The text discussing this figure is misleading in its current form.

We agree with the reviewer that the resistance of Epcam- cells to Doxorubicin, Paclitaxel and IR are less important compared to Cisplatin/5FU or Gemcitabine. We have corrected our text

as follows: Interestingly, Epcam- EMT tumor cells were resistant to topoisomerase I inhibitor topotecan, topoisomerase II inhibitor etoposide, anti-metabolite gemcitabine. Epcam- tumor cells were also resistant to anti-microtubule paclitaxel, DNA intercalant and topoisomerase II inhibitor doxorubicin as well as ionizing radiation, although these treatments were less efficient in inducing cell death in Epcam+ tumor cells (Fig. 1i).

- Figure 2D: Epcam+ control is missing. No statistical analysis.

We thank the reviewer to point this mistake, we have now included Epcam+ control and ad hoc statistical analysis. As requested by several referees, we assessed whether sh-RNA mediated RhoJ and RhoQ knockdown (KD) sensitized Epcam+ tumor cells to chemotherapy. The proportion of active caspase-3 positive cells 24 hours following Cisplatin/5FU was similar in Epcam+ control tumor cells compared to RhoJ KD Epcam+ cells and RhoQ KD Epcam+ cells which is consistent with the low expression of RhoJ in Epcam+ cells compared to Epcam- cells (Extended Data Fig. 2e-f).

- Figure 2C: Epcam+ control is missing.

We thank the reviewer for pointing out this mistake, we have now included Epcam+ control (Extended data Fig. 2e-f)

- Figure 2F: No statistical analysis.

We thank the reviewer for pointing out this mistake. We have now added appropriate statistical analysis.

- Page 5: “In the absence of chemotherapy, we found that the majority of tumors (62%) were mixed tumors presenting epithelial (Epcam+) and mesenchymal (Epcam-) tumor cells, 29% were fully mesenchymal tumors with more than 90% of Epcam- tumor cells, and 9% were well-differentiated epithelial tumors with more than 90% of Epcam+ tumor cells.” – Which figure do these numbers refer to? This part of the paper was very confusing to read and hard to understand.

We apologize for being unclear. These numbers refer to the figure 1e which analyzes the distribution of Epcam marker in YFP+ tumor cells from untreated and treated SCC. To add clarity to the manuscript we have now formulated our sentence as follows:

We monitored EMT by quantifying the proportion of Epcam+ cells by FACS analysis in untreated and treated tumors. In the absence of chemotherapy, we found that most tumors (62%) were mixed tumors presenting epithelial (Epcam+) and mesenchymal (Epcam-) tumor cells, 29% were fully mesenchymal tumors, and 9% were well-differentiated epithelial tumors.

- Figure 3C: The Epcam- cells transfected with control hairpins show a clear reduction in expression levels relative to Epcam- WT cells (particularly in the upper part of cluster 1). Wouldn't this suggest that the shRNA transduction alone already causes changes in gene expression?

We agree with the reviewer that a reduction in expression levels of some proteins is observed in Epcam- cells transfected with control hairpins compared to Epcam- WT cells. However, the differences in protein expression can be explained by the heterogeneity among the different cell lines.

The overall effect of RhoJ expression has been evaluated by performing a two-way ANOVA test to compare samples with and without RhoJ expression. Proteins with a p-value < 0.05 were considered to be significantly regulated. 99 proteins of interest were selected, the log2 transformed intensities of these proteins were Z-scored and these values were plotted in a heatmap after non-supervised hierarchical clustering.

- Figure 6C: total CDK1 is missing.

Total CDK1 has been added to the figure (Extended Data Fig. 5 d)

- Data and methodology: Overall, the scientific approach is valid. The methods section is reasonably detailed although it is not clear how DNA damage foci were quantified in ImageJ. Was this an automated analysis or was it done by hand? Have the authors taken into account different nucleus sizes? In Figure 4D, nuclear sizes appear quite variable – this would obviously affect the total number of foci per cell.

We thank the reviewer to find our scientific approach valid. To answer the question, the quantification of DNA damage foci in each nucleus was performed by hand using the manual cell counting plugin of ImageJ software by a blinded investigator. Quantification of the nuclear size showed no difference between Epcam+ and Epcam- cells. There is a small reduction in the nucleus size in RhoJ KO cells in the absence of treatment. However, this difference in the size of the nucleus disappeared following chemotherapy. In consequence the difference in nuclear size does not account for the difference in the number of foci observed following chemotherapy.

- Statistics: Mostly appropriate. However, there are several graphs that lack statistical analysis. In addition, is there a reason why the authors summarize their data in many figures with SEM rather than SD? Wouldn't they want to indicate the variability within their samples rather than the proximity to the population mean?

We agree with the reviewer who found our statistics mostly appropriate, and we have now completed the graphs with more appropriate statistical analysis and indicated the SD rather than the SEM, to indicate the variability within our samples, when it was required.

- Clarity and context: Overall, the manuscript is well structured. However, the authors tend to overstate but not explain certain important ideas. For example, what are the “implications for

the development of new strategies to overcome resistance to anticancer therapy” (abstract and last sentence of the discussion)? Given the difficulties to develop clinically relevant Rho inhibitors, this is a bold statement and, at the minimum, warrants further discussion in the text. Also, the statement that “cellular effectors of DDR are not often activated at the mRNA level” is simply not true and this is also as such not stated in the cited review by Polo and Jackson. Please amend. Readability of the manuscript could be further improved 1) by using fewer acronyms (do “tumor cells” really need to be abbreviated as TC?), 2) by making sure to define all acronyms (for example, what does “Epcam Co sh” mean?! I assume it refers to the control hairpin, but I shouldn’t have to guess) and 3) by removing the many typos.

We thank the reviewer to find our manuscript well structured. We apologize for the bold statement saying that our findings will have important implications for the development of new strategies to overcome resistance to anti-cancer therapy. We believe that the contribution of RhoJ in promoting EMT tumor cells survival and the comprehension of the mechanisms of therapy resistance mediated by nuclear actin remodeling and involving replicative stress response are very interesting and will stimulate further research to develop new strategies to overcome resistance to anticancer therapy. As suggested by the referee, we had removed the last sentence from the abstract.

We apologize for the wrong statement saying that “cellular effectors of DDR are not often activated at the mRNA level” and we removed it. We replaced it by “In addition to the transcriptional regulation of the cellular effectors of DNA damage response (DDR) that accompanied genotoxic stress induced by chemotherapy, DDR factors are activated at the protein level by the protein stability, phosphorylation, ubiquitination and other posttranslational modifications that control DNA repair and cell survival “

We agree with the reviewer to improve readability of the manuscript by removing the typos and some unnecessary acronyms. We replaced Epcam Co sh by Epcam- Control sh and defined in the Figure legends Epcam- control short hairpin (sh) RNA cells (Epcam- Control sh) and RhoJ depleted Epcam- cells (Epcam- RhoJ sh).

o Page 5: They should briefly explain that Vimentin is used as a marker for mesenchymal cells and K14 is a marker for epithelial cells to make it easier to understand the experimental set up and the connection to EMT.

To add clarity, as suggested by the reviewer, we have now explained that Vimentin is used as a marker for mesenchymal cells and K14 is a marker for epithelial cells

- References: Overall, comprehensive and correct. The authors might want to consider adding two recent publications that further link RhoJ expression to EMT (DOI: 10.1016/j.bbrc.2021.06.004 and DOI: 10.1007/s13311-020-00910-w).

We thank the reviewer for pointing out those two interesting papers that could be relevant for our study as RhoJ may have different impact on EMT depending on the origin of the tumor involved. We have now cited those two papers in the background about RhoJ as suggested by the reviewer.

Referee #3 (Remarks to the Author):

Debaugnies et al. aim to unravel mechanisms underlying the EMT-associated therapy resistance in cancer. A causal link of therapy resistance (to various modalities, including chemotherapy, radiotherapy and targeted therapy) and EMT is evident since many years. However the underlying molecular mechanisms are not well understood. Known resistance mechanisms include increased expression of drug efflux pumps and drug-metabolizing enzymes, as well as general inhibition of apoptotic pathways due to an EMT-associated stemness state. The authors used a genetic mouse model of skin squamous carcinoma (SCC), which is well established in the group, isolated an epithelial fraction of tumor cells (Epcam+ tumor cells), which is highly sensitive to chemotherapy (Cisplatin and 5-FU) and a partially mesenchymal fraction (Epcam- tumor cells), which is highly resistant. Using this pair of cell types they identified the small GTPase RhoJ to be preferentially expressed in Epcam- tumor cells. By further applying proteomics, transcriptomics, overexpression and depletion studies (both *in vitro* and *in vivo*) they demonstrate that RhoJ limits chemotherapy-induced tumor cell killing by promoting DNA damage response and the new formation of replication forks, thereby attenuating replicative stress and allowing cell cycle progression and proliferation under chemotherapy.

Since therapy resistance, often linked to an EMT-phenotype, is a major cause of cancer-associated death, the topic and the results are of high relevance and potential translational impact in cancer research and also related fields (cell biology and cell cycle regulation/DNA damage response, EMT in cancer). If validated in other systems and entities the results would be interesting for the development of novel therapeutic strategies and drug development against therapy resistance.

This is a well-presented study on a relevant topic by leading experts in the field. The outstanding result is the description of a factor (RhoJ), which is overexpressed in the EMT state and (partially) responsible for EMT-associated therapy resistance and thus could become a promising therapeutic target. The study is well performed including proper statistical analyses.

We thank the reviewer for his/her positive comments.

However, open questions, discrepancies and concerns, particularly to the underlying mechanisms of the described effects, remain to be addressed:

1. The major critics concerns data presented in Fig.5: Data, particularly shown in Fig. 6, support the claim of the authors that RhoJ promotes DNA damage response and the formation of replication forks, thereby attenuating replicative stress and allowing cell cycle progression. However, the molecular link of RhoJ to these effects is likely indirect. Here the recently published role of nuclear actin in cell cycle regulation, DNA damage response and spreading of replication forks (ref. 43) comes into play. The authors describe (Fig.5), a two fold increase

in nuclear actin filaments in Epcam- tumor cells, which can be inhibited by RhoJ KO. This is an interesting correlation, but deeper understanding if and how RhoJ is directly involved in nuclear actin (pattern) formation is lacking. Since general effects of RhoJ on DNA damage response are already described (e.g. ref.25,25) a deeper understanding of such a causal connection would strongly increase the impact and novelty of the paper. E.g.

what is the role of actin polymerising/remodelling factors described to be dependent on RhoJ expression (e.g. Arp2/3 and others, see fig. 3a, c, d).

In addition, since only 5-10% of cells have nuclear actin filaments, what about the 90-95% of the rest of cells?

Is the expression of nuclear actin as well as the formation of different patterns depending on the cell cycle state?

Is there a role/function of the different expression nuclear actin patterns (depending on RhoJ expression) on DNA damage response and therapy resistance (e.g. as described in Figs. 1 and 6)?

We thank the reviewer for all these important and interesting questions. We have addressed the different questions raised by the reviewers.

To assess the role/function of the different expression nuclear actin patterns on DNA damage response and therapy resistance, we quantified the different patterns of nuclear actin filament observed that can be classified in five categories. Upon chemotherapy treatment, we detected strong induction of Pattern 2 characterized by elongated actin filaments in Epcam- cells compared to the induction of Pattern 1 characterized by thin, short and branched actin filaments in Epcam+ cells. Pattern 1 and pattern 3, characterized by hairy filaments which are short filament with a dense and multipolar organization, were predominant in RhoJ KO cells independently of the presence of chemotherapy (Fig. 6d-e). When chemotherapy was combined with Latrunculin B or SMIFH2, the nuclear actin filaments became disorganized with a predominance of Pattern 4 characterized by thick and twisted actin and pattern 5 characterized by a severe disruption of actin filaments and was associated with increased cell death in Epcam- cells supporting the notion that the chemotherapy-induced long nuclear filament organization involves RhoJ and mediates resistance to chemotherapy (Fig. 6d-f)

The presence of the nuclear actin polymerization only in a subset of the cells has been commonly observed in the different studies that investigate nuclear actin in response to DNA damage (Belin et al., *Elife*, 2015; Hurst et al., *Trends Cell Biol*, 2019). The significance of this remains unclear. The fraction of transfected cells and the dynamic nature of the nuclear actin polymerization may account for our inability to detect nuclear actin in all cells.

To determine the cell cycle state in which the cells presenting nuclear actin filament are, we performed immunostaining of phospho-histone H3 to identify mitotic cells and found no cells expressing nuclear actin filament positive for phospho-histone H3 (Extended Data Fig. 7a). We tried to use BrdU or EdU to identify cells in S-phase. However, the steps of DNA denaturation necessary to stain BrdU or EdU destroyed the nuclear actin GFP signal. Therefore, we performed immunostaining of PCNA which marks sites of active DNA replication to identify cells in S-phase. It appears that a proportion of cells presenting nuclear actin filaments were PCNA positive (Extended Data Fig. 7b). Altogether those data suggest

that one part of the cells presenting nuclear actin filaments are in S-phase (PCNA+, pH3-) and the other part is in G0-G1-phase (PCNA-, pH3-) (Extended Data Fig. 7a-b).

We have tested whether the regulation of nuclear actin by RhoJ regulates DNA replication and DNA repair. Using actin polymerization inhibitor, we found that inhibiting actin, decreased the survival of Epcam- tumor cells but not the Epcam+ and RhoJ KD Epcam- (Fig 6f), showing that actin polymerization acts downstream of RhoJ in regulating cell survival following chemotherapy. We also found that inhibition of actin polymerization inhibited the activation of new origin of DNA replication and the DNA repair capacity of Epcam- tumor cells in a RhoJ dependent manner (Fig 6g and 6h).

We have now presented and discussed these new data in the revised manuscript (Fig. 6 and Extended Data Fig. 7 and P19-21).

2. The authors state that the proportion of Epcam+/Epcam- tumor cells was unchanged in WT and RhoJ KO tumors (Fig. 2i-j), showing that RhoJ does not control EMT *per se*. Although not being significant, there seems to be a loss of tumors with a low share of Epcam- tumor cells (Fig. 2j), indicating that more tumor cells underwent EMT in RhoJ KO tumors. This would indicate that RhoJ is inhibiting EMT, whereas in the other data RhoJ effects seem to be coupled to the (partial) EMT phenotype. This is contradictory. What is the effect of RhoJ on EMT eventually? In order to clarify this, the RhoJ KO tumors could be separated into groups (E-M-mixed) according to their differentiation state by histology and then the Epcam -/+ cells grouped accordingly (as in Figure 1g). Moreover, the influence of RhoJ overexpression (Fig. 2f) on EMT marker expression and migration/invasion should be addressed in tumor cells *in vitro* in order to strengthen the authors' claim. In summary, if RhoJ would have no effect on the EMT state, but is a crucial (and sufficient! as claimed from Fig. 2f) factor for therapy resistance, why do all resistant cells undergo an EMT (as shown e.g. in Fig. 1e)? This should also be discussed/explained.

We thank the referee for raising this interesting point. We have now added more data to understand whether RhoJ dependent function was related to EMT modulation. FACS quantification of the proportion of YFP+ Epcam+ and YFP+ Epcam- tumor cells was used to assess the degree of EMT in WT and RhoJ KO tumors. The majority of WT SCCs (50%, n=22) and RhoJ KO SCCs (59%, n=16) were mixed tumors composed of cells that have partially undergone EMT or were mesenchymal tumors composed of cells that have completely undergone EMT (45%, n=20 in WT tumors and 37%, n=10 in RhoJ KO tumors) made up of more than 90% of Epcam- tumor cells. A minority of WT tumors (5%, n=2) and RhoJ KO tumors (4%, n=1) were well-differentiated SCCs made up of more than 90% of Epcam+ cells showing that similar degree of EMT was observed in WT and RhoJ KO tumors, suggesting that RhoJ does not regulate EMT *per se* (Fig. 2i-j). Using RNA-seq and q-RT-PCR analysis we found that epithelial and mesenchymal markers including K14, Epcam, Vimentin, Zeb1, Prrx1 and Pdgfra were unchanged in RhoJ KO Epcam- cells and RhoJ overexpressing Epcam+ cells indicating that RhoJ does not regulate epithelial and mesenchymal gene expression (Extended Data Fig. 2d and 4f). We did not find either a significant modulation of

the protein level by MS of epithelial markers (ITGB6, EPCAM and ESRP1) or the mesenchymal markers (VIM and CDH2) upon RhoJ depletion in Epcam- cells (Extended Data Fig. 4g).

As suggested by the reviewer, we have investigated the role of RhoJ in migration. We found that RhoJ shRNA knockdown decreases cell migration *in vitro* in Epcam- tumor cells (Extended Data Fig. 2b-c) consistent with previous studies of endothelial cells, breast cancer and melanoma cell lines (Chen et al., *Front Cell Dev Biol*, 2020; Kaur et al., *Arterioscler Thromb Vasc Biol*, 2011; Ho et al., *Pigment Cell Melanoma Res*, 2013) that have demonstrated that RhoJ controls cell migration by altering cytoskeleton organization.

As discussed in the manuscript, we think that the reason for why most of resistant cells are Epcam- cells is due to the fact that chemotherapy preferentially kills Epcam+ tumor cells giving a selective advantage to Epcam- cells. Our data indicates that RhoJ acts downstream of EMT to accelerate DNA repair and cell survival following chemotherapy in EMT tumor cells but does not control the transcriptional program associated with EMT *per se*.

We have now presented and discussed these new data in the revised manuscript (Fig. 2i-j, Extended Data Fig. 2b-d and 4f-g and P6, 8-9, 11-12 and 22).

3. Fig. 2n,o: The relevance for human cancer has only been addressed in one cell line (MDA-MB-231). Here, concentrations of cyclophosphamide are very high (Molar range!), arguing for unspecific toxicity. In fact, cyclophosphamide is a pro-drug that should be metabolized *in vivo* by CytP450 (usually in the liver). The active, anti-proliferative and cytotoxic substances may eventually differ *in vitro* and *in vivo*. To circumvent these problems, the impact of RhoJ loss on sensitivity to other clinically relevant drugs (such as those used in Fig. 1i) should be used in more than one human cell line. In addition, it could be clarified, whether there is also a clinical impact of RhoJ expression in patients on survival and metastasis in non-melanoma patients.

We thank the reviewer for all these important questions.

As requested by the referee, we studied the effect of RhoJ in human cell line in response to different chemotherapy regimen. To assess the relevance of our findings to human cancers, we generated RhoJ knockdown using short hairpin RNA in MDA-MB-231, a human triple-negative breast cancer cell line presenting EMT which expresses high levels of RhoJ and presenting mutations in p53. RhoJ depletion in MDA-MB231 is associated with a decrease in proliferation. As suggested by the reviewer, we have extended the analysis of RhoJ-knockdown MDA-MB-231 to other clinically relevant drugs. We found that RhoJ-knockdown MDA-MB-231 cells were more sensitive to Cisplatin, Doxorubicin and paclitaxel and that this increase in cell death was accompanied by lower level of DNA repair as shown by the increased levels of γ -H2AX accumulation.

However, as Reviewer#1 suggested to remove these data, we have thus removed this data in the revised manuscript.

To assess whether the level of RhoJ expression was associated with clinical outcome in non-melanoma patients, we analyzed the correlation between RHOJ expression and overall survival in human cancer of different origins using RNA-seq dataset of The Cancer Genome Atlas (TCGA) and downloaded from the National Cancer Institute portal at <https://portal.gdc.cancer.gov>. Optimal-cutpoints were used to divide the two groups of low and high expressing cases represented in Kaplan-Meier survival curves below. Population numbers for low or high expressing cases are shown; $p = \text{log-rank Mantel-Cox test}$. The level of RHOJ expression correlated with poor overall survival in lung squamous cell carcinoma, esophageal carcinoma and cervical squamous cell carcinoma and adenocarcinoma. However, it was associated with increased overall survival in head and neck squamous cell carcinoma, uterine carcinosarcoma and skin cutaneous melanoma. Since the RNA-seq data originates from complete tumor samples, the expression of RhoJ may not be representative of the epithelial tumor cells populations and may be contaminated with other tumor cells

fractions including endothelial cells which are known to express high level of RhoJ (Kaur et al., *Arterioscler Thromb Vasc Biol* ,2011). Therefore, the correlation of the expression of RhoJ and the overall survival from the TCGA dataset could be misleading. If the reviewer thinks these data add information to the current paper, we can easily add them in the supplemental dataset.

4. The authors put lot of emphasis on chemoresistance in the manuscript and state that `Altogether these data demonstrate that RhoJ is both necessary and sufficient to confer resistance to chemotherapy in both mouse and human cancers presenting EMT`. However, the conclusion that RhoJ promotes resistance is not justified by the data provided, as only a short-term (initial) cell death response had been analyzed *in vitro* as well as *in vivo*. This conclusion should be weakened or tumor-bearing mice (Ctrls and RhoJ KO) should be treated for longer periods (e.g. 4 weeks, as in Fig. 1b) to demonstrate absence/shrinkage of the non-responding fraction of tumors.

We agree with the referee that only a short-term (initial) cell death response had been initially analyzed *in vitro* as well as *in vivo*. As requested by the referee, we have now studied the long-term response to Cisplatin/5FU treatment of tumor-bearing mice (Control and RhoJ KO). After three weeks of Cisplatin/5FU administration, the growth of the SCCs from mice transplanted with RhoJ knockdown Epcam- tumor cells was reduced compared to the growth

of the SCCs from RhoJ wild-type Epcam- cells transplanted mice, consistent with the difference in drug sensitivity observed in short term response to Cisplatin/5FU treatment (Fig. 2n).

We have now presented and discussed these new data in the revised manuscript (Fig 2n and P9-10).

5. The role of RhoJ in cell cycle progression and DDR remains unclear. Specific DNA damage responses are tightly interlinked with distinct cell cycle phases and progression under unperturbed conditions and under genotoxic stress, such as chemotherapy. For example, RAD51 and RPA foci formation are restricted to S and G2 phases of the cell cycle. However, the authors incompletely reconcile their DDR, fiber assay and cell cycle data. Several critical points and discrepancies have to be addressed:

a) The authors state that “No difference in cell cycle distribution was observed between Epcam+, Epcam- and RhoJ KO tumor cells in untreated condition.” However, the representative flow plots in Fig. 6a, show that RhoJ KO Epcam-negative tumor cells display more G1 and less G2 than EPCAM-negative tumor cells. Moreover, there seem to be less early and late S-phase cells in RhoJ KO Epcam-negative tumor cells than in both other cell types, implying that only mid-S phase cells (with highest replication rates) are unaffected by loss of RhoJ. Complete cell cycle profiles with statistical analyses (as in Fig. 6b) have to be provided to justify the abovementioned statement.

We thank the referee for this interesting remark. We apologized for the erroneous claim saying that “No difference in cell cycle distribution was observed between Epcam+, Epcam- and RhoJ KO tumor cells in untreated condition.” After analyzing more samples, we have now performed a complete analysis of cell cycle profile and we have made the required corrections to the manuscript. To determine whether the cell cycle is differentially regulated in Epcam+, Epcam- and Epcam- RhoJ KD tumor cells in response to chemotherapy, we assessed the DNA content and BrdU incorporation before and at different time points after Cisplatin/5FU administration using BrdU/7AAD FACS analysis. RhoJ KO tumor cells displayed higher proportion of cells in GO-G1 associated with a reduced proportion of cells in S-phase in untreated condition compared to Epcam+ and Epcam- cells. At 12 hours following Cisplatin/5FU treatment, the proportion of BrdU positive cells increased in all conditions. However, 24 hours after Cisplatin/5FU administration, the percentage of cells in S phase was strongly reduced in Epcam+ and RhoJ KO tumor cells whereas Epcam- tumor cells continued to synthesize DNA, as revealed by BrdU incorporation (Fig. 5a-b).

We have now presented and discussed these new data in the revised manuscript (Fig 5a-b and P14-15).

b) Fig. 3c shows that RhoJ KD tumor cells contain decreased amount of DNA replication proteins under unchallenged conditions (green pattern in heat map), which is partially restored upon chemotherapy. This implies that the major effect of RhoJ-KD may be in unchallenged S-phase of EMT-tumor cells rather than being a specific consequence of chemotherapy. Consistently, RhoJ-KO tumor formation is strongly impaired (Fig. 2h). Thus, the data argue

for a chemo-independent role of RhoJ in cell cycle progression that may well affect DNA replication. The cell cycle profile does however not show a dramatic change in the fraction of pan S-phase cells. How can this be explained (see also below)? It is in general important to clarify, whether RhoJ KO/KD and OE has any effect on viability, proliferation and cell cycle progression under unchallenged conditions. Does RhoJ also play a role in mitosis or cytokinesis?

We thank the reviewer for these interesting questions. As suggested by the referee, we evaluated the effect of RhoJ on viability, cell cycle progression and proliferation under unchallenged conditions.

Depletion of RhoJ in Epcam⁻ cells decreased proliferation and the growth of cells *in vitro* in untreated conditions (Extended data Fig. 2a and Fig. 5a-b) as well as *in vivo* (Extended data Fig. 3f-g). The viability, based on the level of active caspase-3, was unchanged in untreated condition in RhoJ KD Epcam⁻ cells and RhoJ OE Epcam⁺ cells compared to control cells (Fig. 2e and g). In contrast, RhoJ KO Epcam⁻ cells presented a very small increase in active caspase-3 level compared to WT Epcam⁻ cells in untreated condition (Fig. 2m).

We have now presented and discussed these new data in the revised manuscript (Fig 2m, 5a-b and extended data Fig. 2a and 3f-g and P8-9 and 14-15).

c) Chemotherapy with Cisplatin/5-FU induces DNA replication stress, triggering a DDR to activate the intra-S checkpoint, causing slowing of DNA synthesis or even its complete abolishment by replication fork stalling and inhibition of (late and backup) origin firing. For the cell cycle analysis in Fig. 6 a and b, the scheme shows that the BrdU pulse given to assess BrdU incorporation (hence DNA synthesis), is given 30 mins prior to sampling. The data shows that chemotherapy does not even slightly downregulate BrdU incorporation rate in any of the cell types, but even increases the fraction of BrdU⁺ cells within 12 hours (particularly in EPCAM-tumor cells). This means there must be massive global DNA synthesis maintained (even induced) in EPCAM-negative tumor cells at 12, and 24 hours of chemotherapy. Actually, at 12 hours into chemotherapy, the EPCAM⁻ tumor cells accumulate cells that incorporate BrdU within 30 minutes (as compared to untreated cells). How can this stunning effect be explained? BrdU pulsing 30 mins prior to sampling should cause S-phase cells with strongly reduced DNA synthesis rates, particularly those with chemotherapy-induced (global) replication fork stalling, to become BrdU negative by falling below detection limit, thus eventually being gated as G1/G2. The authors are kindly asked to comment and to verify the treatment scheme in Fig. 6b.

We are sorry for being unclear. We agree with the reviewer that chemotherapy with Cisplatin/5-FU induces DNA replication stress, triggering a DDR to activate the intra-S checkpoint, causing slowing of DNA synthesis or even its complete abolishment by replication fork stalling and inhibition of (late and backup) origin firing.

BrdU was administered 30 min prior to sampling as described in the methods. To assess DNA content and BrdU incorporation, we performed BrdU/7AAD FACS analysis before and at

different time points after Cisplatin/5FU administration. The overall incorporation of Brdu was reduced 12h after chemotherapy as shown by the decrease in fluorescence intensity of the cell population gated in S-phase (Fig. 5a) suggesting that the cells are allowed to enter the S-phase but present a slowdown in S-phase progression. To confirm this point, we have assessed the level of EdU incorporation using immunofluorescence microscopy and confirmed the reduction in the intensity of Edu in Epcam+, Epcam- and RhoJ KO cells 12h after chemotherapy administration (Extended data Fig. 5a-b). Moreover, we also found that the speed of the fork progression was decreased upon chemotherapy (Fig. 5f-g). Thus, the increased proportion of cells in S-phase measured in the lower part of the S-phase gate, with the FACS analysis 12h after chemotherapy administration could be explained by the entrance of the cells in S-phase accompanied by a relative block in S-phase progression reflecting intra S-phase checkpoint activation.

We have now presented and discussed these new data in the revised manuscript (Fig. 5a-b , 5f-g and Extended Data Fig. 5a-b and P14-15).

d) The data on CDK1 Y15 phosphorylation (Fig. 6c) is intriguing, but cannot explain the cell cycle profiles of RhoJ KO tumor cells. The precise function of the different CDKs are complex-dependent and partially redundant. CDK1 is a strong driver of G2-to-M transition promoting mitotic entry, with a smaller impact on G1-to-S transition. An increase in CDK1 Y15 would thus be expected to block entry into mitosis and thus a G2 arrest. RhoJ KO cells contain higher inhibitory Y15-CDK1 after 24 hours (Fig. 6c), but seem to undergo G1 arrest in response to chemotherapy (Fig. 6a). This is controversial and should be clarified. It would imply a more prominent role of CDK1 in G1-S transition (over mitotic entry) in the chemotherapeutic response of the tumor cells used in this study. To gain more insights into these essential mechanistic aspects, the activity of CDK 2,4,6 as predominant G1-S-transition drivers, should be analyzed to clarify the cell cycle progression phenotype of RhoJ KO tumor cells and the chemo-sensitization.

We thank the reviewer for these interesting questions. We agree with the reviewer that the data on CDK1 Y15 phosphorylation are not sufficient to explain the cell cycle profiles of RhoJ KO tumor cells and that the precise function of the different CDKs are complex-dependent and partially redundant. An increase in CDK1 Y15 is expected to block entry into mitosis and thus a G2 arrest but has also been demonstrated to play a role in late S-phase progression. As requested by the reviewer, we assessed whether the regulation of cell cycle induced by chemotherapy was associated with different levels of CDKs in Epcam+, Epcam- and Epcam- RhoJ KD tumor cells (Extended Data Fig. 5c-f). Using western blot, we studied the level of the inhibitory CDK1 phosphorylation on residue Y15, the activating CDK1 phosphorylation on residue T161, the activating CDK2 phosphorylation on residue T160 and the activating CDK4 phosphorylation on residue T172. There is no specific antibody available to detect regulating phosphorylation of CDK6. CDK1 phosphorylation on residue Y15 was transient in Epcam- tumor cells peaking at 12 hours and decreasing thereafter, while it was sustained in Epcam+ and Epcam- RhoJ KO tumor cells. CDK1 phosphorylation on residue T161 was induced upon chemotherapy treatment in all cell types but appeared earlier in Epcam- cells and was already detected at 12h post chemotherapy suggesting that RhoJ negatively regulates CDK1 inhibition and allows S-phase progression and mitosis following

chemotherapy. CDK2 phosphorylation on residue T160 was induced in the same way in all cell types upon chemotherapy treatment allowing S-phase entry. CDK4 phosphorylation on T172 was sustained at 12h in Epcam- cells promoting G1-S transition while it was mainly detected in untreated Epcam+ cells and almost undetectable in RhoJ KO cells in control condition and following chemotherapy which could explain the increase proportion of cells in G0 in RhoJ KO cells. The overall signal of CDK6 was increased in Epcam+ cells compared to Epcam- and RhoJ KO cells.

We have now presented and discussed these new data in the revised manuscript (Extended Data Fig. 5c-f and P15-16).

e) On page 15, the authors state that RhoJ is `activating both homologous recombination and non-homologous end-joining DNA repair mechanisms. This conclusion is not justified by the provided data and seems to be wrong. The presented data in figure 4d document reduced 53BP1 foci in the presence of induced RPA and RAD51 foci in RhoJ KO cells, indicating that non-homologous end joining (NHEJ) is inhibited in the RhoJ KO, but homology-directed repair (HDR) increased. Along that line, 53BP1 foci are peaking at 12h, and decrease to ctrl levels after 24h in all cell lines, suggesting transient engagement of NHEJ. Notably, the peak of 53BP1 foci is highest in EPCAM- cells, arguing for preferential NHEJ usage. This altogether suggests that RhoJ may play a role in double strand break repair pathway choice (e.g. by controlling end-resection) favoring NHEJ over HDR, despite the relative enrichment of DNA replication factors and DNA synthesis. What is the precise function of RhoJ here? This must be adequately discussed, notably in conjunction with the cell cycle profiles.

We agree with the reviewer that the peak of 53BP1 in Epcam- cells support the notion that the Epcam- cells activates NHEJ. However, we also found an increase in the number of Rad51 and RPA2 foci at 12h following chemotherapy in Epcam- cells, supporting the notion that both NHEJ and HR are active in Epcam- cells. We have discussed this point in the revised manuscript.

Minor points:

1) Complete lists of differential genes or proteins transcriptome and mass spectrometry data should be provided. In addition, showing comprehensive (e.g. GSEA, GO/KEGG, pathway enrichments analyses) may support the authors conclusions. For example, it is difficult to evaluate, how the modest downregulation of factors linked to actin reorganization justify the conclusion that there is a strong biological impact on actin cytoskeleton attributable to the loss of RhoJ?

We agree with the reviewer. As suggested by the reviewer, we have now made the transcriptome raw data accessible as download file from GEO :GSE205985. The mass spectrometry proteomics data have been deposited to the ProteomeXchange Consortium via the PRIDE partner repository with the dataset identifier PXD025737. (login username: reviewer_pxd025737@ebi.ac.uk and password: GKfBF90U).

We have added in the supplementary file an excel file with the normalized dataset of the RNA sequencing (Sourcedata_Fig2a_Fig3ab_RNAseq) and two excel files with the differentially regulated proteins identified with the LC-MS/MS analysis (Sourcedata_Fig3c_HeatMap and Sourcedata_Fig6a_APMS). Furthermore, we have performed gene Ontology analysis using the Enrichr online software and the analysis now appears in the supplementary figures (Extended data Fig.4a-d)

2) The authors state that skin tumorigenesis in Lgr5CreER/KRASG12D/p53cKO/Rosa-YFP mice is induced by intraperitoneal injection of Tamoxifen. LGR5 is strongly expressed in (tumor-initiating) stem cells in the gastrointestinal system. KRAS mutation and p53 loss in these cells could cause tumor formation in the gut, and thus influence general health and chemotherapy response. Therefore, it should be clarified/stated, whether there is gastrointestinal tumor formation in the used mice and, whether this may be changed upon RhoJ KO upon systemic Tamoxifen exposure.

We thank the reviewer for this question. We have used these mice for multiple studies and projects ongoing in the lab for the last 10 years (Lapouge et al., PNAS, 2011; Latil et al., Cell Stem Cell, 2017; Nassar et al., Nat Med, 2015; Pastushenko et al., Nature, 2018; Pastushenko et al., Nature, 2021; Revenco et al., Cell Rep, 2019) and never observed gastrointestinal tumor formation. Although KRasG12D can confer a biased in self-renewing division in intestinal stem cells, mutations activating Wnt signaling (APC deletion or expression of constitutive active beta-catenin) is required to induce intestinal tumor (Barker et al., Nature, 2009; Di Nicolantonio & Bardelli, Clin Cancer Res, 2013; Feng et al., Gastroenterology, 2011; Haigis et al., Nat Genet, 2008; Hung et al., PNAS, 2010; Martin et al., Clin Cancer Res, 2013; Nakayama & Oshima, J Mol Cell Biol, 2019; Sansom et al., PNAS, 2006). As suggested by the reviewer, we have assessed whether macroscopic or microscopic tumors could be observed following KRasG12D and p53 deletion in Lgr5 expressing cells following tamoxifen administration. We did not find macroscopic intestinal tumors in mice in WT and RhoJ KO harboring skin SCC. Furthermore, microscopic examination did not reveal the presence of gut tumors. 5 animals were examined per condition. The animals were sacrificed when skin SCC size reaches the limit imposed by the ethical protocol, which corresponds to maximum 5 months after tamoxifen induction. We do not exclude that this model could develop gut tumor at much later timepoint than analyzed in this manuscript.

We have stated this point in the method sections.

3) It is unclear how many different shRNAs against RhoJ have been used, as one shRNA is shown in the figures, whereas several of them are provided in the material section. It should be clarified, how many, and which shRNA have actually been used.

We apologize for being unclear regarding the different shRNAs against RhoJ that have been used. Three different shRNA were used at the same time to target the same gene. Cell lines were coinfecting with a mix of lentiviruses from three separated transfections.

4) IPOND-MS has not been performed, but data from a published study have been used. This is legitimate, but in its current wording in the main text it is misleading, "...following replicative stress, as determined by iPOND..." Although the reference is provided in the end, it should be clarified in the text as well to avoid confusion.

We apologize for being unclear. To avoid confusion we have modified the text as follows :
“following replicative stress, as determined by previously published iPOND”

5) The authors state that “More importantly, the proteins differentially regulated by RhoJ KD were strongly enriched on newly replicated DNA or recruited to origin of replications following replicative stress, as determined by iPOND.” It is important to avoid confusion here, as iPOND as such (without special treatments of the cells) cannot clearly distinguish between origins of replication and sites of ongoing replication.

We apologize for being unclear. We thought we made the distinction in our sentence, but we are unsure to understand the comment of the reviewer.

Referee #4 (Remarks to the Author):

The Blanpain group has previously identified key steps in epithelial to mesenchymal transitions within the generation of cutaneous squamous cell carcinoma. They used the same methods here to identify populations of cells that are resistant to chemotherapy, namely the Epcam⁻ cells in the nascent tumor. They started by inducing tumorigenesis with activation of Kras and deletion of p53, and then treated with 5FU chemotherapy. They quantified the fraction of Epcam⁺ cells in tumors that responded to chemo versus those that did not, and found a tight correlation with Epcam positivity. They identified RhoJ specifically as highly upregulated in resistant cells as opposed to all other Rho isoforms. Experimentally they then deleted RhoJ while inducing tumorigenesis and found an abrogation of tumor appearance in terms of number of SCC and fraction of tumor-free mice in the absence of RhoJ. While there is numerous studies implicating RhoJ in a variety of tumors, cutaneous SCC has not been linked to RhoJ as far as this reviewer can discern.

Next, they profiled tumor samples with and without RhoJ expression and identified a number of pathways differentially expressed. Pathways enriched included many related to cell division, while those suppressed were related to DNA repair and apoptosis. This second finding led the authors to explore a role in DNA repair for RhoJ, looking at DNA damage with a variety of methods in Epcam⁺ vs Epcam⁻ in RhoJ ko tissues.

Finally, the authors attempt to link well-established roles for Rho proteins in cytoskeletal regulation to DNA-repair. They looked at the organization of nuclear actin filaments in the tumor cell types described above, and found a correlation between nuclear actin filament and loss of RhoJ expression, and suggested that RhoJ acts to regulate resistance to chemotherapy by regulating DNA-repair through control of nuclear filaments.

While a role for RhoJ has been described in many other studies in non-SCC cancers, only one previous study has implicated RhoJ and CDK in regulation of DNA-repair in cancer progression. The observations made here are interesting and well-documented, but there are some additional experiments that could improve the study.

1, throughout most of the manuscript, it is surprising that the authors did not appear to measure cell proliferation. Tumor expansion occurs upon induction of proliferation, or inhibition of apoptosis, or both. In figure 1, they measured apoptosis but not proliferation, nor did they measure tumor size across the various conditions, instead just described proportions of cells. Without a measure of proliferation or tumor size, the differences shown in proportions lack context.

We thank the reviewer for this interesting suggestion. As suggested by the reviewer, we have now assessed proliferation across the various conditions. We found that K14+ epithelial tumor cells were more proliferative than K14- mesenchymal tumor cells in RhoJ WT SCCs from Lgr5CreER/KRasG12D/p53cKO/Rosa-YFP mice as shown by the increase in Ki67-positive cells. Upon RhoJ deletion, proliferation of K14- mesenchymal tumor cells was decreased as shown by the reduction of Ki67-positive cells in RhoJ KO SCCs from Lgr5CreER/KRasG12D/p53cKO/ RhoJcKO/Rosa-YFP mice.

We have now presented and discussed these new data in the revised manuscript (Extended Data Fig. 3f-g and P9).

2, in figure 2 the authors describe the # of SCCs and % of tumor free mice. This cancer model typically produces papilloma and scc, which have different attributes beyond just malignant vs benign. It would be important to understand whether RhoJ ko cells fail to produce scc and/or papilloma. In addition, the authors did not show any data to prove the RhoJ ko was actually successful. They showed data for shRNA in cell lines, but not the genetic ko. It would also be helpful to see where RhoJ is expressed in tumors vs normal skin. Also, from 2J, there are some tumors with Epcam+ cells, what happens to these tumors? are they more or less cancerous? And finally, again, what happened with proliferation in these tumors, can the phenotype only be explained by measuring apoptosis?

We thank the reviewer for this question. In contrast to chemical induced carcinogenesis, such as DMBA/TPA, the genetic mouse skin SCC induced by KRas/G12D and p53 deletion does not give rise to papilloma that progress to carcinoma but instead leads directly to the formation of malignant carcinoma (Lapouge et al., PNAS, 2011; White et al., PNAS, 2011, Latil et al., Cell Stem Cell, 2017; Mauri et al., Nature Cancer, 2021). While tumor initiation was impaired upon RhoJ deletion as shown with the difference observed in term of tumor number per mice and tumor latency (Fig. 2h), we did not observe the appearance of papilloma in RhoJ KO mice suggesting that the malignant progression of the tumors was not affected.

We agree with the reviewer that it is important to demonstrate that RhoJ deletion was achieved by tamoxifen administration. We have now shown by PCR the validation of RhoJ deletion in tumors from Lgr5CreER/KRAS^{G12D}/p53^{cKO}/RhoJ^{cKO}/Rosa-YFP mice. The validation of KO tumors was assessed by PCR using primers combination that allows us to identify wild-type and floxed allele as described (Kim et al., Cancer Cell, 2014). The absence of the floxed allele associated with the presence of the Cre recombinase in tumors generated from RhoJ KO mice confirms the success of the recombination of the allele upon tamoxifen administration (Extended Data Fig. 3e).

We agree with the reviewer that it would be helpful to see where RhoJ is expressed in tumors vs normal skin. Unfortunately, there is no specific antibody that works on immunofluorescence in mouse tissues. By RNA-seq, we found that RhoJ is expressed in normal interfollicular epidermis and hair follicle cells with higher expression in hair follicle, which is 2-3 times lower than in Epcam- tumor cells (our unpublished data).

We have now presented and discussed these new data in the revised manuscript (Fig. 2h and Extended Data Fig. 3e and P9).

3, Do these tumors produce metastasis? if so, what is the effect of the various chemotherapeutics? what is effect of RhoJ ko?

We agree with the referee that it would be interesting to elucidate the role of RhoJ in metastasis formation. Here, we showed that RhoJ shRNA knockdown decreases cell migration in wound scratch assays in Epcam- tumor cells (Extended data Fig. 2b-c) consistent with previous studies of endothelial cells, breast cancer and melanoma cell lines (Chen et al., Front Cell Dev Biol, 2020; Kaur et al., Arterioscler Thromb Vasc Biol, 2011; Ho et al., Pigment Cell Melanoma Res, 2013), suggesting that RhoJ could play a role in metastasis formation. The difference in the number of tumors per mice between WT and RhoJ KO precludes us to draw any firm conclusion concerning the intrinsic property of RhoJ in regulating metastasis formation. We believe that the precise investigation of the role of RhoJ in metastasis formation and the understanding of the underlying mechanisms is a study on its own that is probably beyond the scope of this current study.

4, the authors performed gene expression profiling, so they could in fact use these data to look at proliferation. Indeed, the major difference described in Fig 3 was elevated expression of genes related to proliferation in the WT suggesting the RhoJ expressing cells are much more proliferative, beyond the effect of apoptosis shown in other figures.

We agree with the reviewer that a reduction in expression levels of genes and proteins related to proliferation was shown by the transcriptomic (Fig. 3a and Extended data Fig. 4a) and the proteomic profiling (Fig. 3c and Extended data Fig. 4c) of Epcam- cells transfected with RhoJ shRNA compared to Epcam- control cells. As suggested by the reviewer, we have now directly assessed proliferation across the various conditions. We found that RhoJ KD in EMT tumor cells was associated with a decreased in proliferation and a slight reduction in cell growth *in vitro* (Fig. 5a-b and Extended Data Fig. 2a) and that RhoJ KO in EMT tumor cells was associated with a decrease of proliferation *in vivo* as shown by the reduction of Ki67-positive cells (Extended Data Fig. 3f-g).

We have now presented and discussed these new data in the revised manuscript (Fig. 5a-b, Extended Data Fig. 2a, Extended Data Fig. 3f-g and P8-9 and 14-15).

5, in Figure 4, the authors explore dna repair. The quantification of panel C seems to incongruent with the facs plots in terms of the numbers shown in the boxes versus plotted on the graph. Also, the patterns of dna damage between C and D appear somewhat different, is there anything to be learned about the nature of the dna damage in the cells from this discrepancy?

We thank the reviewer for the question. There is a good correlation between the number of gamma-H2AX positive cells shown in the FACS and the number presented in the bar chart.

Gamma-H2AX expression and foci is a sign of unrepaired double strand break. We have now clarified this point in the revised manuscript.

6, Figure 5 has some interesting images, but is not well labeled. Which is wt? ko? This interesting observation that warrants significant further exploration. Where is RhoJ located in the nucleus to regulate this? What is the pattern in Epcam+ vs Epcam-? what does the pattern mean for DNA repair? if one uses orthogonal methods to abrogate nuclear fibers, does that influence dna repair?

We thank the reviewer for the comments. We apologize for the poorly labelled figure. We have now reformatted the figure showing representative images of the different nuclear actin filament pattern in Epcam+, Epcam- and RhoJ KO Epcam- cells in control and treated conditions (Fig. 6d).

To assess the localization of RhoJ in the nucleus, we used transduced Epcam- tumor cells with lentivirus expressing HA-tagged RhoJ and we performed immunofluorescence analysis. Immunostaining of the HA-tag showed peri and intra-nuclear accumulation of RhoJ in response to chemotherapy treatment indicating a nuclear function of RhoJ in response to chemotherapy (Extended Data Fig. 3b).

As suggested by several referees, we have now performed many new experiments to decipher the mechanisms by which RhoJ mediates the resistance to chemotherapy in EMT tumor cells. To understand how RhoJ regulates chemoresistance, we sought to identify the proteins interacting with RhoJ. To this end, we performed affinity-purification followed by mass spectrometry (AP-MS) using HA pull-down of Epcam- cells expressing 3HA-tagged RhoJ in untreated conditions and 12 hours after Cisplatin/5FU administration. We identified 118 proteins that were significantly enriched in untreated HA-tagged RhoJ expressing cells and 98 proteins that were significantly enriched in treated HA-tagged RhoJ expressing cells. Among them we found previously identified RhoJ interacting proteins such as GPRC5A (Bagci et al., Nat Cell Biol, 2020). Interestingly we identified, upon chemotherapy treatment, an enrichment of protein implicated in the regulation of actin filament dynamics (FLNB and TLN1) and IPO9, a protein that regulates nuclear import of actin (Dopie et al., PNAS, 2012) in RhoJ protein complex. KD of IPO9 has been shown to decrease nuclear actin filament and impairs DNA repair (Belin et al., elife 2015). These findings suggest that RhoJ physically interacts with actin regulator IPO9. To confirm this possibility, we performed co-immunoprecipitation using antibodies against endogenous IPO9 protein, followed by western blot revealing the presence of RhoJ-3HA protein. This Co-IP experiment confirmed that IPO9 is found in a protein complex with RhoJ (Fig. 6b), further suggesting that RhoJ regulates the response to chemotherapy through the regulation of nuclear actin, which have previously been shown to control DNA repair (Andrin et al., Nucleus, 2012; Belin et al., Elife, 2015; Caridi et al, Nature, 2018; Caridi et al., Nat Cell Biol, 2019; Schrank et al., Nature 2018; Hurst et al., Trends Cell Biol, 2019; Hyrskyluoto et al., Curr Opin Cell Biol, 2020; Lamm et al., Nat Cell Biol, 2020).

We have tested whether the regulation of nuclear actin by RhoJ regulates DNA replication and DNA repair. Using actin polymerization inhibitor LatrunculinB, we found that inhibiting actin polymerization, decreased the survival of Epcam- tumor cells but not the Epcam+ and RhoJ KD Epcam- (Fig. 6f), showing that actin polymerization acts downstream of RhoJ in regulating cell survival following chemotherapy. We also found that inhibition of actin

polymerization inhibited the recruitment of new origin of DNA replication and the DNA repair capacity of Epcam- tumor cells in a RhoJ dependent manner (Fig 6g and 6h). These new data now allow to connect all the data together and identify a clear mechanism by which RhoJ regulates resistance to chemotherapy in EMT tumor cells.

To define which subset of the F-actin networks was relevant for RhoJ expressing cells, we treated the cells with chemical inhibitors of ARP2/3 (CK666) which prevents the formation of F-actin branched-network, N-Wasp (Wiskostatin) which prevents the activation of ARP2/3 complex and formins (SMIFH2) which preferentially inhibits the formation of F-actin bundles. Only SMIFH2 combined with Cisplatin/5FU increased the level of cell death in Epcam- cells following chemotherapy similarly to Latrunculin B suggesting that RhoJ-mediated nuclear actin polymerization in response to chemotherapy was dependent of formins and independent of ARP2/3 complex (Fig. 6f and Extended Data Fig. 7c). Treating Epcam-cells with Latrunculin B or SMIFH2 alone did not induce cell death suggesting a specific role of nuclear actin polymerization in response to replicative stress (Extended Data Fig. 7d).

To assess the role/function of the different expression nuclear actin patterns on DNA damage response and therapy resistance, we quantified the different patterns of nuclear actin filaments observed that can be classified in five categories. Upon chemotherapy treatment, we detected strong induction of Pattern 2 characterized by elongated actin filaments in Epcam- cells compared to the induction of Pattern 1 characterized by thin, short and branched actin filaments in Epcam+ cells. Pattern 1 and pattern 3, characterized by hairy filaments which are short filaments with a dense and multipolar organization, were predominant in RhoJ KO cells independently of the presence of chemotherapy (Fig. 6d-e). When chemotherapy was combined with Latrunculin B or SMIFH2, the nuclear actin filaments became disorganized with a predominance of Pattern 4 characterized by thick and twisted actin and pattern 5 characterized by a severe disruption of actin filaments and was associated with increased cell death in Epcam- cells supporting the notion that the chemotherapy-induced long nuclear filament organization involves RhoJ and mediates resistance to chemotherapy (Fig. 6d-f).

We have now presented and discussed these new data in the revised manuscript (Fig.6 and Extended Data Fig. 3b and 7 and P18-21).

7, the authors express most of the results as Epcam+ vs Epcam- vs RhoJ-Epcam-. But if there are Epcam+ cells in RhoJ ko tumors, what is the effect of loss of RhoJ on these cells? These cells have been left out of all the quantifications. Since they started by showing that RhoJ was induced in Epcam- cells, perhaps there is no effect, but this would be a nice confirmation of that idea.

We agree with the reviewer that analyzing the Epcam+ cells from RhoJ KO tumors would confirm the idea that loss of RhoJ has no effect on Epcam+ cells since we started by showing that RhoJ was induced in Epcam- cells. Indeed, the administration of Cisplatin/5FU to RhoJ KO mice harboring tumors showed that RhoJ KO tumors presented increased cell death in Epcam- cells compared to RhoJ WT tumors whereas the level of cell death was similar in Epcam+ cells from treated RhoJ KO mice and RhoJ WT mice (Fig. 2l).

We have now presented and discussed these new data in the revised manuscript (Fig 2l and P9).

8, in figure 6, there is again discrepancy between the facs plots and the quantification, perhaps the reviewer just doesnt understand how this was carried out. Regardless, here the authors look at cell division and see no difference between genotypes in homeostatic conditions. If this is the case in tumors, then why was the overall phenotype so different? In vivo, is there significant genotoxic stress as has been setup in the experimental conditions here?

We apologize for the apparent discrepancy between the FACS plots and the quantification. The FACS plots correspond to one of the experiments performed and is representative of the cell cycle profile of Epcam+, Epcam- and Epcam- RhoJ KO BrdU 7AAD labelled 12h and 24h after Cisplatin/5FU treatment. We have now performed additional experiments and we have reformatted the graphics (Fig. 5b) to better visualize the individual results.

We apologize for the erroneous affirmation saying that “No difference in cell cycle distribution was observed between Epcam+, Epcam- and RhoJ KO tumor cells in untreated condition.” After analyzing more samples, we have now performed a complete analysis of cell cycle profile and we have made the required corrections to the manuscript. Cell cycle assessment using BrdU/7AAD FACS analysis revealed that RhoJ KO tumor cells displayed higher proportion of cells in GO-G1 associated with a reduced proportion of cells in S-phase in untreated condition compared to Epcam+ and Epcam- cells which is line with the phenotype observed *in vivo*.

We found that there is a very good concordance between the short and the long-term response to Cisplatin/5FU study performed *in vivo*. The growth of the SCCs from mice treated with three to four weeks of Cisplatin/5FU administration at the standard doses found in the literature (3,5mg/kg of Cisplatin and 15 mg/kg of 5FU administered intraperitoneally) showed a clear difference in tumor growth between WT and RhoJ KO tumors (Fig. 1b and 2n). The level of cell death induced in FACS isolated YFP+ Epcam+ and Epcam- cells from 24h Cisplatin/5FU treated SCCs was significantly increased compared to control conditions as determined by the quantification of the percentage of activated caspase-3 positive cells (Fig 1f).

We have now presented and discussed these new data in the revised manuscript (Fig 1b, 1f, 2n, 5a-b and P5-6,10 and 14-15).

Reviewer Reports on the First Revision:

Referees' comments:

Referee #1 (Remarks to the Author):

In this revised manuscript, the authors have taken a serious approach to the comments raised in previous review. The majority of my points have been addressed in full, and there is significant additional work. My main previous question was.. how exactly is RhoJ doing this? While this is still not completely resolved, the authors have made a significant effort to address this. It is clearly complicated, but I think that the story is now at an appropriate point to be considered complete, and worthy of publication at this level.

Referee #2 (Remarks to the Author):

I would like to acknowledge the huge amount of work that the authors have put into revision of their manuscript. Since their last submission, they have added a large amount of (high quality) data that address all of my major concerns. In particular, the new data make a much stronger case for involvement of the replication stress response and of nuclear actin polymerization in RhoJ-dependent chemoresistance.

I have only a few additional points that I feel should be addressed before publication:

- The LatB-only and SMIFH2-only controls are essential for interpretation of the data, which is why I feel they should be moved to the main figure.
- Page 20 "Immunostaining of nuclear actin with phospho-Histone H3 and PCNA showed that nuclear actin filaments are expressed in G0/G1 and S-phases of the cell cycle (Extended Data Fig. 7a-b).": First, the claim that the nuclear actin filaments are expressed in G0/G1 is not accurate since the authors do not present a co-staining of PCNA and pH3 (and, hence, do not show PCNA/pH3 double-negative cells). Second, the data needs to be quantified.

Referee #3 (Remarks to the Author):

All my comments and concerns have been addressed, mostly by a number of additional experiments. In summary, this led to clear improvements of the manuscript. However, two critical points remain to be addressed.

1) Regarding Comment No1:

The authors provided new data to elucidate the mechanism of how RhoJ acts on nuclear actin to regulate DNA replication and chemotherapeutic response. However, despite the new data, the direct function of RhoJ in regulating DNA replication and repair via nuclear actin still appears vague. The new data corroborate that RhoJ affects nuclear actin and affects DNA replication, but remains correlative. Thus, the mechanistically required link between these two well-described effects has still

not been fully established yet. Specifically, it is essential to demonstrate unambiguously that the cells that are actively undertaking DNA replication display (one or more specific patterns of) nuclear actin and that this pattern in these cells is dependent on RhoJ (i.e. knockdown or knockout) in unchallenged and in chemotherapy-treated conditions. It is also important to determine the frequency of a certain nuclear actin pattern in the different cell cycle phases (most importantly in S-phase).

Indeed the authors approached this question using a combination of the nuclear actin staining with PCNA and phospho-H3 (p-H3) immunofluorescence (IF). However, PCNA as single marker and p-H3 do not allow gating for S-phase cells. Unlike the authors state, PCNA as such marks both G1 (mainly late G1) and S-phase cells, which might perhaps be distinguished by the staining pattern of PCNA (localization, pan staining, big or small foci), while p-H3 marks cells in late G2 and mitosis. Thus, PCNA positive cells that are negative for p-H3 do not necessarily reflect S-phase cells, as concluded by the authors. These might be (late) G1 cells and/or S-phase cells, leaving the PCNA-positive cells as being potentially either (late) G1 or S-phase cells. Collectively, it seems from the provided analysis that the cells presenting nuclear actin are from all cell cycle phases except late G2 and mitosis. In addition to the apparent limited specificity regarding S-phase in this analysis (using PCNA as single marker for DNA synthesis), there is no quantification of the staining, which is however essential when studying sub-fractions of cells. Lastly, only representative images of the chemo-treated cells are provided, making it altogether very difficult to judge the distribution and heterogeneity of cells positive for nuclear actin and their correlation with the used markers. In summary, this is still too vague to justify the conclusion that RhoJ acts on nuclear actin in DNA replicating cells to induce chemoresistance. Without direct proof and quantifications, the molecular function of RhoJ remains unclear.

The authors should demonstrate experimentally that nuclear actin in cells synthesizing DNA is present and reliably detectable in their system and that either its pattern (preferably), or its abundance is under the control of RhoJ. For example, this could potentially be tackled in RhoJ Knockdown/knockout cells versus control cells in the presence/absence of chemotherapy +/- selected actin remodeling conditions by

i) EdU-labeled cells in combination with nuclear actin staining. Notably, visualization of EdU could be achieved by click chemistry to maintain full compatibility with downstream stainings. Therefore, unlike classical staining of BrdU using antibodies, there is no denaturation of the DNA involved or required at all for EdU visualization.

ii) S-phase cells (in comparison to G1 and G2) sorted by FACS followed by IF staining and/or subcellular fractionation and immunoblotting. Notably, PCNA would not be a good marker for S-phase cells. Reporter systems such as Fucci and modifications thereof, e.g. PIP-Fucci (Grant et al., Cell cycle 2018) are available.

iii) if the first two options fail, it may be possible to enrich S-phase cells by cell synchronization (and release, depending on the synchronization technique) before IF staining and/or subcellular fractionation and immunoblotting.

Even if the nuclear actin in S-phase cells cannot be detected and/or the mechanism eventually cannot be elucidated in detail for biological rather than merely technical reasons, all data should be quantified comprehensively, stated and properly discussed.

2) Regarding comment No 3 (and No 5b): The new data using different chemotherapies are very intriguing. In fact, the authors show that RhoJ-KO/KD cells in general are far less viable, notably even

in untreated conditions. This disadvantage seems not to be aggravated upon chemotherapy! Along that line, DNA damage accumulation (γH2AX) shows an identical pattern in response to chemotherapy. Thus, the new data do not support the conclusion that RhoJ KO/KD cells are more sensitive to genotoxic stress and chemotherapy. Directly responding to the statement in the rebuttal letter that the authors “have thus removed this data in the revised manuscript”, I strongly urge these data to be included in the manuscript. This is an important piece of information for interpreting the role of RhoJ that should be shared with the community and be discussed in the manuscript.

In conclusion, I feel that the DDR to chemotherapy should not be spotlighted too much, because several lines of evidence provided by the authors show a general defect of RhoJ-KO cells in cell cycle progression under normal conditions (in vitro and in vivo), while other evidence indicates no major effect on viability under normal conditions (e.g. cleaved caspase 3 positivity). The effect on proliferation seems most likely attributable to problematic progression through unperturbed S-phase due to defects in DNA replication, ultimately manifesting the proliferation defect in vitro and in vivo. Importantly, this defect persists upon chemotherapy without being strongly aggravated (see above). The strong impact of RhoJ in DNA replication and cell proliferation without additional chemotherapy should be more clearly stated, because it documents a general biological function of RhoJ specifically in EPCAM- EMT cells. The discussion on RhoJ's influence on chemotherapy should eventually be refined by emphasizing the effect and its specificity in untreated cells in vitro and in vivo.

Minor comment:

The wording at the beginning of the discussion “the mechanism of...” should be weakened to “one mechanism of...”, as this exclusivity has not been addressed here and should thus not be claimed.

Referee #4 (Remarks to the Author):

The authors have made substantial improvements to the manuscript including new data and improved quantification and clarification. The results presented strongly support the idea that RhoJ plays an important role in chemoresistance in SCC. My only concern is with the language used to describe the results.

1, I am not sure it is fair to say that RhoJ is 'necessary and sufficient' to mediate the chemoresistance. In most of the data presented, manipulation of RhoJ does not completely reverse/induce the chemoresistance. For instance, in Fig 2m, the Epcam- tumors have some response, and this response is boosted in the RhoJ KO, so it would be more accurate to say that RhoJ promotes chemoresistance. In fact, this is how the results are described in the discussion, but in the results section, the 'necessary and sufficient' language is found.

2, The manuscript is framed to suggest that the role of RhoJ is centered on chemoresistance. However, the deletion of RhoJ decreases the number of SCCs overall (regardless of chemo treatment, Fig 2h), so it is also plausible that RhoJ is playing a more general role in biology of the cell including cytoskeleton and dna repair leading to a diminished capacity for transformation, which has

been argued previously for melanoma and other cancers.

3, It would have been nice to see evidence of chemoresistance in vivo with the RhoJ KO. The authors showed that RhoJ KO tumors have a lower volume, and showed chemoresistance by the %of Casp3 cells (Fig 2m, and H2AX in Figure 4), but it would have been more impactful to show for instance Fig 2n with and without chemo treatment over a longer period to determine the consequences on tumor persistence, volume, metastasis etc. Of course, how the drug treatments described in Fig 6 would affect tumor progression/chemoresistance in actual Epcam+/Epcam-/RhoJKO tumors in vivo to determine the overall cancer relevance would also be great, but is perhaps too much to ask at this point. In summary, the authors have shed important new light on mechanism of action for RhoJ in SCC, but more could be done to illuminate the consequences.

Author Rebuttals to First Revision:

Referees' comments:

We thank the referees for their positive comments.

We were delighted that the reviewers found that our revision addressed most of the issues raised during the first round of review.

We have now addressed their remaining comments and performed experiments to address the questions related to the presence of nuclear actin in relation to the cell cycle and performed new experiments to assess the growth of WT and RhoJ KO tumors in vivo.

We hope that we have now addressed all their comments and suggestions.

Referee #1 (Remarks to the Author):

In this revised manuscript, the authors have taken a serious approach to the comments raised in previous review. The majority of my points have been addressed in full, and there is significant additional work. My main previous question was.. how exactly is RhoJ doing this? While this is still not completely resolved, the authors have made a significant effort to address this. It is clearly complicated, but I think that the story is now at an appropriate point to be considered complete, and worthy of publication at this level.

We are grateful to the reviewer for recognizing the significant additional work that was accomplished to answer the comments regarding the complexity of the question and we are delighted that the reviewer thinks that our story is now worthy to be published at this level.

Referee #2 (Remarks to the Author):

I would like to acknowledge the huge amount of work that the authors have put into revision of their manuscript. Since their last submission, they have added a large amount of (high quality) data that address all of my major concerns. In particular, the new data make a much stronger case for involvement of the replication stress response and of nuclear actin polymerization in RhoJ-dependent chemoresistance.

We thank the referee for recognizing the huge amount of work that has been performed to address his/her previous concerns and for his/her additional suggestions.

I have only a few additional points that I feel should be addressed before publication:

- The LatB-only and SMIFH2-only controls are essential for interpretation of the data, which is why I feel they should be moved to the main figure.

We agree with the reviewer. We have now moved the LatB-only and SMIFH2-only controls to the main figures.

These data are now shown in the revised manuscript (Fig. 6i).

- Page 20 "Immunostaining of nuclear actin with phospho-Histone H3 and PCNA showed that nuclear actin filaments are expressed in G0/G1 and S-phases of the cell cycle (Extended Data Fig. 7a-b).": First, the claim that the nuclear actin filaments are expressed in G0/G1 is not accurate since the authors do not present a co-staining of PCNA and pH3 (and, hence, do not show PCNA/pH3 double-negative cells). Second, the data needs to be quantified.

We agree with the reviewer comments. As requested by this reviewer and another one, we assessed the cell cycle state of the tumor cells presenting nuclear actin filaments by performing and quantifying co-immunostaining of nuclear actin, phospho-histone 3 (pH3) and EdU. We used pH3 as a marker of late G2 and mitosis and EdU labelling of the cells, instead of PCNA, to precisely identify the proportion of cells in S-phase since PCNA is also a marker of late G1. We used the Click-iT Plus EdU cell proliferation kit (Thermo Fisher Scientific) which is optimized to preserve GFP fluorescent signal and allowed us to identify the nuclear actin GFP signal of the chromobody that we employed.

We confirmed that nuclear actin filaments are present in G0/G1 and S-phases of the cell cycle as shown by EdU/pH3 double negative cells and EdU positive/pH3 negative cells.

In untreated condition, about 20% of Epcam- cells presenting nuclear actin filaments were EdU positive. At 12 hours following Cisplatin/5FU treatment, the proportion of cells positive for both nuclear actin and EdU increased to 40%. Interestingly the combination of Cisplatin/5FU treatment with Latrunculin B led to a reduction of the proportion of Epcam-cells double positive for nuclear actin filaments and EdU similar to the untreated condition. There was no significant increase of EdU and nuclear actin double positive cells in Epcam-RhoJ KO cells. These data support the notion, that chemotherapy increased the proportion of cells with nuclear actin in proliferating cells in a RhoJ and actin polymerization dependent manner.

These data are now shown in the revised manuscript (Extended Data Fig. 8 a-c).

Referee #3 (Remarks to the Author):

All my comments and concerns have been addressed, mostly by a number of additional experiments. In summary, this led to clear improvements of the manuscript.

We thank the reviewer for the positive assessment of our revised manuscript and the important additional questions and suggestions.

However, two critical points remain to be addressed.

1) Regarding Comment No1:

The authors provided new data to elucidate the mechanism of how RhoJ acts on nuclear actin to regulate DNA replication and chemotherapeutic response. However, despite the new data, the direct function of RhoJ in regulating DNA replication and repair via nuclear actin still appears vague. The new data corroborate that RhoJ affects nuclear actin and affects DNA replication, but remains correlative. Thus, the mechanistically required link between these two well-described effects has still not been fully established yet. Specifically, it is essential to demonstrate unambiguously that the cells that are actively undertaking DNA replication display (one or more specific patterns of) nuclear actin and that this pattern in these cells is dependent on RhoJ (i.e. knockdown or knockout) in unchallenged and in chemotherapy-treated conditions. It is also important to determine the frequency of a certain nuclear actin pattern in the different cell cycle phases (most importantly in S-phase).

Indeed the authors approached this question using a combination of the nuclear actin staining with PCNA and phospho-H3 (p-H3) immunofluorescence (IF). However, PCNA as single

marker and p-H3 do not allow gating for S-phase cells. Unlike the authors state, PCNA as such marks both G1 (mainly late G1) and S-phase cells, which might perhaps be distinguished by the staining pattern of PCNA (localization, pan staining, big or small foci), while p-H3 marks cells in late G2 and mitosis. Thus, PCNA positive cells that are negative for p-H3 do not necessarily reflect S-phase cells, as concluded by the authors. These might be (late) G1 cells and/or S-phase cells, leaving the PCNA-positive cells as being potentially either (late) G1 or S-phase cells. Collectively, it seems from the provided analysis that the cells presenting nuclear actin are from all cell cycle phases except late G2 and mitosis. In addition to the apparent limited specificity regarding S-phase in this analysis (using PCNA as single marker for DNA synthesis), there is no quantification of the staining, which is however essential when studying sub-fractions of cells. Lastly, only representative images of the chemo-treated cells are provided, making it altogether very difficult to judge the distribution and heterogeneity of cells positive for nuclear actin and their correlation with the used markers. In summary, this is still too vague to justify the conclusion that RhoJ acts on nuclear actin in DNA replicating cells to induce chemoresistance. Without direct proof and quantifications, the molecular function of RhoJ remains unclear.

The authors should demonstrate experimentally that nuclear actin in cells synthesizing DNA is present and reliably detectable in their system and that either its pattern (preferably), or its abundance is under the control of RhoJ. For example, this could potentially be tackled in RhoJ Knockdown/knockout cells versus control cells in the presence/absence of chemotherapy +/- selected actin remodeling conditions by

- i) EdU-labeled cells in combination with nuclear actin staining. Notably, visualization of EdU could be achieved by click chemistry to maintain full compatibility with downstream stainings. Therefore, unlike classical staining of BrdU using antibodies, there is no denaturation of the DNA involved or required at all for EdU visualization.
- ii) S-phase cells (in comparison to G1 and G2) sorted by FACS followed by IF staining and/or subcellular fractionation and immunoblotting. Notably, PCNA would not be a good marker for S-phase cells. Reporter systems such as FUCCI and modifications thereof, e.g. PIP-FUCCI (Grant et al., Cell cycle 2018) are available.
- iii) if the first two options fail, it may be possible to enrich S-phase cells by cell synchronization (and release, depending on the synchronization technique) before IF staining and/or subcellular fractionation and immunoblotting.

Even if the nuclear actin in S-phase cells cannot be detected and/or the mechanism eventually cannot be elucidated in detail for biological rather than merely technical reasons, all data should be quantified comprehensively, stated and properly discussed.

As requested by the reviewer, we assessed the cell cycle state of the tumor cells presenting nuclear actin filaments by performing and quantifying co-immunostaining of nuclear actin, phospho-histone 3 (p-H3) and EdU. We used p-H3 as a marker of late G2 and mitosis and EdU labelling of the cells, instead of PCNA, to precisely identify the proportion of cells in S-phase since PCNA is also a marker of late G1. We used the Click-iT Plus EdU cell proliferation kit (Thermo Fisher Scientific) which is optimized to preserve GFP fluorescent signal and allowed us to identify the nuclear actin GFP signal of the chromobody that we employed.

In untreated condition, about 20% of Epcam- cells presenting nuclear actin filaments were EdU positive. At 12 hours following Cisplatin/5FU treatment, the proportion of cells positive for both nuclear actin and EdU increased to 40%. Interestingly the combination of Cisplatin/5FU treatment with Latrunculin B led to a reduction of the proportion of Epcam-cells double positive for nuclear actin filaments and EdU similar to the untreated condition. Moreover, the nuclear actin filaments in Epcam- tumor cells that are EdU positive were mainly in Pattern 1 and 2 in basal conditions and after Cisplatin/5FU treatment. There was no significant increase of EdU and nuclear actin double positive cells in Epcam- RhoJ KO cells.

These data show that chemotherapy increases the proportion of replicative cells with nuclear actin in a RhoJ and actin polymerization dependent manner.

These data have been added to the revised manuscript (Extended Data Fig. 8 a-d).

2) Regarding comment No 3 (and No 5b): The new data using different chemotherapies are very intriguing. In fact, the authors show that RhoJ-KO/KD cells in general are far less viable, notably even in untreated conditions. This disadvantage seems not to be aggravated upon chemotherapy! Along that line, DNA damage accumulation (gH2AX) shows an identical pattern in response to chemotherapy. Thus, the new data do not support the conclusion that RhoJ KO/KD cells are more sensitive to genotoxic stress and chemotherapy. Directly responding to the statement in the rebuttal letter that the authors “have thus removed this data in the revised manuscript”, I strongly urge these data to be included in the manuscript. This is an important piece of information for interpreting the role of RhoJ that should be shared with the community and be discussed in the manuscript.

As suggested by the reviewer, we included to the manuscript the data related to the role of RhoJ in human cancer cell line. To assess the relevance of our findings to human cancer, we generated RhoJ knockdown using short hairpin RNA in MDA-MB-231, a human triple-negative breast cancer cell line presenting EMT which expresses high levels of RhoJ and displaying p53 mutations. RhoJ depletion in MDA-MB231 was associated with a decrease in cell growth. In addition, RhoJ-knockdown MDA-MB-231 cells were more sensitive to cisplatin, doxorubicin and paclitaxel.

These data have been added to the revised manuscript (Extended Data Fig. 4a-d).

In conclusion, I feel that the DDR to chemotherapy should not be spotlighted too much, because several lines of evidence provided by the authors show a general defect of RhoJ-KO cells in cell cycle progression under normal conditions (in vitro and in vivo), while other evidence indicates no major effect on viability under normal conditions (e.g. cleaved caspase 3 positivity). The effect on proliferation seems most likely attributable to problematic progression through unperturbed S-phase due to defects in DNA replication, ultimately manifesting the proliferation defect in vitro and in vivo. Importantly, this defect persists upon chemotherapy without being strongly aggravated (see above). The strong impact of RhoJ in DNA replication and cell proliferation without additional chemotherapy should be more clearly stated, because it documents a general biological function of RhoJ specifically in EPCAM- EMT cells. The discussion on RhoJ's influence on chemotherapy should eventually be refined by emphasizing the effect and its specificity in untreated cells in vitro and in vivo.

We thank the reviewer for his/her suggestions. We have now discussed these points in the revised manuscript.

Minor comment:

The wording at the beginning of the discussion “the mechanism of...” should be weakened to

“one mechanism of...”, as this exclusivity has not been addressed here and should thus not be claimed.

As suggested by the reviewer, we changed the wording “the mechanism of...” by “one mechanism of...”, to avoid misleading claim.

Referee #4 (Remarks to the Author):

The authors have made substantial improvements to the manuscript including new data and improved quantification and clarification. The results presented strongly support the idea that RhoJ plays an important role in chemoresistance in SCC. My only concern is with the language used to describe the results.

We thank the reviewer for the positive assessment of our revised manuscript and the suggestions concerning the need to rephrase some of the results in the manuscript.

1, I am not sure it is fair to say that RhoJ is 'necessary and sufficient' to mediate the chemoresistance. In most of the data presented, manipulation of RhoJ does not completely reverse/induce the chemoresistance. For instance, in Fig 2m, the Epcam- tumors have some response, and this response is boosted in the RhoJ KO, so it would be more accurate to say that RhoJ promotes chemoresistance. In fact, this is how the results are described in the discussion, but in the results section, the 'necessary and sufficient' language is found.

We agree with the reviewer comments and we changed the wording in the results section.

2, The manuscript is framed to suggest that the role of RhoJ is centered on chemoresistance. However, the deletion of RhoJ decreases the number of SCCs overall (regardless of chemo treatment, Fig 2h), so it is also plausible that RhoJ is playing a more general role in biology of the cell including cytoskeleton and dna repair leading to a diminished capacity for transformation, which has been argued previously for melanoma and other cancers.

We agree with the reviewer comments, and we have discussed this point in the revised manuscript.

3, It would have been nice to see evidence of chemoresistance in vivo with the RhoJ KO. The authors showed that RhoJ KO tumors have a lower volume, and showed chemoresistance by the % of Casp3 cells (Fig 2m, and H2AX in Figure 4), but it would have been more impactful to show for instance Fig 2n with and without chemo treatment over a longer period to determine the consequences on tumor persistence, volume, metastasis etc. Of course, how the drug treatments described in Fig 6 would affect tumor progression/chemoresistance in actual Epcam+/Epcam-/RhoJKO tumors in vivo to determine the overall cancer relevance would also be great, but is perhaps too much to ask at this point. In summary, the authors have shed important new light on mechanism of action for RhoJ in SCC, but more could be done to illuminate the consequences.

As suggested by the reviewer, we performed new experiments to assess the growth of WT and RhoJ KO SCCs in the absence or in the presence of long-term chemotherapy treatment *in vivo*. As suggested by the reviewer we found that the RhoJ KO tumors grew a bit more slowly compared to the WT in the absence of chemotherapy. In the presence of chemotherapy, the WT continued to grow although at a slower rate compared to the WT tumors in the absence of chemotherapy, whereas the RhoJ KO tumors did not grow at all in the presence of chemotherapy. These new data better described the consequences of RhoJ KO in relation to tumor growth as suggested by the reviewer.

These new data have now been presented in the revised manuscript and discussed in the text.

These data are now shown in the revised manuscript (Fig. 2n).

Reviewer Reports on the Second Revision:

Referees' comments:

Referee #3 (Remarks to the Author):

All my comments and concerns to the manuscript and its first revision have been addressed in the 2nd revised version, which led to clear improvements of the manuscript.
In my view the manuscript is now acceptable for publication.